# APOE expression and secretion are modulated by mitochondrial dysfunction

Meghan E Wynne[1], Oluwaseun Ogunbona[1,2], Alicia R Lane[1], Avanti Gokhale[1], Stephanie A Zlatic[1], Chongchong Xu[3], Zhexing Wen[1,3,4], Duc M Duong[5], Sruti Rayaprolu[4], Anna Ivanova[5], Eric A Ortlund[5], Eric B Dammer[5], Nicholas T Seyfried[5], Blaine R Roberts[5], Amanda Crocker[6], Vinit Shanbhag[7], Michael Petris[7], Nanami Senoo[8], Selvaraju Kandasamy[8], Steven Michael Claypool[8], Antoni Barrientos[9], Aliza Wingo[4], Thomas S Wingo[4], Srikant Rangaraju[4], Allan I Levey[4], Erica Werner[1]*, Victor Faundez[1]*

[1]Department of Cell Biology, Emory University, Atlanta, United States; [2]Department of Pathology and Laboratory Medicine, Emory University, Atlanta, United States; [3]Department of Psychiatry and Behavioral Sciences, Emory University, Atlanta, United States; [4]Department of Neurology and Human Genetics, Emory University, Atlanta, United States; [5]Department of Biochemistry, Emory University, Atlanta, United States; [6]Program in Neuroscience, Middlebury College, Middlebury, United States; [7]Department of Biochemistry, University of Missouri, Columbia, United States; [8]Department of Physiology, Johns Hopkins University, Baltimore, United States; [9]Department of Neurology and Biochemistry & Molecular Biology, University of Miami, Miami, United States

*For correspondence: ewerner@emory.edu (EW); vfaunde@emory.edu (VF)

Competing interest: The authors declare that no competing interests exist.

**Abstract** Mitochondria influence cellular function through both cell-autonomous and non-cell autonomous mechanisms, such as production of paracrine and endocrine factors. Here, we demonstrate that mitochondrial regulation of the secretome is more extensive than previously appreciated, as both genetic and pharmacological disruption of the electron transport chain caused upregulation of the Alzheimer's disease risk factor apolipoprotein E (APOE) and other secretome components. Indirect disruption of the electron transport chain by gene editing of SLC25A mitochondrial membrane transporters as well as direct genetic and pharmacological disruption of either complexes I, III, or the copper-containing complex IV of the electron transport chain elicited upregulation of APOE transcript, protein, and secretion, up to 49-fold. These APOE phenotypes were robustly expressed in diverse cell types and iPSC-derived human astrocytes as part of an inflammatory gene expression program. Moreover, age- and genotype-dependent decline in brain levels of respiratory complex I preceded an increase in APOE in the 5xFAD mouse model. We propose that mitochondria act as novel upstream regulators of APOE-dependent cellular processes in health and disease.

## Editor's evaluation

This study presents compelling evidence that ApoE is upregulated in various models of mitochondrial respiratory chain dysfunction. The work is of fundamental interest for those studying neurodegeneration and the role of ApoE in Alzheimer's disease; future work will be needed to reveal the molecular basis of this dramatic phenotype.

## Introduction

Mitochondria are necessary for maintaining cellular and organismal health and function by generating energy and serving as hubs for diverse metabolic and signaling pathways (*Nunnari and Suomalainen, 2012*). The majority of the mitochondrial functions described so far are cell-autonomous. However, mitochondria are also capable of influencing cellular function from a distance in a non-cell-autonomous manner. These non-cell-autonomous mechanisms, mostly elicited after cellular or mitochondrial damage, encompass intercellular transfer of mitochondria to secretion of endocrine and paracrine factors (*D'Acunzo et al., 2021*; *Durieux et al., 2011*; *Hayakawa et al., 2016*; *Liu et al., 2021*). These secreted factors include proteins encoded in the nuclear genome, such as alpha-fetoprotein, inflammatory cytokines and type I interferons, and growth factor mitokines (*Bar-Ziv et al., 2020*; *Chung et al., 2017*; *Dhir et al., 2018*; *Durieux et al., 2011*; *Jett et al., 2022*; *Kim et al., 2013*; *Riley and Tait, 2020*; *Shimada et al., 2012*; *West et al., 2015*). A second class of non-cell-autonomous factors are mitochondrially derived peptides, encoded in the mitochondrial genome (*Kim et al., 2017*). Mitokines and mitochondrially derived peptides modulate cell survival, metabolic and lipid homeostasis, body weight, longevity, and aging, a primary risk factor for cognitive decline in humans (*Chung et al., 2017*; *Flippo and Potthoff, 2021*; *Klaus and Ost, 2020*; *Mullican et al., 2017*; *Tsai et al., 2018*). In models of Alzheimer's disease, the mitochondrially derived peptide humanin can reduce apoptosis, inflammation, accumulation of plaque-forming Aβ peptides, and cognitive deficits (*Hashimoto et al., 2001*; *Tajima et al., 2005*; *Yen et al., 2013*). The ability of factors encoded in the nuclear and mitochondrial genomes to regulate inflammation, lipid metabolism, aging, and Alzheimer's disease mechanisms suggests that mitochondrial-dependent modulation of protein secretion could modify neurological disease pathogenesis prior to cell death.

Here, we sought to identify proteins whose expression and secretion are modulated by mitochondrial function through an unbiased interrogation of human transcriptomes and proteomes. We focused our attention on factors whose expression is sensitive to mutations affecting the inner mitochondrial membrane citrate transporter SLC25A1 and the ADP-ATP transporter SLC25A4 (ANT1). We chose these mitochondrial transporters because they have been genetically implicated in neurodevelopment, brain metabolism, psychiatric disease, and neurodegeneration (*Balaraju et al., 2020*; *Chaouch et al., 2014*; *Edvardson et al., 2013*; *Gokhale et al., 2019*; *Akita et al., 2018*; *Lin-Hendel et al., 2016*; *Nota et al., 2013*; *Rigby et al., 2022*; *Siciliano et al., 2003*). For example, SLC25A1 is a causal gene in two genetic diseases: a severe neurometabolic disease (combined D-2- and L-2-hydroxyglutaric aciduria) and a congenital myasthenic syndrome presenting with intellectual disability (*Balaraju et al., 2020*; *Chaouch et al., 2014*; *Nota et al., 2013*) (OMIM 615182–618197). In addition, SLC25A1 is part of the chromosomal interval deleted in 22q11.2 deletion syndrome, a microdeletion syndrome associated with neurodevelopmental, psychiatric, and neurodegenerative diseases (*Butcher et al., 2013*; *Schneider et al., 2014*; *Zinkstok et al., 2019*). SLC25A1 has been implicated as a hub factor underlying a mitochondrial protein network, which includes SLC25A4, that is disrupted in 22q11.2 deletion syndrome cells (*Gokhale et al., 2019*). Since SLC25A1 and SLC25A4 coprecipitate (*Gokhale et al., 2019*), we hypothesized the existence of common downstream secretory and mitochondrial targets elicited by their mutation. We discovered that loss of SLC25A1 or SLC25A4 affected the secreted proteome as well as the mitochondrially annotated proteome. Apolipoprotein E (APOE) was among the secreted factors whose expression was increased in both SLC25A1 and SLC25A4 mutants. We focused on APOE since it is the main carrier of lipids and cholesterol in the brain (*Mahley, 2016*) and it is tied to cognitive function, neuroinflammation, and neurological disease risk (*Belloy et al., 2019*; *Lanfranco et al., 2021*; *O'Donoghue et al., 2018*; *Parhizkar and Holtzman, 2022*). Importantly, the APOE4 allele is known as the strongest genetic risk factor for sporadic Alzheimer's disease (*Belloy et al., 2019*). We found that APOE expression was increased by mutations of mitochondrial SLC25A transporters, which indirectly compromised the integrity of the electron transport chain, and was also increased by directly mutagenizing either assembly factors or subunits of complexes I, III, and IV of the electron transport chain. While the APOE4 allele is thought to cause mitochondrial dysfunction in Alzheimer's disease (*Area-Gomez et al., 2020*; *Chen et al., 2011*; *Mahley, 2023*; *Orr et al., 2019*; *Tambini et al., 2016*; *Yin et al., 2020*), our study places mitochondria upstream of APOE, uncovering a novel function for these multifaceted organelles.

## Results

### Genetic disruption of inner mitochondrial membrane transporters alters the secretome

Our goal was to identify secreted factors whose expression is modulated by genetic defects in nuclear-encoded mitochondrial genes. We hypothesized that changes in the secretome would affect the capacity of conditioned media to support cell growth in a genotype-dependent manner. Thus, we applied conditioned media from wild-type (*SLC25A1+*) and SLC25A1-null HAP1 cells (*SLC25A1Δ*) to cells from both genotypes and measured cell growth. We used this near-haploid human leukemia cell line since it has a short doubling time, and thus rapid protein turnover, making it well-suited to rapidly respond to changes in subproteomes, such as the secretome and mitoproteome. We dialyzed conditioned media from wild-type and *SLC25A1Δ* cells to exclude effects of metabolites, pH, and small peptides present in media (*Figure 1A*). Dialyzed conditioned media from wild-type and *SLC25A1Δ* cells supported wild-type cell growth (*Figure 1A*). Wild-type cells similarly responded to dialyzed media from both genotypes, increasing growth by 50% as compared to non-dialyzed media (*Figure 1A* compare columns 1, 3 and 2, 4). In contrast, while *SLC25A1Δ* cells fed with wild-type dialyzed conditioned media doubled in number (*Figure 1A*, compare columns 5 and 7), dialyzed conditioned media from *SLC25A1Δ* cells fed onto themselves did not support their growth as compared to media from wild-type cells (*Figure 1A* compare columns 7–8 and 6–8). These results suggest that wild-type cells and *SLC25A1Δ* cells condition media differently.

To identify compositional differences between wild-type and *SLC25A1Δ* conditioned media, we analyzed the proteome and transcriptome of *SLC25A1Δ* cells (*Figure 1B, D–F–*). Fetal bovine serum in media prevented us from a direct analysis of the conditioned media by mass spectrometry. We annotated the *SLC25A1Δ* proteome and transcriptome with the human secretome database (*Uhlén et al., 2019*) and the Mitocarta 3.0 knowledgebase (*Rath et al., 2021*) to comprehensively identify differences in secreted factors and the consequences of the *SLC25A1* mutation on mitochondria. We simultaneously analyzed the proteome and transcriptome of *SLC25A4Δ* cells to determine whether changes in the *SLC25A1Δ* proteome and transcriptome resulted specifically from the loss of SLC25A1 or could be generalized to another inner mitochondrial membrane transporter (*Figure 1C and G*). We selected SLC25A4, as it encodes an ADP-ATP translocator that interacts with SLC25A1 (*Gokhale et al., 2019*). Tandem mass tagging mass spectrometry and RNAseq revealed that *SLC25A1Δ* cells underwent more extensive changes of their proteome and transcriptome than *SLC25A4Δ* cells (compare *Figure 1B* with C and F with G). For example, 668 proteins significantly changed their expression in *SLC25A1Δ* cells compared to 110 proteins in *SLC25A4Δ* cells (log2 fold of change of 0.5 and $p<0.05$, *Figure 1B and C*). Similarly, the *SLC25A1Δ* transcriptome was represented by 2433 transcripts whose expression was changed in *SLC25A1*-null cells, a fourfold difference compared to the 560 transcripts found in *SLC25A4Δ* cells (log2 fold of change of 1 and $p<0.001$, *Figure 1F and G*). Principal component analysis and 2D-tSNE analysis indicated that the whole measured proteome and transcriptome of *SLC25A1Δ* cells diverged strongly from wild-type cells, while *SLC25A4Δ* cells were more closely related to wild-type cells than *SLC25A1Δ* cells (*Figure 1D and H*). The same outcome was obtained by unsupervised clustering when considering proteins and transcripts significantly changed in at least one of these genotypes (*Figure 1E, I*), as *SLC25A4Δ* clustered with wild-type cells rather than *SLC25A1Δ* cells. Despite the abundance of altered gene products in these *SLC25A1Δ* and *SLC25A4Δ* datasets, there was limited overlap in proteomes and transcriptomes, with only 84 proteins and 385 mRNAs shared by both genotypes (*Figure 1J*). Notably, the congruency of the shared proteomes and transcriptomes reached only 0.9% of all the gene products whose expression was modified. This represents 27 proteins and transcripts similarly modified in *SLC25A1Δ* and *SLC25A4Δ* datasets (*Figure 1J*). Of these 27 common hits, one was annotated to mitochondria, FASTKD2, and five were annotated to the secreted human proteome, including soluble proteins such as apolipoprotein E (APOE) and cytokine receptor-like factor 1 (CRLF1; *Elson et al., 1998*; *Wernette-Hammond et al., 1989*; *Figure 1J*). APOE protein and transcript were among the most upregulated factors in both *SLC25A1Δ* and *SLC25A4Δ* cells (*Figure 1B–C and F–G*).

Close to 10% of all the *SLC25A1Δ* and *SLC25A4Δ* proteome hits were proteins annotated to mitochondria, with a discrete overlap of 10 mitochondrial proteins between these two mutant genotypes, mostly downregulated constituents of complex III of the electron transport chain (*Figure 1K*, UQCRB11, UQCRB, UQCRC2, and UQCRQ) as well as a factor required for the assembly of respiratory

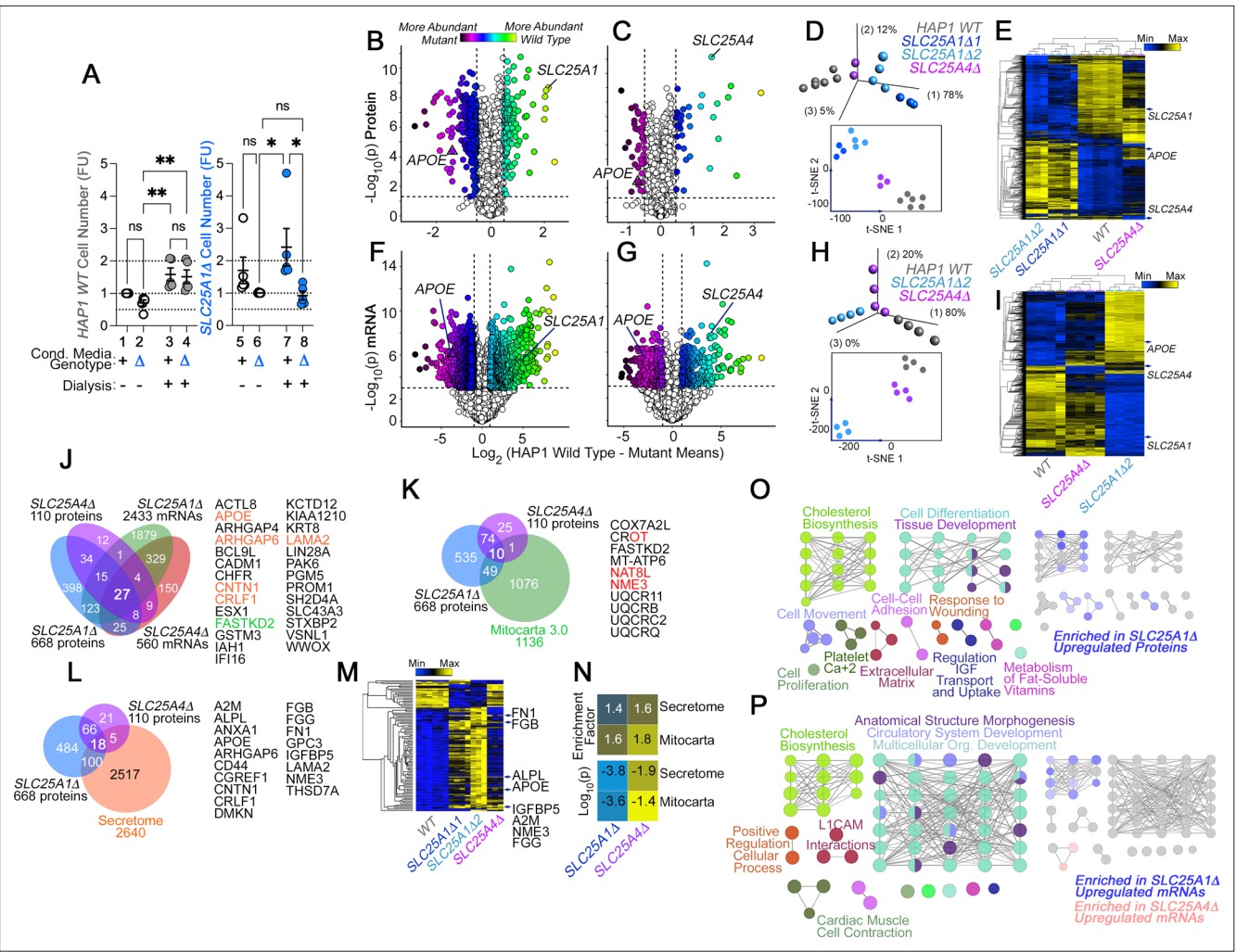

**Figure 1.** The secreted and mitochondrial proteomes are Modified by Inner Mitochondrial Membrane Transporter Mutants. (**A**) Cell number determinations of wild-type (columns 1–4) and *SLC25A1*-null HAP1 cells (*SLC25A1Δ*, columns 5–8) grown in the presence of conditioned media from each genotype. Conditioned media was applied to cells for 48 hr before (columns 1, 2, 5, and 6) or after dialysis (columns 3, 4, 7, and 8). Cell number was determined by Alamar blue cell viability assay. FU, Normalized Alamar Blue Fluorescence Units. Mean ± SEM, n=5, Two-Way ANOVA followed by Benjamini, Krieger and Yekutieli corrections. (**B–C**) Volcano plots of TMT proteomic data from wild-type HAP1 cells (n=3), *SLC25A1Δ* (B, n=3 for two independent CRISPR clones), and *SLC25A4Δ* mutants (C, n=3), depicted are log₁₀ p values and log₂ fold of change. (**D**) Principal component analysis and 2D-tSNE analyses of datasets in B-C. (**E**) Hierarchical clustering of all proteome hits where differential expression is significant with an α<0.001 in at least one mutant genotype. (**F–G**) Volcano plots of RNAseq data from wild-type HAP1 cells (n=4), *SLC25A1Δ* (B, n=4 for one independent CRISPR clone), and *SLC25A4Δ* mutants (C, n=4), depicted are log₁₀ p values and log₂ fold of change. (**H**) PCA and 2D-tSNE analyses of datasets in F-G. Subject grouping was determined by k-means clustering. (**I**) Hierarchical clustering of all RNAseq hits where differential expression is significant with an α=0.001 in at least one mutant genotype. (**J**) Venn diagram of protein and transcript hits shared by *SLC25A1Δ* and *SLC25A4Δ* mutants. Twenty-seven shared protein and RNA hits are annotated to either the human secretome (orange font) or annotated to Mitocarta 3.0 (green font). CROT was downregulated and upregulated in *SLC25A1Δ* and *SLC25A4Δ* mutants, respectively. (**K**) Venn diagram of protein hits in *SLC25A1Δ* and *SLC25A4Δ* mutants annotated in Mitocarta 3.0. (**L**) Venn diagram of protein hits in *SLC25A1Δ* and *SLC25A4Δ* mutants annotated in the Human Secretome (*Uhlén et al., 2019*). (**M**) Hierarchical clustering of all proteins annotated to the human secretome across genotypes. (**N**) Magnitude of compromise in secreted and mitochondrial proteomes in *SLC25A1Δ* and *SLC25A4Δ* mutants. p value was calculated with exact hypergeometric probability. (**O–P**) Gene ontology analysis of proteome (**O**) and transcriptome (**P**) in *SLC25A1Δ* and *SLC25A4Δ* mutants. Overlapping and mutant-specific ontologies are color-coded by percent of contribution >50% to an ontological category. Gray represents ontologies where all three mutants similarly contribute hits.

supercomplexes, COX7A2L (*Figure 1K*; *Lobo-Jarne et al., 2018*). The enrichment of secreted proteome annotated proteins was modest yet significant in *SLC25A1Δ* and *SLC25A4Δ* cells. Surprisingly, this degree of enrichment in components of the secretome was comparable to the enrichment of Mitocarta 3.0 annotated proteins in both mitochondrial mutants (*Figure 1L, M and N*). These

results show that mutations affecting two inner mitochondrial membrane transporters, SLC25A1 and SLC25A4, similarly affect the secreted and mitochondrial proteomes.

We analyzed *SLC25A1Δ* and *SLC25A4Δ* datasets for additional commonalities at the ontological level, using the ClueGO tool to annotate datasets based on genotype and whether a factor was up- or down-regulated. The annotated datasets were used to simultaneously query the KEGG, REACTOME and GO CC databases. The proteome and transcriptome of both mutants identified developmental ontologies as shared terms, irrespective of whether factors were up- or down-regulated (*Figure 1O–P* gray nodes and *Supplementary file 1*; tissue development GO:0009888, Bonferroni corrected *P*=1.9E-26 and 2.3E-11 for the transcriptome and proteome, respectively). However, there were ontologies that stood out by their genotype- and up-regulation-dependent specificity. For instance, the most prominent ontology annotated to up-regulated *SLC25A1Δ* proteome and transcriptome hits was steroid biosynthesis (KEGG:00100, Bonferroni corrected *P*=3.6E-8 and 1.9E-8 for the transcriptome and proteome, respectively; *Figure 1O–P*). However, the expression of genes annotated to sterol biosynthesis ontologies was not modified in *SLC25A4Δ* mutants, even though the proteome and transcriptome of *SLC25A4Δ* cells showed increased expression of APOE, a cholesterol transport lipoprotein. These findings suggest that expression of APOE and other hits common between these two mutant genotypes occurs independently from modifications in cholesterol synthesis pathways.

## APOE expression is uncoupled from changes in cholesterol synthesis pathways

We focused on APOE and the sterol biosynthesis pathways to validate our proteome and transcriptome data. We also determined whether cholesterol synthesis pathways correlated with increased APOE expression in *SLC25A1Δ* and *SLC25A4Δ* cells. Electrochemical MesoScale ELISA determinations of APOE with a human-specific antibody revealed increased APOE in cell lysates and conditioned media from *SLC25A1Δ* and *SLC25A4Δ* cells (*Figure 2A* and *Figure 2—figure supplement 1*; *Chikkaveeraiah et al., 2012*; *Gaiottino et al., 2013*). APOE protein expression and secretion into media were increased ~5–20 times in two CRISPR *SLC25A1Δ* clones and *SLC25A4Δ* cells (*Figure 2A* compare column 1 with 2–4 and *Figure 2—figure supplement 1* compare columns 1, 3, and 5 with 7, 9, and 11). APOE signal in complete media unexposed to cells was undetectable (*Figure 2A* compare media columns 1 with 5). APOE present in media and cells was sensitive to protein synthesis inhibition with cycloheximide (*Figure 2—figure supplement 1* compare columns 1–2 and 7–8) and to disruption of the secretory pathway with brefeldin A (*Figure 2—figure supplement 1* compare columns 3–4 and 9–10). Additionally, the lysosome protease inhibitor E-64 minimally affected APOE levels; thus, making unlikely the contribution of lysosomes to the genotype-dependent differences in APOE levels (*Figure 2—figure supplement 1* compare columns 5–6 and 11–12). We confirmed the increased levels of APOE in cells by immunoblot with a different APOE antibody. We used recombinant human APOE as a standard (*Figure 2B*). To exclude that an APOE expression increase was a haploid HAP1 cell peculiarity, we confirmed the increased levels of APOE in the diploid human neuroblastoma cell line SH-SY5Y where we CRISPRed out the SLC25A1 gene (*Figure 2C SLC25A1Δ/Δ*). Much like HAP1 cells, *SLC25A1Δ/Δ* cells increased secretion of APOE by ~fourfold, in both cells and conditioned media, compared with wild-type cells (*Figure 2C*, compare lanes 1 and 2). These results reveal a robust upregulation of both cellular and secreted APOE across mutant cell types.

If APOE expression depends on modifications in cholesterol pathways then the expression of genes annotated to cholesterol metabolism and cholesterol content should be similarly modified in *SLC25A1Δ* and *SLC25A1Δ/Δ* cells. We measured the transcript levels of APOE and genes involved in cholesterol uptake and synthesis, in *SLC25A1Δ*, *SLC25A4Δ*, and *SLC25A1Δ/Δ* cells. We focused on the LDL receptor (LDLR), as well as cholesterol synthesis enzymes, ACAT2, MSMO1 and HMGCR, the latter the rate-limiting enzyme of the cholesterol synthesis pathway (*Brown and Goldstein, 1980*; *Mazein et al., 2013*). We chose these genes as upregulated hits from the transcriptome of *SLC25A1Δ* cells. We used VAMP2 and RPS20 as housekeeping gene controls (*Figure 2D–F*). APOE mRNA increased ~threefold in all three mutant cells. In contrast, the expression of cholesterol synthesis pathway genes was increased in *SLC25A1Δ* (*Figure 2D*) but not in *SLC25A4Δ* and *SLC25A1Δ/Δ* cells (*Figure 2E and F*). The upregulation of cholesterol synthesis pathway genes resulted in a significant increase of cholesterol and all cholesterol-ester species content in *SLC25A1Δ* cells, as determined by mass spectrometry (*Figure 2G and J*). Triglyceride and other measured lipid families were similar in

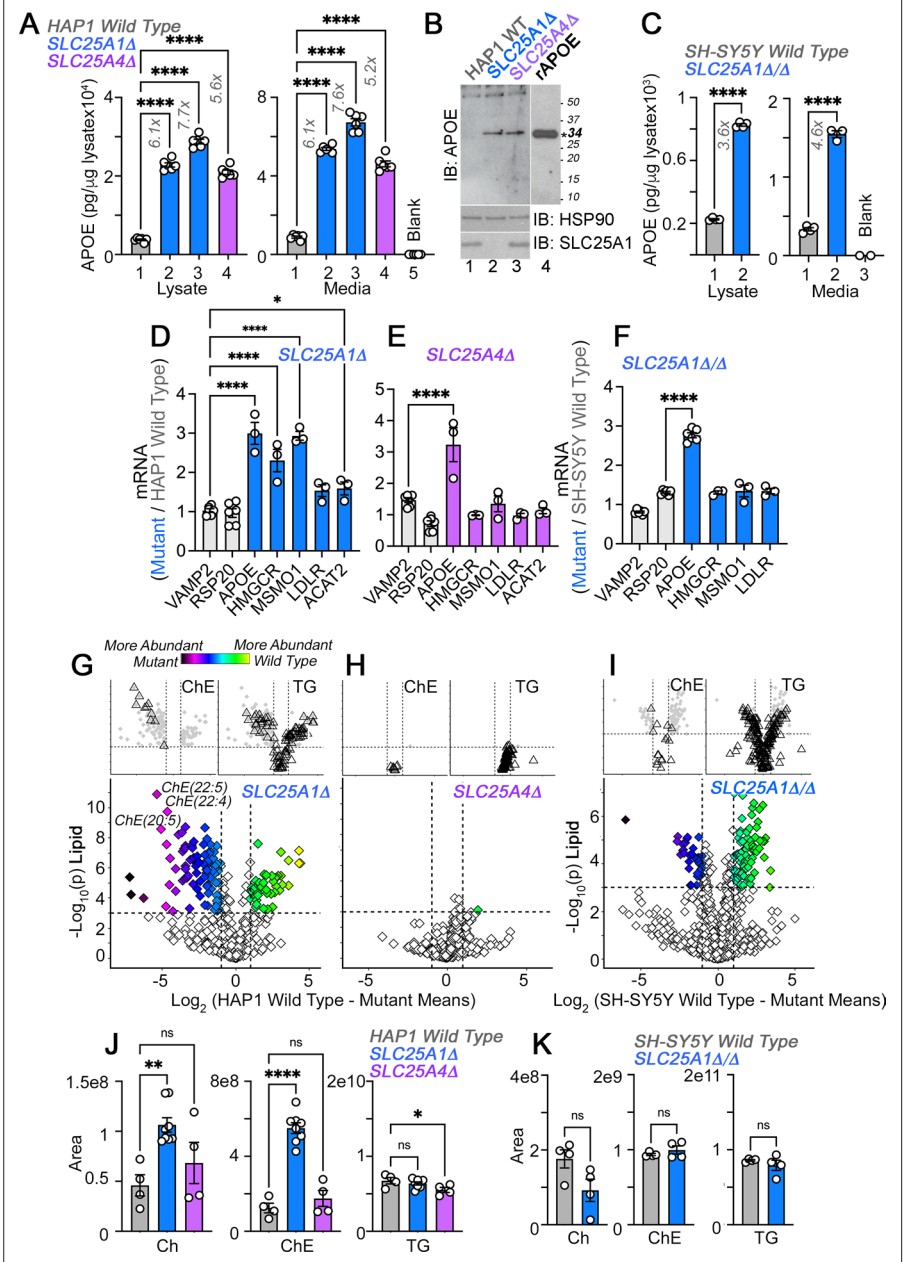

**Figure 2.** APOE transcripts and protein are upregulated independent from cholesterol levels in *SLC25A1* and *SLC25A4* mutants. (**A**) MesoScale electrochemiluminescence solid phase ELISA determinations of human APOE in wild-type (column 1), *SLC25A1Δ* (columns 2 and 3), and *SLC25A4Δ* (column 4) HAP1 mutant cell lysates and conditioned media. Two independent *SLC25A1Δ* clones were tested (columns 2–3). Column 5 depicts complete media not exposed to cells. n=4. (**B**) APOE immunoblot of cellular extracts from wild-type, *SLC25A1Δ*, and *SLC25A4Δ* HAP1 mutant cells. HSP90 was used as a loading control. Lane 4 presents recombinant human APOE (rAPOE). In bold is the predicted molecular weight of rAPOE. (**C**) MesoScale ELISA measurements of human APOE in wild-type and *SLC25A1Δ/Δ* SH-SY5Y mutant cell lysates and conditioned media. (**D–F**) qRT-PCR quantification of APOE, sterol metabolism annotated genes, and housekeeping controls (VAMP2 and RPS20) in wild-type and diverse mutant cell lines. (**D**) and (**E**) show transcript levels in *SLC25A1Δ* and *SLC25A4Δ* HAP1 mutant cells, respectively. (**F**) depicts transcript levels in *SLC25A1Δ/Δ* SH-SY5Y mutant cells. All data are expressed as transcript ratio between mutant and wild-type. n=3 for D-F. (**G–I**) Volcano plots of positive mode untargeted lipidomics performed in *SLC25A1Δ*, *SLC25A4Δ*, and *SLC25A1Δ/Δ* mutant HAP1 and SH-SY5Y cells and their controls. Upper inserts present the distribution of cholesterol ester and triglyceride species marked by triangles. Depicted are $\log_{10}$ p values and $\log_2$ fold of change. n=4 per clone for the two *SLC25A1Δ* clones, n=4 for *SLC25A4Δ*, and n=4 for

*Figure 2 continued on next page*

*Figure 2 continued*

*SLC25A1Δ/Δ*. (**J**). Total cellular levels of free cholesterol (Ch), cholesterol ester (ChE), and triglycerides (TG) in wild-type, *SLC25A1Δ*, and *SLC25A4Δ* HAP1 cells. (**K**). Total cellular levels of free cholesterol (Ch), cholesterol ester (ChE), and triglycerides (TG) in wild-type and *SLC25A1Δ/Δ* SH-SY5Y cells. Average ± SEM, One-way ANOVA followed by Bonferroni or Holm-Šydák's (**D–F**) multiple corrections, or unpaired t-test (**K**). See available source data for (**B**).

The online version of this article includes the following source data and figure supplement(s) for figure 2:

**Source data 1.** Original blots.

**Source data 2.** Original blots.

**Figure supplement 1.** Effects of diverse mon-mitochondrial inhibitors on APOE expression and secretion.

wild-type and *SLC25A1Δ* cells (*Figure 2G and J*). In contrast, cholesterol and cholesterol-ester species were not modified in *SLC25A4Δ* (*Figure 2H and J*) and *SLC25A1Δ/Δ* cells (*Figure 2I and K*), even though the expression of APOE was upregulated to the same extent in all these mutant cells. These results make unlikely the hypothesis that increased APOE expression is coupled to an upregulation of cholesterol synthesis pathways.

## Perturbation of the electron transport chain complexes I and III increases APOE expression

We turned our attention to common defects in *SLC25A1Δ* and *SLC25A4Δ* HAP1 cells that could explain the increased expression of APOE. A shared phenotype in both mutants was a drop in the levels of complex III subunits and the supercomplex III-IV assembly factor COX7A2L (*Figure 1K*). We hypothesized that defects in the integrity of the electron transport chain could mediate the increased levels of APOE. To test this hypothesis, we first determined the effects of SLC25A1 and *SLC25A4* mutations on the organization of the electron transport chain by blue native electrophoresis. Second, we mutagenized assembly factors and subunits of complexes I (NDUFS3 and NDUFAF7), III (COX7A2L and HIGD1A), and IV of the electron transport chain (COX7A2L, HIGD1A, COX17, 18, 19, and 20) and measured APOE levels (*Lobo-Jarne et al., 2018*; *Nývltová et al., 2022*; *Timón-Gómez et al., 2020a*; *Zurita Rendón et al., 2014*). We determined the robustness of the increased APOE phenotype by studying mutants in three human cell lines (HAP1, SH-SY5Y, and HEK293), which differ in several properties, including their genetic background, rate of growth, and tissue of origin.

We measured the expression of electron transport chain subunits by proteomics in HAP1 cells. We found that even though complex III was affected in both *SLC25A1Δ* and *SLC25A4Δ* cells (*Figures 1K and 3A* and *Figure 3—figure supplement 1A*), the most pronounced defect was in the expression of complex I subunits in *SLC25A1Δ* cells (*Figure 3A*). We scrutinized the integrity of respiratory chain complexes in *SLC25A1Δ* cells by SDS-PAGE, blue native electrophoresis, and bidimensional gel electrophoresis (*Figure 3B–D*). We performed immunoblot analysis of SDS-PAGE resolved respiratory complex subunits with antibodies against subunits that undergo degradation in misassembled respiratory complexes (*Civiletto et al., 2018*; *Ghazal et al., 2021*). These experiments revealed that the most degraded subunits were those from complexes I and III (*Figure 3B*). We then analyzed respiratory complexes by blue native electrophoresis and found decreased expression of high molecular weight complexes containing NDUFS3, UQCRC2, and COX4, which correspond to subunits of the respiratory complexes I, III, and IV, respectively (*Figure 3C*). Two-dimensional gel electrophoresis showed that high molecular weight respiratory complexes containing subunits of complexes I and III were diminished in *SLC25A1Δ* cells (*Figure 3D*). Similarly, *SLC25A4Δ* HAP1 cells had reduced levels of complex III subunits (*Figure 3—figure supplement 1A*), thus affecting the migration of complex III-containing supercomplexes in blue native electrophoresis (*Figure 3—figure supplement 1B*). The effect of the *SLC25A4* mutation on APOE expression was not due to a defect in the SLC25A4 transport activity, as the SLC25A4 inhibitor bongkrekic acid did not increase the levels of APOE (*Figure 3—figure supplement 1C*, compare columns 1 with 2 and 3), even at concentrations that inhibit mitochondrial respiration to the same extent as the *SLC25A4* mutation (*Figure 3—figure supplement 1D*; *Gutiérrez-Aguilar and Baines, 2013*).

The loss of integrity in complexes I and III in *SLC25A1Δ* and *SLC25A4Δ* HAP1 cells suggest that respiratory chain defects could be responsible for the increased expression of APOE. We tested this hypothesis by targeting complexes I and III using genetic and pharmacological approaches (*Figure 3*

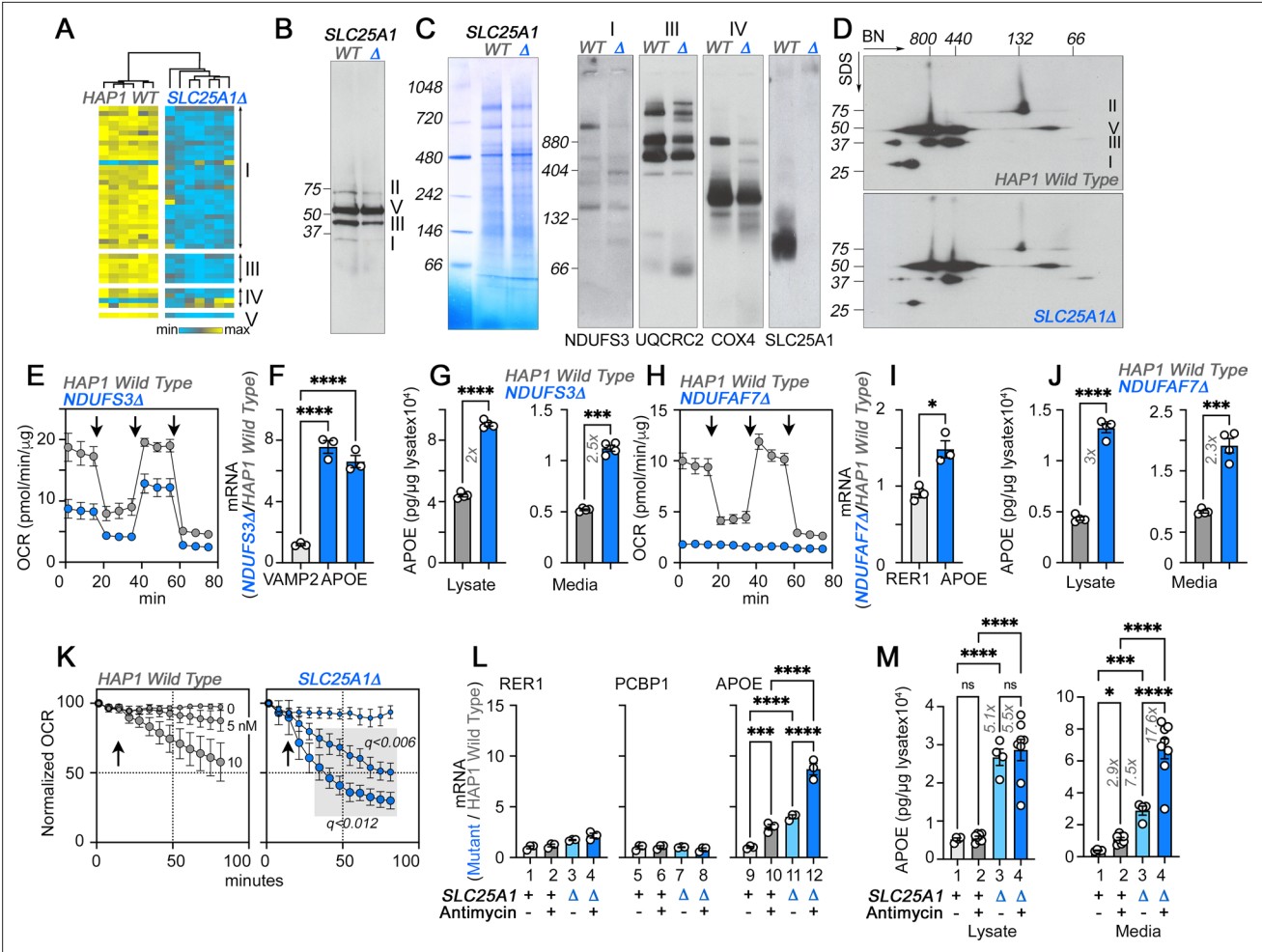

**Figure 3.** The integrity of respiratory chain complex I is required to control APOE expression. (**A**) Expression of respiratory complex subunits in wild-type and *SLC25A1Δ* HAP1 cells quantified by TMT mass spectrometry. Kendal Tau hierarchical clustering analysis. (**B**) Immunoblots with OxPhos antibody mix in mitochondrial fractions from wild-type and *SLC25A1Δ* cells. (**C**) Blue native electrophoresis of mitochondrial fractions from wild-type and *SLC25A1Δ* cells. Shown are Coomassie stained native gel and immunoblots probed with antibodies against complex, I, III, IV, and SLC25A1. (**D**) Blue native electrophoresis followed by SDS-PAGE then immunoblot with antibodies against complex, I, II, III, and V in mitochondrial fractions from wild-type and *SLC25A1Δ* cells. (**E–G**). Seahorse stress test, APOE qRT-PCR, and APOE MesoScale ELISA analysis respectively in wild-type and *NDUFS3Δ* HAP1 cells. In (**F**), APOE was measured with two primer sets. (**H–J**) Seahorse stress test, APOE qRT-PCR, and APOE MesoScale ELISA analysis respectively in wild-type and *NDUFAF7Δ* HAP1 cells. VAMP2 or RER1 transcripts were used as controls in **F** and **J**. All qRT-PCR data are expressed as ratio between mutant and wild-type. (**E** to **M**) average ± SEM. One-Way ANOVA followed by Šydák's multiple correction (**F**), or unpaired t-test with Welch's correction (**G, I, and J**). Arrows in **E** (n=4) and **H** (n=3) show sequential addition of oligomycin, FCCP, and rotenone-antimycin during the Seahorse stress test. (**K**) *SLC25A1Δ* cells are more sensitive to antimycin than wild-type HAP1 cells. Wild type and *SLC25A1Δ* cells were exposed to vehicle or increasing concentrations of antimycin. Basal cellular respiration was measured for 90 min after additions (arrow) using Seahorse. Data are presented normalized to basal respiration in the absence of drug. Average ± SEM, n=3, Gray square shows significant differences between wild-type and *SLC25A1Δ* drug-treated cells as determined by multiple unpaired t-tests followed by corrections with the Benjamini-Krieger-Yekuiteli method (FDR = 5%). (**L–M**) APOE qRT-PCR and APOE MesoScale ELISA in wild-type and *SLC25A1Δ* HAP1 cells, respectively, treated with vehicle or antimycin. Twenty nM antimycin was used in qPCR experiments. Twenty to 80 nM was used in MesoScale ELISA experiments. RER1 (columns 1–4) and PCBP1 transcripts (columns 5–8) were used as housekeeping controls. All qRT-PCR data are expressed as ratio between mutant and wild-type. Average ± SEM, One-way ANOVA followed by Benjamini-Krieger-Yekuiteli multiple comparison corrections (FDR = 5%). See available source data for (**B and C**).

The online version of this article includes the following source data and figure supplement(s) for figure 3:

**Source data 1.** Original gels and blots.

**Source data 2.** Original gels and blots.

**Figure supplement 1.** SLC25A4 null HAP1 cells disrupt complex III and increase expression of APOE.

**Figure supplement 1—source data 1.** Original gels and blots.

*Figure 3 continued on next page*

*Figure 3 continued*

**Figure supplement 1—source data 2.** Original gels and blots.

**Figure supplement 2.** Effects of inhibitors of the electron transport chain on APOE levels.

**Figure supplement 3.** Interactions between SLC25A1 and NDUFS3.

**Figure supplement 3—source data 1.** Original blots.

**Figure supplement 3—source data 2.** Original gels and blots.

and *Figure 3—figure supplement 2*). We genetically perturbed complex I assembly and function by knocking-out either NDUFS3 or NDUFAF7 or by using the complex I inhibitor piericidin A (*Figure 3—figure supplement 2*; *Bridges et al., 2020*). NDUFS3 encodes NADH dehydrogenase [ubiquinone] iron-sulfur protein 3, a non-catalytic subunit of complex I necessary for complex I assembly and activity (*Bénit et al., 2004*; *D'Angelo et al., 2021*). NDUFAF7 encodes NADH:ubiquinone oxidoreductase complex assembly factor 7, a methylase necessary for the early stages of complex I assembly (*Zurita Rendón et al., 2014*). We targeted NDUFS3 and NDUFAF7 because these genes localize to genetic loci associated with increased risk of Alzheimer's disease (*de Rojas et al., 2021*; *Kunkle et al., 2019*). Moreover, SLC25A1 and NDUFS3 share 91% of their proximity interactions (*Antonicka et al., 2020*), an observation we corroborated by co-immunoprecipitation of SLC25A1 and NDUFS3 (*Figure 3—figure supplement 3*). NDUFS3 and NDUFAF7 CRISPR mutants compromised mitochondrial respiration, as shown by Seahorse oximetry (*Figure 3E and H*). Both mutants increased the expression of APOE mRNA, as compared to reference housekeeping transcripts, (*Figure 3F I*, RER1 and VAMP2) as well as APOE protein in cells and in conditioned media (*Figure 3G and J*). These results demonstrate that the integrity and function of complex I regulate APOE expression and secretion.

We inhibited the function of complex III with the specific inhibitor antimycin I (*von and Link, 1986*). We inquired whether inhibition of complex III by antimycin would upregulate the expression of APOE in wild-type and in *SLC25A1Δ* cells. Since complex III levels were partially reduced in *SLC25A1Δ* cells (*Figure 3A, B and C*), we reasoned that inhibition of residual complex III activity in *SLC25A1Δ* cells would reveal additive respiratory chain mechanisms affecting APOE expression. We measured respiration in wild-type and *SLC25A1Δ* cells in the absence and presence of increasing concentrations of antimycin (*Figure 3K*). *SLC25A1Δ* cells were more sensitive to low doses of antimycin (*Figure 3K*), a phenotype predicted for cells with reduced content of complex III. Antimycin addition increased APOE mRNA 3-fold in wild-type cells as compared to vehicle (*Figure 3L* compare columns 9–10), whereas the expression of two housekeeping mRNAs, RER1 and PCBP1, was minimally affected by complex III inhibition (*Figure 3L* compare columns 1–2 and 5–6). *SLC25A1Δ* cells doubled their APOE mRNA when treated with antimycin as compared to vehicle-treated *SLC25A1Δ* cells (*Figure 3L* compare columns 11–12). Antimycin also increased APOE secretion in both wild-type and *SLC25A1Δ* cells by ~2–3 fold (*Figure 3M*). No such increase was detectable in cell lysates in both genotypes (*Figure 3M*). These data show that the effect of antimycin on APOE mRNA expression and protein secretion was additive with the *SLC25A1Δ* upregulation phenotype. Taken together, these results demonstrate that the expression of APOE is sensitive to complex III inhibition.

We tested the robustness of the increased APOE phenotype in HEK293 cells. We mutagenized *SLC25A4* alone or in conjunction with its two homologs, *SLC25A5* and 6. *SLC25A4* is the main ADP/ATP carrier expressed in HAP1 cells (*Figure 4—figure supplement 1*). In contrast, *SLC25A5* and 6 are the main species expressed by HEK293 cells (*Lu et al., 2017*). TMT proteomic analysis of whole cell extracts from *SLC25A4Δ/Δ* HEK293 cells revealed minimal changes in protein levels (*Figure 4A*). In contrast, we found 223 differentially expressed proteins in *SLC25A4,5,6Δ/Δ* triple knock-out cells (*Figure 4A*). Of these 223 proteins, 32 were annotated to the secretome, including APOE (*Figure 4B and C*). We confirmed increased APOE levels by MesoScale ELISA in *SLC25A4,5,6Δ/Δ* triple knock-out cells, which increased APOE ~twofold, in both cell extracts and conditioned media, as compared with wild-type and *SLC25A4Δ/Δ* HEK293 cells (*Figure 4D*). Among the 223 differentially expressed proteins, we also identified 42 proteins annotated to Mitocarta 3.0 (*Figure 4B*). These included increased levels of 4 of the 10 subunits in complex III (*Figure 4B and E*). Formation of complex III supercomplexes was selectively compromised in *SLC25A4,5,6Δ/Δ* triple knock-out cells as determined by blue native electrophoresis (*Figure 4F*). These mitochondrial proteome modifications and defective supercomplexes formation reduced mitochondrial respiration by fivefold in triple knock-out

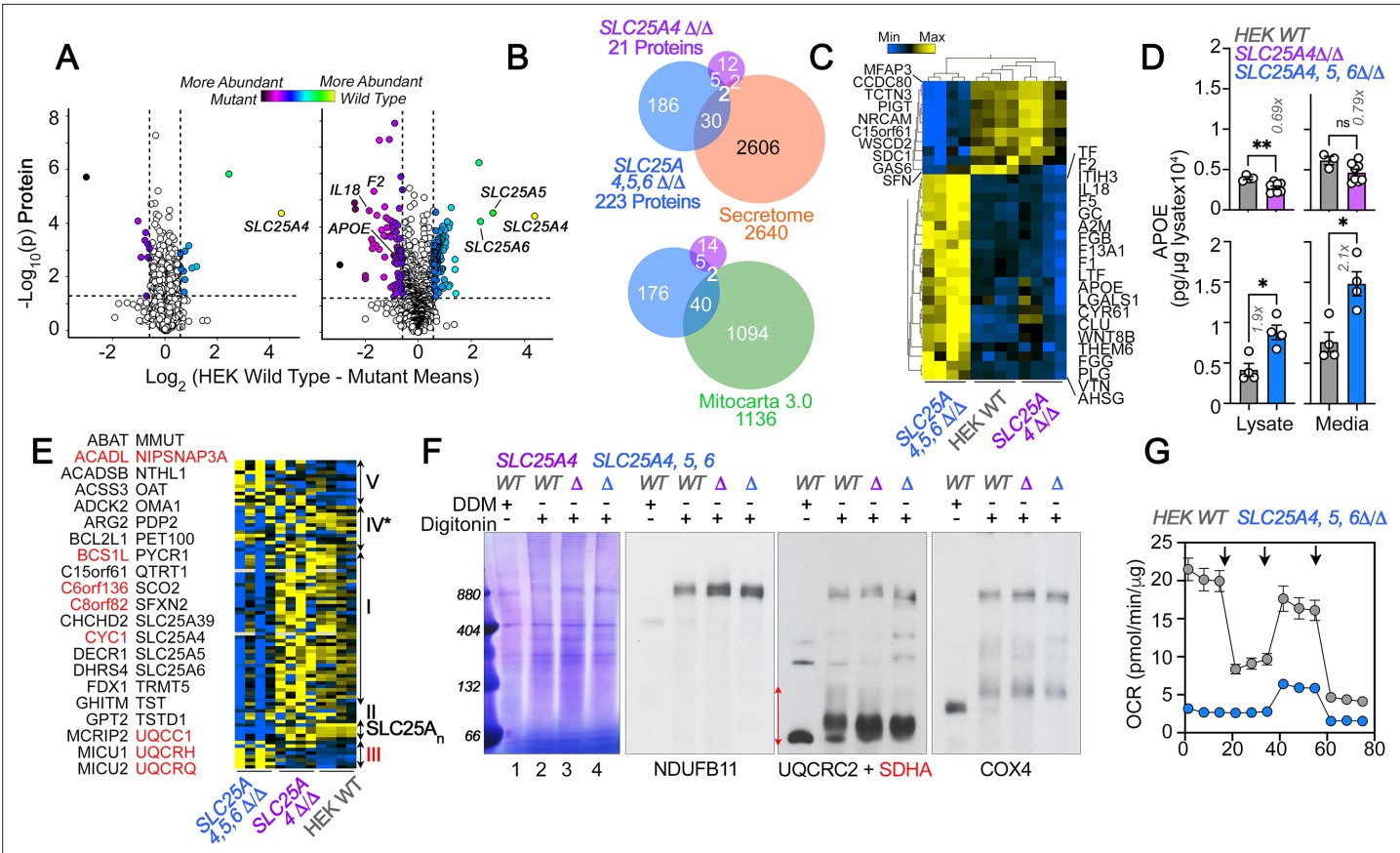

**Figure 4.** Robustness and redundancy of adenine nucleotide translocators regulating APOE expression. (**A**) Volcano plots of TMT proteomic data from wild-type HEK293 cells (n=4), *SLC25A4Δ/Δ* and triple knock-out *SLC25A4,5,6Δ/Δ* (B, n=4), depicted are $\log_{10}$ p values and $\log_2$ fold of change. (**B**). Venn diagram of protein hits in *SLC25A4Δ/Δ* and triple knock-out *SLC25A4,5,6Δ/Δ* mutants annotated in Mitocarta 3.0 or the Human Secretome (***Uhlén et al., 2019***). (**C**) Hierarchical clustering of all proteins annotated to the human secretome across genotypes. (**D**) MesoScale ELISA determinations of human APOE in wild-type and mutant HEK293 cells. Shown are APOE content in lysates and conditioned media. Mann-Whitney test. (**E**) Mitocarta 3.0 annotated hits in the triple knock-out *SLC25A4,5,6Δ/Δ* proteome, see panel **B**. Red font indicates increased levels in mutant. Hierarchical clustering of all proteins annotated to electron transport chain subunits and SLC25A transporters across genotypes. (**F**) Blue native electrophoresis followed by immunoblotting with antibodies against complex, I, II, III, and IV in mitochondrial fractions from wild-type and *SLC25A4Δ/Δ* and triple knock-out *SLC25A4,5,6Δ/Δ* cells. n-dodecyl-β-d-maltoside (DDM) was used to disrupt supercomplexes. Red arrow and font denote region blotted with SDHA antibodies. (**G**) Seahorse stress test wild-type and triple knock-out *SLC25A4,5,6Δ/Δ* cells as in ***Figure 3***. N=3. See available source data for (**F**).

The online version of this article includes the following source data and figure supplement(s) for figure 4:

**Source data 1.** Original gels and blots.

**Source data 2.** Original gels and blots.

**Figure supplement 1.** SLC25A4 null HAP1 cells disrupt complex III and increase expression of APOE.

cells (***Figure 4G***). We conclude that increased APOE expression is robustly observed in diverse cell types where the integrity of complex III of the electron transport chain is compromised.

## Direct and indirect mechanisms affecting complex IV biogenesis increases APOE expression

In order to address if the integrity of respiratory complexes, other than complexes I and III, could modulate APOE expression, we focused on complex IV in the electron transport chain. The assembly of this complex requires a pathway that begins at the plasma membrane to deliver copper, which is necessary for complex IV biogenesis (***Figure 5A***; ***Cobine et al., 2021***; ***Garza et al., 2022***). At mito-chondria, complex IV requires assembly factors to generate the complex itself (HIGD1A, COX18, and COX20), factors to generate complex IV-containing supercomplexes (COX7A2L and HIGD1A), a copper transporter present in the inner mitochondrial membrane (SLC25A3), and metallochaperones

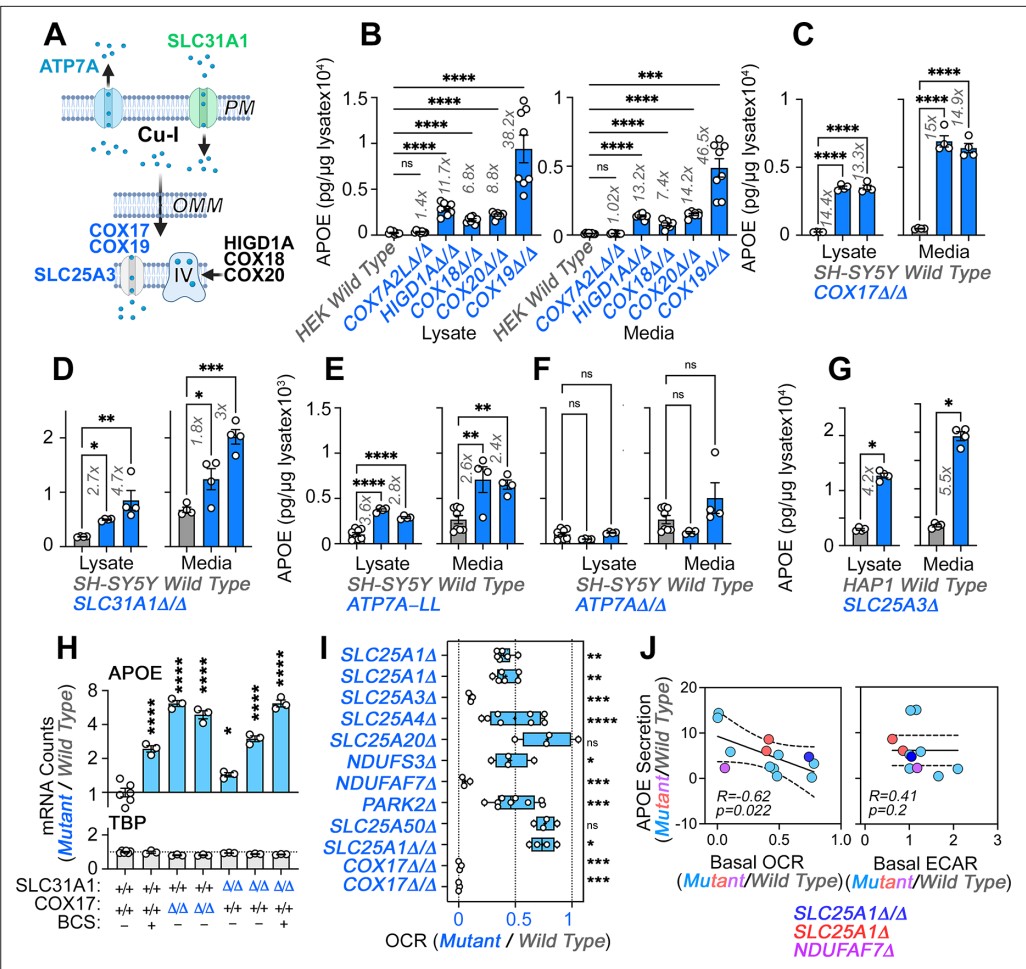

**Figure 5.** Direct and indirect disruption of complex IV increases APOE expression. (**A**) Direct (COX17-20, HIGD1A) and indirect (SLC25A3, SLC31A1, ATP7A) mechanisms required for complex IV assembly. (**B**) MesoScale ELISA determinations of human APOE in wild-type and HEK293 cell clones null for the genes indicated in blue font. Shown are APOE content in lysates and conditioned media. (**C**) MesoScale ELISA determinations of human APOE in wild-type and two independent *COX17Δ/Δ* mutant SH-SY5Y cell clones. (**D**) MesoScale ELISA determinations of human APOE in wild-type and two independent *SLC31A1Δ/Δ* mutant SH-SY5Y cell clones were studied. (**E**) MesoScale ELISA determinations of human APOE in wild-type and two independent *ATP7AΔ/Δ* mutant SH-SY5Y cell clones transfected with the endocytosis-deficient ATP7A-LL construct. (**F**) MesoScale ELISA determinations of human APOE in wild-type and two independent *ATP7AΔ/Δ* mutant SH-SY5Y cell clones. (**G**) MesoScale ELISA determinations of human APOE in wild-type and *SLC25A3Δ* mutant HAP1 cells were studied. N=8 for **B** and n=4 for **C-G**. (**H**) NanoString mRNA quantification of human APOE and TBP transcripts in wild-type, and two independent mutant clones of either *COX17Δ/Δ* or *SLC31A1Δ/Δ* mutant SH-SY5Y cells. Wild type and *SLC31A1Δ/Δ* cells were treated with vehicle or 200 micromolar of the copper chelator bathocuproinedisulfonic acid (BCS). TBP was used as a housekeeping control transcript. n=3. (**I**) Seahorse basal cellular respiration across different genotypes normalized to the corresponding wild-type cell. (**J**) Correlation between APOE in conditioned media with either basal cellular respiration (OCR) or the extracellular acidification rate determined by Seahorse (ECAR, n=3–9). Simple linear regression fit and 95% confidence interval is shown. All data are presented as average ± SEM. For **B** to **F**, and **H** One-way ANOVA followed by Benjamini-Krieger-Yekuiteli multiple comparison corrections (FDR = 5%). (**G**) unpaired t-test with Welch's correction.

The online version of this article includes the following source data and figure supplement(s) for figure 5:

**Figure supplement 1.** Total cellular copper in wild type and diverse mutant cells.

**Figure supplement 2.** Upregulation of APOE expression and secretion in *SLC25A20* mutant cells.

**Figure supplement 2—source data 1.** Original blots.

**Figure supplement 2—source data 2.** Original blots.

*Figure 5 continued on next page*

**Figure supplement 3.** Extracellular acidification rate (ECAR) in wild-type and diverse mutant cells.

**Figure supplement 4.** Citrate effects on APOE expression in HAP1 cells.

**Figure supplement 5.** AMPK, mitochondrial stress, and redox responses in wild type and *SLC25A1Δ* HAP1 cells.

that deliver copper I ions to complex IV (COX17 and COX19). In turn, the mitochondrial copper metallochaperone COX17 receives its copper from the plasma membrane copper uptake transporter SLC31A1 (*Figure 5A*; *Cobine et al., 2021*; *Garza et al., 2022*). We knocked out genes belonging to this pathway and assessed the expression of APOE in three cell models.

We mutated the mitochondrial factors COX7A2L, HIGD1A, COX18, COX19, or COX20 in HEK293 cells and measured APOE levels by MesoScale ELISA (*Figure 5B*). We first focused on COX7A2L and HIGD1A, as these two factors are required for the assembly of complex III-IV supercomplexes, yet they differ in that COX7A2L does not affect the biogenesis of individual complexes, thus keeping mitochondrial respiration intact (*Lobo-Jarne et al., 2018*). In contrast, HIGD1A affects both supercomplex formation and the biogenesis of individual complexes III and IV (*Timón-Gómez et al., 2020a*). Mutagenesis of *COX7A2L* did not affect APOE expression but mutagenesis of *HIGD1A* increased APOE levels ~12-fold, both in cell lysates and conditioned media (*Figure 5B*). We confirmed these findings with mutants in the complex IV assembly factors COX18 and 20 (*Bourens and Barrientos, 2017*; *Nývltová et al., 2022*). *COX18* and *20* gene defects increased APOE expression to the same extent as *HIGD1A* (*Figure 5B*).

Complex IV requires copper for its biogenesis and function. Thus, our model predicts that genetic disruption of proteins in the pathway that leads to delivery of copper to complex IV should increase APOE expression (*Figure 5A*). Among the factors that help deliver copper into complex IV, we studied COX17 and COX19 (*Banci et al., 2008*; *Cobine et al., 2021*; *Garza et al., 2022*; *Leary et al., 2013*; *Nývltová et al., 2022*; *Oswald et al., 2009*). Mutagenesis of COX19 in HEK293 cells increased the expression of APOE protein in both cells and conditioned media by ~40-fold (*Figure 5B*). We confirmed this finding in two SH-SY5Y mutants of the metallochaperone COX17 (*Figure 5C*, *COX17Δ/Δ*). We decreased cytoplasmic copper availability by either eliminating copper uptake via mutagenesis of the SLC31A1 transporter (*Figure 5D*, *SLC31A1Δ/Δ* and *Figure 5—figure supplement 1*) or by expressing in ATP7A-null SH-SY5Y cells (*ATP7AΔ/Δ* and *Figure 5—figure supplement 1*) an ATP7A mutant that constitutively extrudes copper from cells due to mutagenesis of its endocytosis sorting signal (*Figure 5E* and *Figure 5—figure supplement 1*, *ATP7A-LL*) (*Petris et al., 1998*; *Zhu et al., 2016*). Both forms of cytoplasmic copper reduction increased APOE expression and secretion in two independent mutant or clonal isolates (*Figure 5D–E*). Elimination of ATP7A alone did not increase APOE expression (*Figure 5F*, *ATP7AΔ/Δ*). Mutagenesis of *SLC25A3*, also increased the expression and secretion of APOE protein in HAP1 cells (*Figure 5G*). We confirmed these results by measuring APOE mRNA levels with NanoString technology in two independent mutants of *COX17* and *SLC31A1*, as well as in wild-type cells incubated with a cell impermeant copper chelator, bathocuproinedisulfonic acid (*Figure 5H*, BCS and *Figure 5—figure supplement 1*). These pharmacological and genetic approaches to reduction of cytoplasmic copper robustly increased APOE mRNA as compared to wild-type cells (*Figure 5H*), while levels of a housekeeping gene, TBP, were unchanged. These results demonstrate that impairing complex IV biogenesis, directly or indirectly, increases the expression of APOE transcript and protein in diverse cell lines. Along with our experiments targeting complexes I and III, these findings support the concept that diverse mechanisms converging on respiratory chain function and assembly can induce the expression and secretion of APOE.

To determine whether there was a relationship between the degree of respiratory chain compromise, glycolysis, and the extent of APOE secretion increase, we correlated the normalized secretion of APOE in diverse mitochondrial mutants to their normalized basal oxygen consumption rate, and the rate of extracellular acidification as a proxy for glycolysis (*Figure 5I–J*). APOE negatively and significantly correlated with basal oxygen consumption (–0.62, *Figure 5J*, r=−0.62, p=0.022). However, we identified an exception in mutants of the carnitine transporter SLC25A20 that increased APOE levels despite normal respiration (*Figure 5—figure supplement 2*). The correlation of APOE expression with

the rate of extracellular acidification did not differ from zero (*Figure 5J*, p=0.2 and *Figure 5—figure supplement 3*). These findings support the idea that respiratory chain integrity and activity rather than glycolytic adaptations in mutants affecting the electron transport chain modulate APOE expression.

We explored alternative mechanisms from electron transport chain assembly and activity required for increased APOE expression and secretion. We ruled out the possibility that either decreased cytoplasmic citrate levels, by mutagenesis of ACLY (*Figure 5—figure supplement 4*) or bioenergetic stress are responsible for the APOE increase in *SLC25A1Δ* cells, using activation of the AMPK pathway as a read-out (*Figure 5—figure supplement 5*). We ruled out accumulation of reactive oxygen species as a causal mechanism for increased APOE, as treatment with the antioxidant N-acetyl cysteine did not affect APOE mRNA levels in wild-type or *SLC25A1Δ* cells (*Figure 5—figure supplement 5*). We also assessed whether activation of the mitochondrial stress response pathway caused APOE elevation (*Figure 5—figure supplement 5*). We found that activation of the mitochondrial stress response with doxycycline (*Quirós et al., 2017*) modestly increased the secretion of APOE in *SLC25A1Δ* cells (*Figure 5—figure supplement 5* panel D columns 7–8) but not in wild-type cells (*Figure 5—figure supplement 5* panel D columns 5–6), even though wild-type cells robustly activated the mitochondrial stress response after doxycycline incubation (*Figure 5—figure supplement 5* panel E).

## Mitochondrial dysfunction induces APOE expression and inflammatory responses in immortalized cells and human astrocytes

The APOE4 allele is the strongest genetic risk factor for sporadic Alzheimer's disease and APOE-associated neuroinflammation is thought to play a prominent role in disease pathogenesis (*Krasemann et al., 2017*; *Parhizkar and Holtzman, 2022*; *Tzioras et al., 2019*; *Zalocusky et al., 2021*). We found that disruption of complex I subunit NDUFS3 and complex I assembly factor NDUFAF7 increases APOE expression (*Figure 3*). The *NDUFS3* and *NDUFAF7* genes are encoded in loci that increase Alzheimer's disease risk (*de Rojas et al., 2021*; *Kunkle et al., 2019*). Moreover, the complex IV subunit COX7C is an additional electron transport gene encoded in a novel Alzheimer's risk locus (*Bellenguez et al., 2022*). These genetic associations between APOE and the respiratory chain with Alzheimer's disease risk prompted us to ask the following questions: (1) Do human brain cells modify the expression of APOE after disruption of the respiratory chain? (2) Is mitochondrially induced APOE expression an isolated event, or does it co-occur in the context of inflammatory response?

To address the first question, we treated wild-type human iPSC-derived neurons and astrocytes with antimycin for 48 hr. Cells were treated with antimycin concentrations that inhibit mitochondrial respiration in neurons and astrocytes but do not affect cell viability (unpublished data). Astrocytes increased APOE protein expression twofold after antimycin treatment (*Figure 6A*). These astrocytic APOE increases were detected both in cell lysates and conditioned media (*Figure 6A*). In contrast, neuronal cells failed to increase APOE expression after exposure to antimycin (*Figure 6A*). We did not detect APOE in neuronal conditioned media (not shown). These findings show that APOE expression and secretion are modulated by intoxication of the electron transport chain in iPSC-derived human astrocytes but not neurons. To determine if APOE expression was coordinated with an inflammatory response, we analyzed the expression of 770 genes using a NanoString mRNA quantification panel. This neuroinflammatory panel reports the activity of 23 neuroinflammation pathways and processes across five brain cell types and 14 cell types of the peripheral immune system. We chose this approach since it is validated in Alzheimer's models, highly sensitive, and measures mRNAs without cDNA amplification (*Das et al., 2021*; *Ramesha et al., 2021*). We performed NanoString quantification in wild-type, *SLC25A1Δ*, and *SLC25A4Δ* HAP1 cells, as well as in iPSC-derived astrocytes treated for 48 hr with sublethal doses of antimycin (*Figure 6B*). We identified upregulation of 3–10% of the genes in the panel across these three cellular models (*Figure 6B*). Among these genes, APOE mRNA was upregulated sixfold in mutant HAP1 cells and 1.5 times in antimycin-treated astrocytes as compared to a housekeeping control, TBP (*Figure 6C*). The genes upregulated in antimycin-treated astrocytes were significantly enriched in genes annotated to the secretome, as compared to the content of secretome-annotated genes built into the panel (*Figure 6D*, compare enrichment factors of 2.8 and 1.6, respectively). Like in *SLC25A1Δ* and *SLC25A4Δ* HAP1 cells, this enrichment of altered secretory transcripts induced by mitochondrial damage was similar to changes in mitochondrially annotated transcripts.

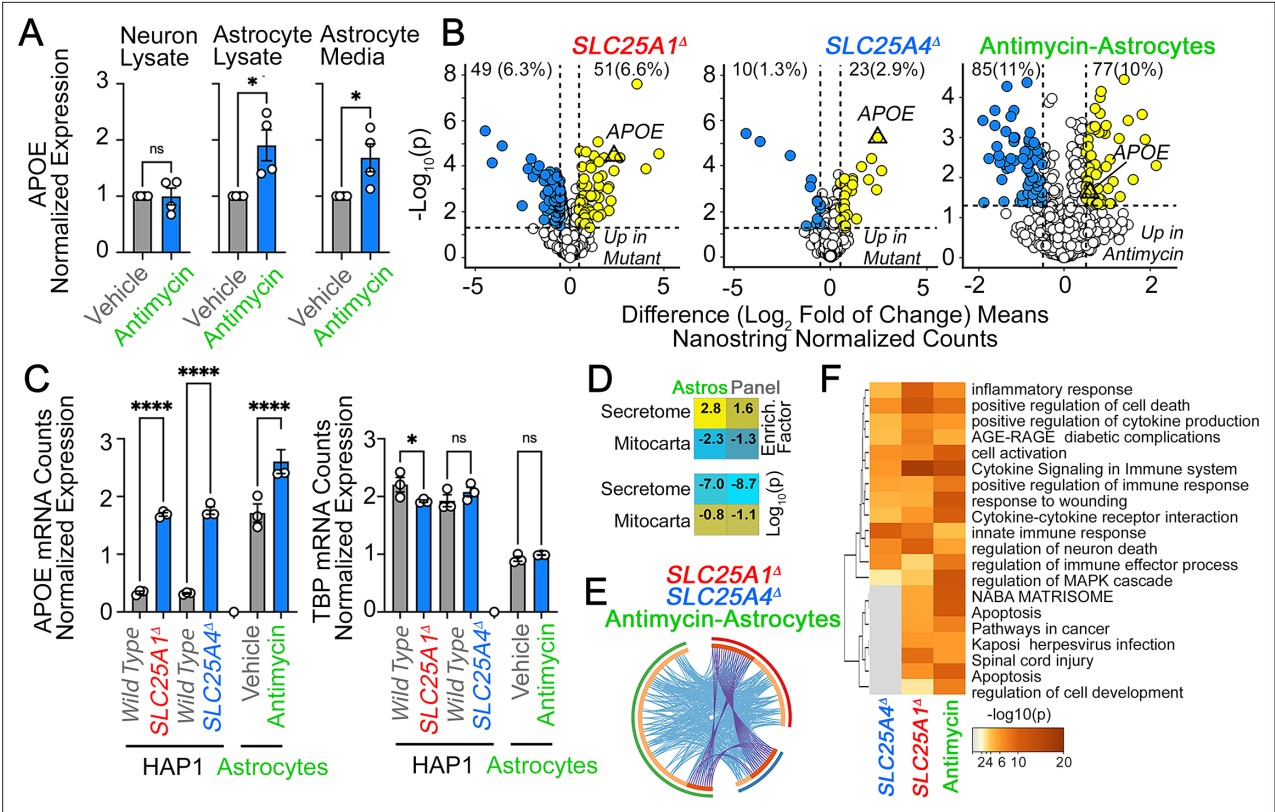

**Figure 6.** APOE expression is increased in human astrocytes after complex III inhibition. (**A**) MesoScale ELISA determinations of human APOE in wild-type iPSC-derived human neurons and astrocytes treated with vehicle or 40–80 nM antimycin for 48 h. APOE determinations were performed in cell lysates and conditioned media and expressed normalized to a control value. n=4. p was obtained with two-sided estimation statistics. Untreated iPSC-derived astrocytes secrete ~70 x 10⁴ pg/μg lysate. (**B–G**) Present analyses of changes in mRNA expression measured with NanoString Neuroinflammation panel. (**B**) Volcano plots of wild-type, *SLC25A1Δ*, and *SLC25A4Δ* HAP1 cells (n=3 per genotype) and iPSC-derived astrocytes treated with vehicle or 80 nM antimycin for 48 hr (n=3). Yellow symbols represent upregulated genes in mutant or drug treated cells. (**C**) mRNA expression of APOE and TBP was expressed as APOE /TADA2B or TBP/TADA2B ratios. TBP and TADA2B are both housekeeping control transcripts One-Way ANOVA followed by Benjamini-Krieger-Yekuiteli multiple comparison corrections (FDR = 5%). (**D**) Magnitude of compromise of significantly upregulated mRNAs annotated to secreted and mitochondrial proteomes in antimycin-treated astrocytes and compared to all genes present in the Neuroinflammation NanoString panel. p value was calculated with exact hypergeometric probability. (**E**) Circos plot of shared upregulated hits in *SLC25A1Δ*, *SLC25A4Δ* HAP1 cells, and iPSC-derived astrocytes treated with 80 nM antimycin. Outside arc represents the identity of each gene list. Inside arc represents a gene list, where each gene member of that list is assigned a spot on the arc. Dark orange color represents genes that are shared by multiple lists and light orange color represents genes that are unique to that gene list. Shared genes are presented by purple lines and different genes that belong to the same functional ontologies are connected by light blue lines. (**F**) Metascape ontology analysis and clustering of genes upregulated in *SLC25A1Δ* and *SLC25A4Δ* HAP1 cells and iPSC-derived astrocytes treated with 80 nM antimycin. Cumulative hypergeometric p-values.

The online version of this article includes the following figure supplement(s) for figure 6:

**Figure supplement 1.** Human iPSC astrocytes treated with antimycin differ in their gene expression as compared to A1 astrocytes.

We identified four upregulated mRNAs common to these three experimental conditions (*Figure 6E*, APOE, TSPAN18, PTPN6, and LOX). The number of hits increased to 45 upregulated mRNAs when considering shared functional annotations (*Figure 6E*). These functional annotations were mostly related to cytokine signaling ontologies (*Figure 6F and R* -HSA-1280215 and GO:0034097 Log q-value=−25.19 and –18.18). To explore whether the 85 upregulated genes in antimycin-treated astrocytes were part of a neurotoxic A1 reactive astrocyte phenotype associated with neurodegeneration, we compared the fold of induction of these transcripts with the changes in expression of these mRNAs in astrocytes after inflammatory LPS administration to elicit the A1 transcriptional phenotype (*Liddelow et al., 2017*). There was no correlation in gene expression among these datasets (*Figure 6—figure supplement 1*). These results suggest that increased expression of APOE induced by disruption of mitochondria co-occurs with inflammatory mechanisms, which are distinct from known expression signatures of reactive astrocytes induced by LPS.

## The expression of APOE and respiratory chain subunits are inversely correlated in a mouse model of Alzheimer's disease

Increased expression of APOE coinciding with an inflammatory response is reminiscent of Alzheimer's pathology. We hypothesized that if the expression of APOE and components of the electron transport chain were to be mechanistically linked, such an association should fulfill two predictions in a preclinical Alzheimer's brain model. First, there should be an inverse correlation in the protein levels of electron transport chain subunits and APOE if mitochondrial disruption were to induce increased APOE expression in diseased brains. Second, alterations in the levels of electron transport chain subunits should precede an APOE increase in diseased brains. To test this hypothesis, we used the 5xFAD mouse model of Alzheimer's disease that expresses human APP and presenilin 1, encoding five human pathogenic mutations that cause familial Alzheimer's disease (*Oakley et al., 2006*). Pathology in these mice progresses with aging, and these transgenes induce age-dependent expression of APOE and inflammatory gene products (*Bai et al., 2020*; *Oakley et al., 2006*). We analyzed a cohort of mice of both genotypes from ages 1.8–14.4 months (equally balanced for males and females per group), representing young adults to middle-aged adult mice. Neuropathology begins at 2 months of age while cognitive impairment begins after 4 months of age in the 5xFAD mouse model (*Girard et al., 2014*; *Oakley et al., 2006*).

We performed TMT mass spectrometry on the cortex from 86 mice and quantified over 8000 proteins from which we performed an analysis on APOE and 914 Mitocarta 3.0 proteins (*Rath et al., 2021*), including all subunits of the five complexes of the electron transport chain and 33 of the SLC25A transporters present in mitochondria (*Figure 7A*). We analyzed whether the expression of brain APOE correlated with the expression of these mitochondrial proteins and determined the effects of age, the 5xFAD genotype, and the interaction of these factors (*Figure 7A–C*). We measured composite protein abundance of all subunits belonging to each one of the respiratory complexes or composite SLC25An transporter protein abundance by calculating the first principal component of all proteins assigned to the complex or composite (Composite protein abundance, *Figure 7D*). APOE expression across all ages inversely correlated with the expression of complex I, II, and IV subunits in 5xFAD but not in wild-type animals (*Figure 7B–C*). We further examined the effects of age and genotype on the expression of APOE (*Figure 7D*) and all complexes of the electron transport chain (*Figure 7D*). The expression of respiratory complex subunits varied with age in wild-type animals (*Figure 7D*, gray symbols). For example, complex I subunits increased to a plateau at 6 months of age in wild-type cortex (*Figure 7D*, gray symbols). APOE expression remained constant with age in wild-type animals (*Figure 7D*, gray symbols). However, APOE expression progressively increased sevenfold in 5xFAD animals (*Figure 7D*, blue symbols), while most electron transport chain complexes decreased their expression with age in 5xFAD animals (*Figure 7C–D*). We detected a significant decrease in the expression of complex I and one of three mitochondrial Alzheimer's disease risk factors, COX7C, in 5xFAD cortex (*Bellenguez et al., 2022*). Importantly, these changes in respiratory chain protein levels preceded the APOE increase observed at 6 months of age in mutant animals (*Figure 7D*). These results reveal that changes in the expression of complex I and COX7C are followed by later increases in APOE in diseased cortex.

### *SLC25A1*-sensitive gene expression correlates with human cognitive trajectory during aging

We asked whether the expression of APOE, electron transport chain subunits, and SLC25A transporters correlates with age-dependent cognition in humans. We analyzed the Banner cohort of 106 individuals, of whom, 104 had normal cognitive performance at the time of enrollment (*Beach et al., 2015*). These adult brain donors were longitudinally assessed to determine their rate of change of cognitive performance over time (i.e. cognitive trajectory), irrespective of neuropathology. Cognitive trajectory refers to change of performance on the Mini-Mental State Exam (MMSE) over time (*Folstein et al., 1975*). The donor's quantitative brain proteomic profiles were obtained postmortem. In this and other cohorts, the expression of mitochondrial proteins, in particular complex I and III subunits, predicts cognitive preservation while proteins annotated to inflammatory ontologies predict a faster cognitive decline (*Johnson et al., 2022*; *Wingo et al., 2019*).

We assessed whether brain protein expression levels of gene products differentially expressed in SLC25A1-null cells, such as electron transport chain subunits (*Figure 1*), correlated with cognitive trajectory in the Banner cohort subjects. We used three differentially expressed gene sets from

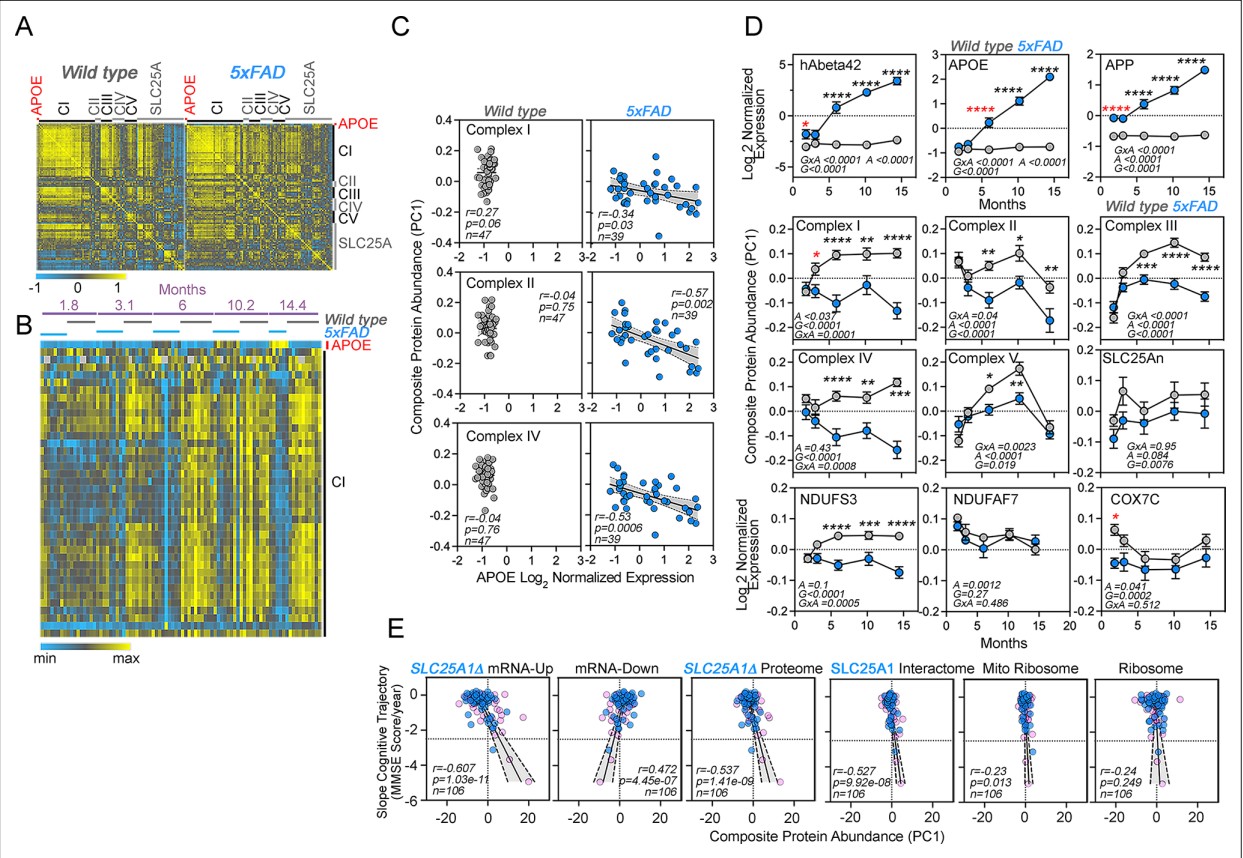

**Figure 7.** Correlative studies of APOE expression and electron transport chain subunits in an Alzheimer's mouse model and aging humans. (**A**) Protein expression similarity matrix of APOE, complexes I to V (CI-CV) of the electron transport chain, and transporters of the SLC25A family in wild-type and 5xFAD mouse models. Data were obtained by TMT mass spectrometry from mouse cortices. Similarity was calculated with Spearman Rank correlation. (**B**) Kendall Tau Hierarchical clustering of complex I subunits and APOE across ages and genotypes. (**C**) Correlation of composite protein abundance for complexes I, II, and IV in wild-type and 5xFAD mouse cortices with APOE levels. (**D**) Quantification of human A beta 42 peptide, APP, APOE, three mitochondrial Alzheimer's risk factors (NDUFS3, NDUFAF7, and COX7C), and the composite protein abundance for complex I to V, as well as members of the SLC25A family of mitochondrial transporters in wild-type and 5xFAD mice (grey and blue symbols, respectively). Two-way ANOVA followed by Šídák's multiple comparisons tests. Factors are age (**A**), genotype (**G**), and their statistical interaction (**I**). Asterisks denote significant differences between genotypes at a defined age. Red asterisks denote the earliest age with differences between genotypes. See **Supplementary file 1** for all Mitocarta hits in the 5xFAD mouse study. (**E**) The SLC25A1 RNAseq, proteome, and interactome correlate with the cognitive trajectory of human subjects. Cytoplasmic and mitochondrial ribosome subunits were used as controls. Graphs depict the correlation between the cognitive trajectory (Mini-Mental State Examination (MMSE)) in subjects belonging to the Banner collection that were longitudinally followed for an average of 14 years (n=106) (**Beach et al., 2015**; **Wingo et al., 2019**). The SLC25A1Δ RNAseq, up and downregulated hits, proteome hits, as well as the SLC25A1 interactome hits principal components were derived by estimating eigenvectors of the expression matrix of protein abundance data. Best-fit regression line drawn in blue and the 99.9% confidence interval for the regression line shaded in gray. No covariate was applied because sex, age at enrollment, and education have been regressed out of cognitive trajectory (**Wingo et al., 2019**). Blue circles represent males, pink circles represent females (48.1%).

The online version of this article includes the following figure supplement(s) for figure 7:

**Figure supplement 1.** Interactions between SLC25A1 and NDUFS3 and neuronal ontology annotated genes downregulated in SLC25A1 mutant cells.

SLC25A1-null cells: the downregulated SLC25A1Δ transcriptome dataset (**Figure 1F**), which is enriched in genes associated with neuronal differentiation, axon guidance, and neurotransmission ontologies (**Figure 1P**); the SLC25A1Δ upregulated mRNAs, which includes APOE and sterol synthesis genes (**Figure 1F and P**); and the differentially expressed proteins in the SLC25A1Δ TMT proteome (**Figure 1B and O**). These differentially expressed gene sets were represented by the first principal component of their protein expression levels in each one of the 106 human subjects in the Banner cohort (**Figure 7E**, PC1).

We confirmed the expression of some of the transcripts present in the downregulated SLC25A1Δ transcriptome by qRT-PCR, both in HAP1 cells and human neuroblastoma cells (**Figure 7—figure**

*supplement 1*), focusing on genes implicated in neurodevelopment and intellectual disability (*Abidi et al., 2002*; *Akita et al., 2018*; *Ayalew et al., 2012*; *Bassani et al., 2012*; *Becker et al., 2009*; *Curto et al., 2019*; *Kröcher et al., 2014*; *Kury et al., 2017*; *Lee et al., 2015*; *Leonardo et al., 1997*; *Picard et al., 2009*; *Shoukier et al., 2013*; *Srivastava et al., 2014*; *Tsuboyama and Iqbal, 2021*; *Vawter, 2000*; *Yan et al., 2018*; *Portnoï et al., 2000*; *Zurek et al., 2016*). In the Banner cohort, the brain protein expression representing the *SLC25A1Δ* upregulated mRNAs correlated with an accelerated cognitive decline ($r=-0.607$, $p<1.03e-11$, *Figure 7E*). In contrast, the brain protein expression representing the downregulated mRNAs associated with a slower rate of cognitive decline in the Banner cohort proteomes ($r=0.472$, $p<4.45e-07$, *Figure 7E*). Like the *SLC25A1Δ* upregulated mRNAs, increased brain expression of the differentially expressed proteins in the *SLC25A1Δ* TMT proteome (*Figure 1B*) also correlated with accelerated cognitive decline ($r=-0.537$, $p<4.45e-07$, Fig. *Figure 7E*). The *SLC25A1Δ* upregulated mRNAs and proteome shared APOE and sterol synthesis as a top ontology (*Figure 1O–P*). These data suggest that genes and proteins sensitive to SLC25A1 expression regulate cellular processes important for cognition.

We used an orthogonal SLC25A1 dataset obtained from SH-SY5Y neuroblastoma cells to further test the hypothesis that the network of proteins regulated by SLC25A1 influences cognition. We focused on the SLC25A1 interactome, the proteins that co-precipitate with SLC25A1. We previously identified the SLC25A1 interactome using quantitative mass spectrometry and found that this network of proteins is enriched in some respiratory chain subunits as well as other SLC25A inner mitochondrial membrane transporters (*Gokhale et al., 2021*). We reasoned that if the SLC25A1 interactome converges on similar biological processes as the *SLC25A1Δ* proteome or transcriptomes, then the SLC25A1 interactome should also have significant correlations with cognitive trajectory in the Banner cohort. In line with this reasoning, we found that greater expression of proteins belonging to the SLC25A1 interactome strongly correlated with a rapid cognitive decline in the Banner cohort ($r=-0.527$, $p<9.92e-08$, *Figure 7E*). We used as controls the mitochondrial and cytoplasmic ribosomes subunit datasets consisting of ~80 subunits each (MitoCarta 3.0 and CORUM complex #306). The mitochondrial ribosome correlated poorly with cognitive decline in the Banner cohort as compared to the SLC25A1 datasets even though the SLC25A1 interactome dataset was of a similar size to the mitochondrial ribosome dataset (*Figure 7E*, compare SLC25A1 interactome composed of 75 proteins and mitochondrial ribosome composed of 77 proteins). In addition, the expression of cytoplasmic ribosomal proteins did not correlate with the cognitive trajectory of the Banner subjects ($r=-0.24$ and $p=0.249$, *Figure 7E*). Taken together, our data suggest that a network of mitochondrial proteins, including the respiratory chain and several members of the SLC25A transporter family, regulates APOE expression and other cellular processes modulating cognition during aging.

## Discussion

Here, we report our discovery that mutations in nuclear-encoded mitochondrial transporter genes, *SLC25A1* and *SLC25A4*, modify the secretome and mitochondrially annotated proteome to a similar extent. Intriguingly, mutations in either one of these mitochondrial transporters upregulated the expression and secretion of the lipoprotein APOE in diverse cellular systems. We focused on APOE, as it is the main risk factor for Alzheimer's disease, a disease associated with compromised mitochondrial function (*Wang et al., 2020*). We show that this mitochondrial-dependent APOE upregulation phenotype occurs in response to loss of integrity of the electron transport chain secondary to either *SLC25A1* or *SLC25A4* mutagenesis. We demonstrate that disruption of the assembly and function of electron transport chain complexes I, III, and IV also increases APOE, arguing that a main initiating event in a cascade that increases APOE levels is the disruption of electron transport chain integrity and function. Both mutations that directly or indirectly compromise the electron transport chain increase APOE levels. For example, genetic disruption of copper loading into complex IV with mutants directly targeting copper loading factors, such as COX17 and COX19, increased APOE levels. A similar APOE phenotype is obtained by indirectly disrupting cellular copper homeostasis at the plasma membrane (SLC31A1 and ATP7A), the mitochondrial inner membrane (SLC25A3), or by pharmacological copper chelation with BCS, all conditions that affect complex IV (*Boulet et al., 2018*; *Cobine et al., 2021*; *Guthrie et al., 2020*). The mitochondrial protein network modulating APOE expression includes proteins encoded by prioritized genes within Alzheimer's disease risk loci necessary for electron transport chain complex I assembly and function (*de Rojas et al., 2021*; *Kunkle et al., 2019*). Furthermore,

we show that this mitochondrial regulation of APOE expression extends to brain cells, iPSC-derived human astrocytes, and co-occurs with an inflammatory gene expression response. Together, our data demonstrate that mitochondria robustly regulate APOE expression and secretion, placing them in a novel position upstream of APOE. We propose that this mitochondria-to-APOE mechanism may operate in the pathogenesis of dementia, a proposition supported by our protein expression correlation studies in mouse Alzheimer's brains and aging human brains.

Our results support the idea that there are other intramitochondrial mechanisms connecting mitochondria to APOE expression and secretion beyond the loss of integrity of the electron transport chain. Mutagenesis of the mitochondrial carnitine transporter SLC25A20 also upregulated APOE, despite these cells respiring normally and having wild-type levels of respiratory chain subunits. Similarly, mutagenesis of a cytosolic enzyme that controls the synthesis of acetyl-CoA using citrate as a substrate, ACLY, also increases APOE expression (*Figure 5—figure supplement 4*). Thus, we postulate that in addition to the integrity of the electron transport chain, there are other mitochondrial and cytoplasmic mechanisms regulating APOE expression.

We tested several alternative hypotheses that could account for a link between mitochondrial dysfunction and heightened nuclear expression of APOE. We measured the rates of oxygen consumption and extracellular acidification, the latter as a proxy for glycolysis (*Zhang and Zhang, 2019*). We found that while the oxygen consumption rate negatively correlated with APOE levels, the rate of extracellular acidification did not correlate with APOE expression (*Figure 5*). These observations make it unlikely that glycolytic adaptations in the mitochondrial mutants used in our studies could account for the APOE phenotype. We also assessed whether decreased cytoplasmic ATP levels led to APOE upregulation. The AMPK pathway senses drops in ATP cytoplasmic levels and coordinates a response to increase ATP generation when cellular energy is depleted (*Herzig and Shaw, 2018*). If decreased ATP levels mediate increased APOE protein levels, SLC25A1-null cells should display activation of the AMPK pathway at baseline. We found that, while SLC25A1-null cells are sensitized to respond to an AMPK-activating drug, the pathway is minimally active in the cells at baseline (*Figure 5—figure supplement 5*). This aligns with our previous finding that ATP levels between SLC25A1-null and wild-type cells do not differ (*Gokhale et al., 2019*). A second possible mechanism is that APOE expression is a coordinated response with an upregulation of cholesterol synthesis via SREBP transcription factors (*Horton et al., 2002*). We found that SLC25A1-null HAP1 cells had increased expression of cholesterol synthesis pathway enzymes, accompanied by elevated free cholesterol and cholesterol esters, consistent with SREBP transcription factor activation. However, these phenotypes were not shared by SLC25A1-null neuroblastoma cells or SLC25A4-null cells, even though these cells also displayed increased APOE expression and secretion (*Figure 2*). Thus, increased APOE resulting from mitochondrial dysfunction is not dependent on cholesterol synthesis pathways. A third mitochondria-to-nucleus pathway we investigated is mediated by activation of transcription factors ATF4 and CHOP (*Quirós et al., 2017*). These factors control the expression of mitokines in response to mitochondrial stress as part of the integrated stress response transcriptional pathway (*Chung et al., 2017*; *Kim et al., 2013*). Administration of doxycycline to trigger the stress response mounted an appropriate transcriptional response (*Quirós et al., 2017*) in both wild-type and SLC25A1-null cells (*Figure 5—figure supplement 5*), although this response was somewhat blunted in the mutant cells, even at baseline (*Figure 5—figure supplement 5*). Despite stress response activation, cellular and secreted levels of APOE protein were unaffected besides a mild increase in secretion in the SLC25A1-null cells (*Figure 5—figure supplement 5*). Furthermore, FCCP, also a potent activator of the mitochondrial stress response (*Quirós et al., 2017*) failed to induce APOE expression in HAP1 cells, even though inhibition of either complexes I or V with piericidin A or oligomycin increased APOE levels (*Figure 3—figure supplement 2*). Thus, the ATF4-dependent stress response alone cannot account for elevated APOE expression and secretion. A fourth mechanism that could account for the increased APOE expression is dependent on redox imbalance. We think this mechanism is unlikely because the antioxidant N-acetyl cysteine decreased the expression of the mitochondrial glutathione-disulfide reductase in both wild-type and SLC25A1-null cells, but it did not change APOE mRNA levels (*Figure 5—figure supplement 5*). Finally, we looked for transcriptional signatures in the SLC25A1 and SLC25A4 upregulated transcriptomes and found neither common predicted transcription factors that could account for the changes in gene expression in both genotypes, nor transcription factors known to regulate APOE transcription such as LXRs or C/EBPβ (Fig. S6C) (*Laffitte et al., 2001*; *Xia et al., 2021*). We

suggest that a screen targeting large swaths of the genome in astrocytes and mutant cell lines used in this study would be an effective method for uncovering further mechanisms by which mitochondria contribute to APOE expression regulation.

Our findings expand previous work demonstrating that mitochondrial distress regulates the secretion of inflammatory cytokines and type I interferons (*Dhir et al., 2018*; *Riley and Tait, 2020*; *Shimada et al., 2012*; *West et al., 2015*), the growth factor mitokines GDF15 and FGF21 (*Chung et al., 2017*; *Kim et al., 2013*), the production of mitochondrially-derived peptides encoded in the mitochondrial genome (*Kim et al., 2017*), and alpha fetoprotein in hepatocytes (*Jett et al., 2022*). Our study makes two key contributions expanding on these non-cell autonomous mechanisms. First, we identified the first apolipoprotein whose expression and secretion is modulated by mutations affecting mitochondria. Second, while we focus on APOE, we think this apolipoprotein is a harbinger of an extended upregulation of the secreted proteome mediated by mitochondria (*Uhlén et al., 2019*). The importance of this secreted proteome change can be inferred from its extent and magnitude, which is on par with the changes we observed in the mitochondrially annotated proteome. Our findings add to the growing notion that mitochondrial regulation of the secreted proteome is a more common process than previously appreciated (*Jett et al., 2022*; *Sturm et al., 2023*). We asked whether the APOE increase may be part of an inflammatory response by using NanoString mRNA quantification to assess the levels of activity of interleukin and interferon pathways in *SLC25A1Δ* and *SLC25A4Δ* cells and antimycin-treated astrocytes. Although *SLC25A4Δ* cells had a paucity of significant hits compared with *SLC25A1Δ* cells and antimycin-treated astrocytes, we found that all three experimental conditions converged on common hits whose number increased when considering hits belonging to shared ontologies. All cells showed increased levels of secreted cytokines and gene ontologies associated with their transcriptional profiles were enriched for cytokine signaling pathways. Thus, our data suggest that APOE upregulation in response to mitochondrial dysfunction could be part of an inflammatory response initiated by mitochondria. Although mitochondria are often viewed as downstream targets of neuroinflammation, our results with antimycin-treated astrocytes add to growing evidence that mitochondria drive inflammatory signaling in the nervous system (*Bader and Winklhofer, 2020*; *Joshi et al., 2019*; *Lin et al., 2022*).

What role does increasing APOE expression and secretion in response to mitochondrial dysfunction serve for the cell? Increased APOE secretion induced by mitochondrial damage could be adaptive or maladaptive depending on the lipidation and contents of APOE. APOE-mediated lipid exchange between cells can alter the lipid microenvironments of cellular membranes, thereby influencing cell signaling and homeostasis (*Martens et al., 2022*; *Tambini et al., 2016*). APOE is a primary lipoprotein in the brain produced mainly by astrocytes, though neurons and other glia also express APOE (*Belloy et al., 2019*; *Martens et al., 2022*). APOE particles play necessary roles in handling toxic lipids by shuttling them between cell types, with differential effects depending on the cell type from which the APOE particle originated, the lipid species loaded in the particle, and the fate of the lipid-loaded particle (*Guttenplan et al., 2021*; *Ioannou et al., 2019*; *Liu et al., 2017*). We speculate that either APOE-dependent removal of toxic factors from *SLC25A1* mutant cells, or wild-type conditioned media delivering a factor missing from *SLC25A1* mutant media, could explain our finding that dialyzed conditioned media from SLC25A1 mutant cells cannot support their own growth. Astrocytes play a prominent role in carrying and processing toxic lipids in the brain through APOE-dependent mechanisms. Since we observed increased APOE secretion in astrocytes, but not neurons, in response to antimycin, we speculate that APOE released from astrocytes following antimycin administration may not sustain either astrocytes and/or neurons. Profiling the contents of antimycin-induced APOE particles released from astrocytes could further clarify their potential impact on the function and health of neighboring cells.

The APOE E4 allele is the strongest genetic risk factor for sporadic Alzheimer's disease and APOE, along with amyloidogenic processing of the amyloid precursor protein (APP) into Aβ, is thought of as an initiating and driving factor in disease etiology (*Frisoni et al., 2022*; *Huang and Mahley, 2014*; *Mahley, 2023*; *Martens et al., 2022*). While mitochondrial dysfunction is acknowledged as an important factor in Alzheimer's disease, mitochondria are typically placed downstream of Aβ and APOE, with Aβ or the APOE4 allele perturbing mitochondrial function (*Area-Gomez et al., 2020*; *Chen et al., 2011*; *Mahley, 2023*; *Orr et al., 2019*; *Tambini et al., 2016*; *Yin et al., 2020*). Some have proposed that mitochondria act as upstream factors in Alzheimer's pathogenesis through metabolic

and bioenergetic effects (*Rangaraju et al., 2018*; *Swerdlow, 2018*; *Wang et al., 2020*). We argue that mitochondria could also participate in Alzheimer's pathogenesis through modulation of APOE allele-dependent mechanisms. The increased expression of APOE, which is inversely correlated with the expression of respiratory chain subunits in the 5xFAD mouse model, supports this proposition (*Figure 7*). For example, increased expression of APOE E4 downstream of mitochondrial dysfunction could either initiate or exacerbate the effects of the APOE4 allele in neurodegenerative processes, since APOE4 is prone to aggregation and poor lipidation (*Gong et al., 2002*; *Hatters et al., 2006*; *Hubin et al., 2019*). Our findings open the possibility that mitochondria could act as initiators or drivers of Alzheimer's pathogenesis through modulation of APOE-dependent disease processes. The correlation of expression levels in the prefrontal cortex of proteins dysregulated in the SLC25A1 proteome and transcriptome with human cognitive trajectory provides evidence for this conception. Together, our work supports the idea that mitochondria influence brain function and cognition in part through modulation of the secretome, including a novel role in regulation of APOE expression and secretion.

# Materials and methods
## Cell lines, gene editing, and culture conditions
Human haploid (HAP1) isogenic cell lines were obtained from Horizon Discovery. In addition to the parental wild-type line (C631, RRID: CVCL_Y019), the following CRISPR/Cas9-edited knockout cell lines were used: SLC25A1 (HZGHC001753c010, RRID: CVCL_TM05 and HZGHC001753c003, RRID: CVCL_TM04), SLC25A3 (HZGHC000792c010, RRID:CVCL_TM31), SLC25A4 (HZGHC000778c011, RRID:CVCL_TM45), SLC25A20 (HZGHC000787c00, RRID:CVCL_TM21), SLC25A50/MTCH2 (HZGHC23788), NDUFS3 (HZGHC4722, RRID:CVCL_XQ89), NDUFAF7 (HZGHC55471), PARK2 (HZGHC003208c002, RRID:CVCL_TC07), and ACLY (HZGHC005811c011, RRID: CVCL_SB23 and HZGHC005811c002, RRID: CVCL_XK97). All HAP1 cells were grown in IMDM (Corning 10–016) with 10% Fetal Bovine Serum (FBS) (VWR, 97068–085) in a 10% CO2 incubator at 37 °C, unless otherwise indicated. In experiments where a single clone of SLC25A1-null cells was used, the cell line HZGHC001753c010 (RRID: CVCL_TM05) was used. In experiments where a single clone of ACLY-null cells was used, the cell line HZGHC005811c002 (RRID: CVCL_XK97) was used. For each knockout cell line, an individual control line of the HAP1 parental line that was received with the particular knockout line was used. Mutants and their unique control line were cultured and handled as parallel pairs to avoid passage- and culture-induced variation.

Human neuroblastoma SH-SY5Y cells (ATCC, CRL-2266; RRID:CVCL_0019) were grown in DMEM media (Corning, 10–013) containing 10% FBS (VWR, 97068–085) at 37 °C in 10% CO$_2$, unless otherwise indicated. SH-SY5Y cells deficient in SLC31A1 were genome edited using gRNA and Cas9 preassembled complexes by Synthego with a knock-out efficiency of 97%. The gRNAs used were UUGGUGAUCAAUACAGCUGG which targeted transcript ENST00000374212.5 exon 3. Wild-type and mutant cells were cloned by limited dilution and mutagenesis was confirmed by Sanger sequencing with the primer: 5'GGTGGGGGCCTAGTAGAATA. SH-SY5Y cells deficient in ATP7A were genome edited by Synthego using gRNA and Cas9 preassembled complexes with a knock-out efficiency of 80%. The gRNAs used were ACAGACUCCAAAGACCCUAC which targeted transcript ENST00000341514 exon 3. Wild-type and mutant cells were cloned by limited dilution and mutagenesis was confirmed by Sanger sequencing with the primer: 5'TGCCTGATAGGTACCACAGTC. SH-SY5Y cells deficient in Cox17 were genome edited by Synthego using gRNA and Cas9 preassembled complexes with a knock-out efficiency of 94%. The gRNAs used were CCAAGAAGGCGCGCGAUGCG which targeted transcript ENST00000261070 exon 1. Wild-type and mutant cells were cloned by limited dilution and mutagenesis was confirmed by Sanger sequencing with the primer: 5'AGGCCCAATAATTATCTCCAGAGC. The Cox17-deficient cells were supplemented with 50 µg/ml uridine (Sigma, U3003) in their growth media. SH-SY5Y cells deficient in SLC25A1 were genome edited by Synthego using gRNA and Cas9 preassembled complexes with a knock-out efficiency of 86%. The gRNAs used were GGGTTCCCGGTCCCTGCAGG which targeted transcript ENST00000215882 exon 2. Wild-type and mutant cells were cloned by limited dilution and mutagenesis was confirmed by Sanger sequencing with the primer: 5'GATGGAACCGTAGAGCAGGG. For each mutant cell line, an individual control line of the SH-SY5Y parental line that was received with the particular mutant line was used. In APOE protein

measurement immunoassays, two separate clones of cells were used to exclude clonal or off-target effects.

ATP7A LL cells contained a di-leucine mutation generated by introducing alanine in L1750 and 1751 by the Emory Genomics Core into the SH-SY5Y ATP7A CRISPR KO cells. HA-tagged ATP7A cDNA in pQCXIP backbone vector was provided by Dr. Michael Petris. SH-SY5Y ATP7A LL and empty vector expressing cells were generated by transfecting ATP7A KO10 cells with lipofectamine 3000 (Invitrogen). Selection was started after 24 hr with 0.5 µg/ml Puromycin (Gibco). Growth of clones was boosted by supplementing the media with 1 µM Copper until sufficient cells grew to freeze down and passage well. All ATP7A and SLC31A1 KO cells were validated by measuring copper content using ICP mass spectrometry.

HEK293 ANT1 single knockout, ANT triple knockout and their wild-type control cells were grown in DMEM containing 10% FBS, supplemented with 2 mM L-glutamine (HyClone, SH30034.01), 100 µg/ml zeocin (Gibco, R25001), and 15 µg/ml Blasticidin S (Sigma, SBR00022). The ANT1, ANT2, ANT3 triple knockout cells were additionally supplemented with 1 mM sodium pyruvate (Sigma, S8636) and 50 µg/ml uridine (Sigma, U3003). gRNAs that targeted ANT1, ANT2, and ANT3 were separately cloned into PX459, a gift from Feng Zhang (Addgene). gRNA sequences are GATGGGCGCTACCGCCGTCT (for ANT1), GGGGAAGTAACGGATCACGT (for ANT2), CGGCCGTGGCTCCGATCGAG (for ANT3), respectively. TREx 293 cells (Invitrogen R71007) were transfected with the PX459 constructs individually in order of ANT3, ANT2, and ANT1. Cell lines lacking only ANT1 were transfected once only with the ANT1 PX459 construct. Following each transfection, cells were selected with puromycin (2 µg/ml) for 72 hr, single clones were isolated by ring cloning and the absence of the protein expression was analyzed by immunoblotting before proceeding to the next transfection.

HEK293 cells knockout cell lines for HIGD1A, COX7A2L, COX18, COX19, and COX20 and their control cell line were grown in DMEM media (Corning, 10–013) containing 10% FBS (VWR, 97068–085), 50 µg/ml uridine (Sigma, U3003), and 1 x GlutaMAX (Gibco, 35050061) at 37 °C in 10% $CO_2$. Details about the generation of the cell lines using TALEN can be found in *Nývltová et al., 2022* (*Nývltová et al., 2022*, COX19), *Bourens and Barrientos, 2017* (*Bourens and Barrientos, 2017*, COX18), *Lobo-Jarne et al., 2018* (*Lobo-Jarne et al., 2018*, COX7A2L), (*Timón-Gómez et al., 2020a*, HIGD1A), and (*Bourens and Barrientos, 2017*, COX20).

This work used cell lines which were obtained from ATCC and Horizon Discovery. Cell lines were authenticated by provider and were tested to be free of mycoplasma in house.

## Antibodies

| Antibody | Dilution | Cat.No | RRID |
|---|---|---|---|
| ACTIN | 1:5000 | A5441 | AB_476744 |
| AMPK | 1:1000 | 2532 S | AB_330331 |
| APOE | 1:250 | 60531 | AB_920623 |
| COX4 | 1:1000 | 4850 | AB_2085424 |
| FLAG | 1:1000 | A190-102A | AB_67407 |
| HSP90 | 1:1000 | 610418 | AB_397798 |
| HSP90 | 1:1000 | 610418 | AB_397798 |
| mouse HRP | 1:5000 | A10668 | AB_2534058 |
| MT-ATP6 | 1:500 | 55313–1-AP | AB_2881305 |
| MTCO2 | 1:1000 | AB110258 | AB_10887758 |
| NDUFB11 | 1:1000 | ab183716 | |
| NDUFS3 | 1:200 | 15066–1-AP | AB_2151109 |
| OXPHOS mix | 1:250 | ab110412 | AB_2847807 |
| P-AMPK | 1:1000 | 2535T | AB_331250 |

*Continued on next page*

*Continued*

| Antibody | Dilution | Cat.No | RRID |
|---|---|---|---|
| rabbit HRP | 1:5000 | G21234 | AB_2536530 |
| SDHA | 1:1000 | 11998 | AB_2750900 |
| SLC25A1 | 1:500 | 15235–1-AP | AB_2297856 |
| TFRC | 1:1000 | 13–6800 | AB_86623 |
| UQCRC2 | 1:500 | ab 14745 | AB_2213640 |
| Antibody | Dilution | Cat.No | RRID |
| SDHA | 1:1000 | 11998 | AB_2750900 |
| SLC25A1 | 1:500 | 15235–1-AP | AB_2297856 |
| HSP90 | 1:1000 | 610418 | AB_397798 |
| MT-ATP6 | 1:500 | 55313–1-AP | AB_2881305 |
| MTCO2 | 1:1000 | AB110258 | AB_10887758 |
| TFRC | 1:1000 | 13–6800 | AB_86623 |
| ACTIN | 1:5000 | A5441 | AB_476744 |
| COX4 | 1:1000 | 4850 | AB_2085424 |
| OXPHOS mix | 1:250 | ab110412 | AB_2847807 |
| NDUFS3 | 1:200 | 15066–1-AP | AB_2151109 |
| UQCRC2 | 1:500 | ab 14745 | AB_2213640 |
| HSP90 | 1:1000 | 610418 | AB_397798 |
| APOE | 1:250 | 60531 | AB_920623 |
| AMPK | 1:1000 | 2532 S | AB_330331 |
| P-AMPK | 1:1000 | 2535T | AB_331250 |
| FLAG | 1:1000 | A190-102A | AB_67407 |
| mouse HRP | 1:5000 | A10668 | AB_2534058 |
| rabbit HRP | 1:5000 | G21234 | AB_2536530 |

## Primers

| Primer | Forward Sequence | Reverse Sequence |
|---|---|---|
| ACAT2 | CCCAGAACAGGACAGAGAATG | AGCTTGGACATGGCTTCTATG |
| ACLY | CTCACTAAGCCCATCGTCTG | TCCTTCAAAGCCTGGTTCTTG |
| APOE | TGGGTCGCTTTTGGGATTAC | TTCAACTCCTTCATGGTCTCG |
| ASNS | ATCACTGTCGGGATGTACCC | TGATAAAAGGCAGCCAATCC |
| ATF3 | GGAGCCTGGAGCAAAATGATG | AGGGCGTCAGGTTAGCAAAA |
| ATF4 | CAGCAAGGAGGATGCCTTCT | CCAACAGGGCATCCAAGTC |
| ATF5 | GAGCCCCTGGCAGGTGAT | CAGAGGGAGGAGAGCTGTGAA |
| CAMK2B | CAGCCAGAGATCACCAGAAG | CACCAGTGACCAGATCGAAG |
| CHAC1 | GTGGTGACGCTCCTTGAAGA | TTCAGGGCCTTGCTTACCTG |
| CHL1 | ATGGCTCCCCAGTTGACA | TGATTTGGTTGAAGGTTGGTAA |
| CHOP | AGCCAAAATCAGAGCTGGAA | TGGATCAGTCTGGAAAAGCA |

*Continued on next page*

*Continued*

| Primer | Forward Sequence | Reverse Sequence |
| --- | --- | --- |
| GABRA5 | CCTCCATATTCACCTGCTTCA | CTGGTTGGCATCTGTGAAAAG |
| GABRB2 | ACTCAGAATCACAACCACAGC | CCACGCCAGTAAAACTCAATG |
| GSR | TTCCAGATGTTGACTGCCTG | GCCTTTGACGTTGGTATTCTG |
| HMGCR | ACAGATACTTGGGAATGCAGAG | CTGTCGGCGAATAGATACACC |
| LDLR | TTCACTCCATCTCAAGCATCG | ACTGAAAATGGCTTCGTTGATG |
| MSMO1 | TGAACTTCATTGGAAACTATGCTTC | TCTTTCAGGAAGGTTTACGTGAG |
| NCAM1 | TTGTTTTTCCTGGGAACTGC | ACTCTCCAACGCTGATCTCC |
| NRG2 | CACTCCTGTTCTCCTTCTCAC | TTTGCTGGTACCCACTGATG |
| PCBP1 | AAGACTTGACCACGTAACGAG | ATGCTTCCTACTTCCTTTCCG |
| PCK2 | CATCCGAAAGCTCCCCAAGT | GCTCTCTACTCGTGCCACAT |
| RER1 | CTTTCTTCGACGCTTTCAACG | CTGTACCTTCTCTTCCCATGTG |
| RPS20 | TGCTGACTTGATAAGAGGCG | GATCCCACGTCTTAGAACCTTC |
| SLC13A5 | AGAGGTTGTGTAAGGCCATG | AGCAAAGTTCACGAGGTCC |
| SLC25A20 | AGAAGCTGTACCAGGAGTTTG | ACTGACCCTCTTTCCCTCC |
| ST8SIA4 | CATTAGGAAGAGGTGGACGATC | AGAGCTATTGACAAGTGACCG |
| TBP | GAGAGTTCTGGGATTGTACCG | ATCCTCATGATTACCGCAGC |
| TRPC3 | CAAATGCAGAAGGAGAAGGC | CGTGTTGGCTGATTGAGAATG |
| TSPAN7 | CCTTATTGCCGAGAACTCCAC | ACACCAGGGACAGAAACATG |
| UNC5A | CTGTACCAGTGACCTCTGTG | AAACGAGGATGAGGACAAGC |
| VAMP2 | TCATCTTGGGAGTGATTTGCG | GGGCTGAAAGATATGGCTGAG |
| WARS | TCAGCAACTCATTCCCACAG | GCAGGGCTGGTTTAGGATAG |

## Generation of iNeurons from human iPSCs

Accutase (Gibco, A11105) was used to disassociate iPSC cells. On day −2, the cells were plated on a 12-well plate coated in matrigel (Corning, 354230) in mTeSR medium (StemCell Technologies, 85850) at a density of at 380,000 cells per well. On day −1, hNGN2 lentivirus (TetO-hNGN2-P2A-PuroR (RRID: Addgene_79049) or TetO-hNGN2-P2A-eGFP-T2A-PuroR (RRID: Addgene_79823)) together with FUdeltaGW-rtTA (RRID: Addgene_19780) lentivirus were added in fresh mTeSR medium containing 4 µg/µl polybrene (Sigma, TR-1003). The lentiviruses were added at 1×106 pfu/ml per plasmid (multiplicity of infection (MOI) of 2). On day 0, the culture medium was replaced with fresh KSR, consisting of KnockOut DMEM (Gibco, 10829018), Knockout Replacement Serum (Gibco, 10828028), 1 X Glutamax (Gibco, 35050061), 1 X MEM Non-essential Amino Acids (NEAA, Gibco, 11140050) and 100 µM β-Mercaptoethanol (Gibco, 21985023). In addition, 2 µg/ml doxycycline (Sigma, D9891) was added to the media, inducing TetO gene expression. Doxycycline was retained in the medium until the end of the experiment. Puromycin selection (5 µg/ml) was started on day 1. On day 4, the culture medium was replaced with Neurobasal medium (Gibco, 21103049), supplemented with B27 supplement (Gibco, 17504044), 1 X Glutamax, 20% Dextrose (Sigma, D9434), 10 ng/ml BDNF (Peprotech, 45002), 10 ng/ml GDNF (PeproTech, 45010), 2 µg/ml doxycycline (Sigma, D9891) and 5 µg/mL puromycin (InvivoGen, ant-pr-1). Beginning on day 7, half of the medium in each well was replaced every week. On day 24, hNGN2-induced neurons were assayed (*Zhang et al., 2013*).

## Differentiation of iPSCs into forebrain-specific neural progenitors and astrocytes

One hour of 1 mg/ml collagenase (Therm Fisher Scientific, 17104019) treatment was used to detach iPSC colonies. Following collagenase treatment, cells were suspended in embryoid body (EB) medium in non-treated polystyrene plates for 7 days. During this time, the medium was changed daily. EB medium consisted of DMEM/F12 (Gibco, 11330032), 20% Knockout Serum Replacement (Gibo, 10828028), 1 X Glutamax (Gibco, 35050061), 1 X MEM Non-essential Amino Acids (NEAA, Gibco, 11140050), 100 µM β-Mercaptoethanol (Gibco, 21985023), 2 µM dorsomorphin (Tocris, 3093) and 2 µM A-83 (Tocris, 692). After 7 days, EB medium was replaced by neural induction medium (hNPC medium), consisting of DMEM/F12, 1 X N2 supplement (Gibco, 17502048), B27 supplement, 1 X NEAA, 1 X Glutamax, 2 µg/ml heparin (Sigma) and 2 µM cyclopamine (Tocris, 1623). On day 7, the floating EBs were then transferred to Matrigel-coated six-well plates to form neural tube-like rosettes. The attached rosettes were kept for 15 days. During this time, the hNPC media of the rosettes was changed every other day. On day 22, the rosettes were picked mechanically and transferred to low attachment plates (Corning) in hNPC medium containing B27 supplement.

For astrocyte differentiation (*Tcw et al., 2017*), resuspended neural progenitor spheres were disassociated with accutase at 37 °C for 10 min. After disassociation, they were placed on Matrigel-coated six-well plates. Forebrain NPCs were maintained at high density in hNPC medium. To differentiate NPCs to astrocytes, disassociated single cells were seeded at a density of 15,000 cells/cm$^2$ on Matrigel-coated plates and grown in astrocyte medium (ScienCell: 1801, astrocyte medium (1801-b), 2% fetal bovine serum (0010), astrocyte growth supplement (1852) and 10 U/ml penicillin/streptomycin solution (0503)). From day 2, cells were fed every 48 hr for 20–30 days. Astrocytes were split when the cells reached 90–95% confluency (approximately every 6–7 days) and seeded again at their initial seeding density (15,000 cells/cm$^2$) as single cells in astrocyte medium.

## Seahorse metabolic oximetry

Extracellular flux analysis was performed on the Seahorse XFe96 Analyzer (Seahorse Bioscience) following manufacturer recommendations. HAP1 cells and SH-SY5Y cells were seeded at a density of 40,000 cells/well and HEK293 cells were seeded at a density of 20,000 cells/well on Seahorse XF96 V3-PS Microplates (Agilent Technologies, 101085–004) after being trypsinized and counted (Bio-Rad TC20 automated Cell Counter) All Hap1 cells were grown in normal growth media (IMDM) with 10% FBS except for Ndufaf7-null cells and their control line, which were also supplemented with uridine at 50 µg/ml (Sigma, U3003). SH-SY5Y cells were grown in normal growth media (DMEM) with 10% FBS, except for COX17-deficient cells and their control line, which were also supplemented with uridine at 50 µg/ml (Sigma, U3003). XFe96 extracellular flux assay kit probes (Agilent Technologies, 102416–100) incubated with the included manufacturer calibration solution overnight at 37 °C without $CO_2$ injection. The following day, wells were washed twice in Seahorse stress test media. The stress test media consisted of Seahorse XF base media (Agilent Technologies, 102353–100) with the addition of 2 mM L-glutamine (HyClone,SH30034.01), 1 mM sodium pyruvate (Sigma, S8636), and 10 mM D-glucose (Sigma, G8769). After washes, cells incubated at 37 °C without $CO_2$ injection for approximately 1 hr prior to the stress test. During this time, flux plate probes were loaded and calibrated. After calibration, the flux plate containing calibrant solution was exchanged for the Seahorse cell culture plate and equilibrated. Seahorse injection ports were filled with 10-fold concentrated solution of oligomycin A (Sigma, 75351), carbonyl cyanide-4-(trifluoromethoxy)phenylhydrazone (FCCP; Sigma, C2920), and rotenone (Sigma, R8875) mixed with antimycin A (Sigma, A8674) for final testing conditions of oligomycin (1.0 µM), FCCP (0.125 µM for Hap1 and HEK293, 0.25 µM for SH-SY5Y), rotenone (0.5 µM), and antimycin A (0.5 µM). In antimycin A sensitivity experiments, a separate solution of antimycin A was made at 100 nm and 50 nm concentrations to give a concentration of 10 nm and 5 nm in the well. In bongkrekic acid (BKA) sensitivity experiments, ready-made BKA solution (Sigma, B6179) was added to seahorse media to obtain solutions at at 25 µM and 50 µM, giving a final concentration of 0.25 µM and 0.5 µM in the well. Seahorse drugs were dissolved in DMSO and diluted in Seahorse stress test media for the Seahorse protocol. The flux analyzer protocol included three basal read cycles and three reads following injection of oligomycin A, FCCP, and rotenone plus antimycin A. In antimycin sensitivity experiments, antimycin A was injected following the basal read cycles and ten reads were taken before the protocol proceeded as usual with injections of oligomycin A, FCCP, and rotenone plus

antimycin A. Each read cycle included a 3 min mix cycle followed by a 3 min read cycle where oxygen consumption rate was determined over time. In all experiments, oxygen consumption rate readings in each well were normalized by protein concentration in the well. Protein concentration was measured using the Pierce BCA Protein Assay Kit (Thermo Fisher Scientific, 23227) according to manufacturer protocol. The BCA assay absorbance was read by a BioTek Synergy HT microplate reader using Gen5 software. For data analysis of oxygen consumption rates, the Seahorse Wave Software version 2.2.0.276 was used. Experiments were repeated at least in triplicate. Non-mitochondrial respiration was determined as the lowest oxygen consumption rate following injection of rotenone plus antimycin A. Basal respiration was calculated from the oxygen consumption rate just before oligomycin injection minus the non-mitochondrial respiration. Non-mitochondrial respiration was determined as the lowest oxygen consumption rate following injection of rotenone plus antimycin A. ATP-dependent respiration was calculated as the difference in oxygen consumption rates just before oligomycin injection to the minimum oxygen consumption rate following oligomycin injection but before FCCP injection. Maximal respiration was calculated as the maximum oxygen consumption rate of the three readings following FCCP injection minus non-mitochondrial respiration.

## Electrochemiluminescent immunoassays for APOE protein measurement

HAP1, SH-SY5Y, and HEK293 cells were plated in six-well dishes (Falcon, 353046). Each experimental condition was plated in three or four replicate wells. iPSC neurons and astrocytes were plated in 24-well dishes and grown to approximately 80% confluence. For iPSC cells, eight wells were used for each experimental condition and material from two wells was combined to generate an experimental replicate. For experiments where there was no drug treatment, cells were left overnight and samples were collected 22–26 hr later. In these experiments, Hap1 cells were plated at a density of 750,000 cells per well and SHSY5Y and HEK293 cells were plated at a density of 1,000,000 cells per well. In experiments with doxycycline (Sigma, D9891) or BKA (Sigma, B6179), cells were left incubating for 46–50 hr in drug or vehicle (cell culture grade water for doxycycline or 0.01 M Tris buffer at pH 7.5 for BKA) before sample collection, with fresh drug and vehicle media applied after approximately 24 hr. In these experiments, Hap1 cells were plated at a density of 300,000 cells per well and SH-SY5Y cells were plated at a density of 500,000 cells per well. In experiments with E-64 (Sigma E8640), brefeldin A (BFA, Sigma B6542) and cycloheximide (CHX, Thermo Scientific AC357420010), HAP1 cells were plated at a density of 750,000 cells/well in and left to grow overnight. The following day, growth media was exchanged for media containing the drug or vehicle (DMSO) and the cells were left for 8 hr before collection of the conditioned media and cell lysis. The following drug concentrations were used: E-64–50 µm, BFA – 5 µg/mL, CHX – 20 µg/mL.

To collect samples of cell lysate and conditioned media, cells were washed twice with ice-cold PBS containing 0.1 mM $CaCl_2$ and 1.0 mM $MgCl_2$, and then lysed on a rocker at 4 °C in Buffer A (10 mM HEPES, 0.15 M NaCl, 1 mM EGTA, and 100 µM $MgCl_2$)+0.5% Triton X-100 containing 10% protease inhibitor cocktail cOmplete (Roche, 118614500, prepared at 50 x in distilled water). The lysis was spun down at 4 °C at 13,000 rpm for 20 min. The supernatant was collected and flash frozen for APOE protein measurement. Conditioned media was spun down at 4 °C at 13,000 rpm for 15 min to clarify before collecting and flash freezing for APOE protein measurement. Growth media unexposed to cells was incubated and collected in the same manner to serve as a control for APOE present in the media that might be detected by the antibody. Cell lysis and media samples were stored at –80 °C for up to a month before APOE was measured. Media was spun down again at 4°C, 2000rpm for 5 min to get rid of any precipitate after thawing. APOE measurements were conducted by the Emory Immunoassay Core facility using electrochemiluminescent detection of the APOE antibody (Meso Scale Diagnostics, F212I) according to manufacturer protocols with the suggested reagents (Diluent 37 (R50A), MSD GOLD Read Buffer A (R92TG), and MSD GOLD 96-well Small Spot Streptavidin SECTOR Plates (L45SA-1)). Samples were not diluted for the electrochemiluminescence protocol (*Chikkaveeraiah et al., 2012*; *Gaiottino et al., 2013*).

## Conditioned media dialysis

HAP1 SLC25A1-null cells and their control cell line were grown for approximately 48 hr to confluency before collecting their media. The conditioned media was spun down for 5 min at 800 *g* and 4 °C and

then filtered with a syringe and 0.22 μm filter (Millipore, SLGV033RS) and stored at 4 °C for no more than three days before being applied to cells. Conditioned media was dialyzed using 10 K molecular weight cutoff cassettes (Thermo Scientific, 88252) according to manufacturer instructions. Cassettes were dialyzed three times consecutively with the volume in the cassette to the naive media (IMDM, Corning 10–016) being 1:1000. The media was exchanged after 3 hr of dialysis for each cycle, with the final dialysis cycle occurring overnight. The dialyzed media was collected the next morning and stored at 4 °C for no more than 3 days before being applied to cells. Viability of cells exposed to the conditioned media was assessed using an Alamar blue assay. For Alamar blue cell viability, HAP1 cells were plated at a density of 2000 cells/well in a 96-well plate and allowed to adhere for 30 min in normal growth media, IMDM (Corning 10–016) with 10% FBS, before this media was swapped out with conditioned media (dialyzed or undialyzed) collected previously. Cells grew in conditioned media for approximately 48 hr, with fresh conditioned media being put on after approximately 24 hr. After 48 hr, viability was measured with the BioTek Synergy HT plate reader using Gen5 software after a 2-hr incubation in Alamar blue (R&D Systems, AR002). Wells without cells and incubated in Alamar blue were used as a background reading.

## RNA extraction, cDNA preparation, and qPCR

In all experiments, cell growth media was changed approximately 24 hr before RNA extraction occurred. At least three replicate plates per experimental condition were used in each experiment. For experiments with doxycycline (Sigma, D9891) and N-Acetyl-L-cysteine (Sigma, A9165), cells incubated in the drug or vehicle (cell culture grade water, Corning 25–055) for 48 hr and fresh drug or vehicle was applied in new media 24 hr before RNA extraction. Doxycycline was applied at 9.75 μM and N-Acetyl-L-cysteine was applied at 2 mM. RNA was extracted from cells using Trizol reagent (Invitrogen, 15596026). Cells were washed twice in ice-cold PBS containing 0.1 mM $CaCl_2$ and 1.0 mM $MgCl_2$ and then 1 ml of Trizol was added to the samples. A cell scraper was used before collecting the sample into a tube. The samples in Trizol incubated for 10 min at room temperature on an end-to-end rotator and then 200 μl of chloroform was added to each tube. After vigorous vortexing and a brief incubation, the chloroform mixture was centrifuged at 13,000 rpm at 4 °C for 15 min. The aqueous layer was collected and 500 μl of isopropanol was added to it. The isopropanol mixture then rotated for 10 min at room temperature, followed by centrifugation at 13,000 rpm for 15 min at 4 °C. The supernatant was discarded and the remaining pellet was washed with freshly prepared 75% ethanol in Milli-Q purified water using a brief vortex to lift the pellet. After washing, the pellet in ethanol was centrifuged again for 5 min at 13,000 rpm at 4 °C. The ethanol was aspirated and the pellet was allowed to air dry. Lastly, the pellet was dissolved in 20–50 μl of molecular grade RNAase-free water and stored overnight at –20 °C before the concentration and purity were checked using the Nanodrop One$^C$ (Thermo Fisher Scientific). cDNA was synthesized using 5 μg RNA as a template per reaction with the Superscript III First Strand Synthesis System Kit (Invitrogen, 18080–051) and its provided random hexamers. RNA was incubated with the random hexamers and dNTPs at 65 °C for 5 min. The samples were then placed on ice while a cDNA synthesis mix was prepared according to the kit instructions. The cDNA synthesis mix was added to each of the tubes and the samples were then incubated at 25 °C for 10 min, followed by 50 min at 50 °C. The reaction was terminated at 85 °C for 5 min. Finally, the samples were treated with kit-provided RNase H at 37 °C for 20 min. A BioRad T100 thermal cycler was used to carry out this synthesis protocol.

For qPCR, the Real-Time qPCR Assay Entry on the IDT website was used to design primers. Primers were synthesized by Sigma-Aldrich Custom DNA Oligo service. Primer annealing and melting curves were used to confirm primer quality and specificity for single transcripts. qRT-PCR was performed using LightCycler 480 SYBR Green I Master (Roche, 04707516001) with 1 μl of the newly synthesized cDNA on the QuantStudio 6 Flex instrument (Applied Biosystems) in a 96-well format. The qRT-PCR protocol carried out by the QuantStudio 6 Flex consisted of initial denaturation at 95 °C for 5 min, followed by 45 cycles of amplification with a 5 s hold at 95 °C ramped at 4.4 °C/s to 55 °C. Temperature was maintained for 10 s at 55 °C, then ramped up to 72 °C at 2.2 °C/s. Temperature was held at 72 °C for 20 s and a single acquisition point was collected before ramping at 4.4 °C/s to begin the cycle anew. The temperature was then held at 65 °C for 1 min and ramped to 97 °C at a rate of 0.11 °C/s. Five acquisition points were collected per °C. Standard curves collected for each individual primer set were used to quantify data using QuantStudio RT-PCR Software version 1.2.

## Lipidomics

HAP1 or SH-SY5Y cells were grown to 80–90% confluency on 15 cm sterile dishes in normal growth media. Material from two to four plates was combined for a single replicate and four replicates were used for an experiment. The cells were washed three times in ice-cold PBS containing 10 mM EDTA and then incubated for approximately 30 min at 4 °C. After incubation, the cells were lifted from the plate on ice and spun down at 800 rpm for 5 min at 4 °C. Since SH-SY5Y cells lifted easily, the incubation step was skipped in these experiments. Cells were resuspended in ice-cold PBS and a cell count was then taken (Bio-Rad TC20 automated Cell Counter) for normalization and to ensure cells were at least 90% alive and at least 20 million cells would be present in each replicate sample. After counting, cells were again spun down at 800 rpm for 5 min at 4 °C and the resulting cell pellets were flash-frozen and stored at –80 °C for a few weeks before lipidomic analysis was performed by the Emory Integrated Metabolomics and Lipidomics Core.

Briefly, for lipidomics HPLC grade water, chloroform, methanol, and acetonitrile were purchased from Fisher Scientific (Hampton, NH, USA). Formic acid and ammonium acetate were purchased from Sigma Aldrich (St. Louis, MO, USA). Lipid standards SPLASH LIPIDOMIX Mass Spec (cat# 330707) were purchased from Avanti Polar Lipids Inc (Alabaster, AL, USA).

Prior to the lipid extraction samples were thawed on ice and spiked with SPLASH LIPIDOMIX deuterium-labeled internal standards (Avanti Polar Lipids Inc). The lipids were extracted from samples using a modified Bligh and Dyer total lipid extraction method (*Bligh and Dyer, 1959*). Briefly, 2 ml of the ice-cold mixture of methanol:chloroform (2:1, v/v) and 0.1% butylated hydroxy-toluene (BHT) to prevent the auto-oxidation of polyunsaturated lipid species was added to the samples. The resulting monophasic mixtures were incubated at 4 °C for 30 min. Then samples were centrifuged at 4000 rpm for 10 min and the supernatant was separated from the pellet by transferring to another tube. For the organic and aqueous phase separation 1 ml of 0.5 M sodium chloride solution was added and samples were vortexed for 10 min and centrifuged at 4000 rpm for 10 min. The organic fractions were separated and collected in glass vials. LC-MS/MS analysis was performed on Q Exactive HF mass spectrometer system coupled to the Ultimate 3000 liquid chromatography system (Thermo Fisher Scientific, USA) and reverse phase C18 column (2.1*50 mm, 17 mm particle size) (Waters, USA). The mobile phase A and B consisted of 40:60 water: acetonitrile (v/v) and 90:10 isopropanol: acetonitrile (v/v) respectively and both phases contain 10 mM ammonium acetate and 0.1% formic acid. The gradient flow parameters of mobile phase B were as follows: 0–1 min 40–45%, 1.0–1.1 min 45–50%, 1.1–5.0 min 50–55%, 5.0–5.1 min 55–70%, 5.1–8.0 min 70–99%, 8.0–8.1 min 99–40%, 8.1–9.5 min 40%. The flow rate was 0.4 mL/min during total 9.5 min run. The temperature for the autosampler and column was set for 4°C and 50°C, respectively. Data acquisition was performed in positive electrospray ionization (ESI) mode and full and data-dependent acquisition (DDA) scans data were collected. LC-MS data processing and lipid species identification were performed using LipidSearch software version 4.2 (Thermo Fisher Scientific).

## Preparing cell lysates for immunoblot

HAP1 cells were grown to 80–90% confluency in normal growth media (IMDM +10% FBS). In experiments with the OXPHOS mix antibody (RRID: AB_2085424), cells were enriched for mitochondrial membranes as detailed below. The cells were washed three times with ice-cold PBS containing 10 mM EDTA and then incubated for approximately 30 min at 4 °C. After incubation, the cells were lifted from the plate on ice and spun down at 800 rpm for 5 min at 4 °C. The resulting cell pellet was resuspended in a lysis buffer of Buffer A+0.5% Triton X-100 containing 5% protease inhibitor cocktail cOmplete (Roche, 118614500, prepared at 50 x in distilled water). Protein concentration was determined using the Bradford Assay (Bio-Rad, 5000006) and samples were diluted to 2 ug/ul. Equal volumes of these cell lysate samples were combined with Laemmli buffer (SDS and 2-mercaptoethanol) reduced and denatured and heated for 5 min at 75 °C. Samples were flash frozen and stored at –80 °C until immunoblotting. In immunoblot experiments with the OXPHOS mix antibody and blue native gel electrophoresis experiments, cell lysates were enriched for mitochondrial membranes as follows. For each experimental condition, two 150 mm dishes with HAP1 cells at 80–90% confluency were used. Trypsin was used to release the cells were and the cell pellet was washed by centrifugation with PBS.

## Immunoblots

HAP1 or SH-SY5Y cell lysates or one nanogram of recombinant APOE (Novus Biologicals, 99158) were suspended in Buffer A+0.5% Tx-100 and Laemmli sample buffer. Along with protein ladder (Bio-Rad, 161–0373), the cell lysates or recombinant protein were loaded on a 4–20% Criterion gel (Bio-Rad, 3450032) for SDS-PAGE and transferred to PVDF membrane (Millipore, IPFL00010) using the semidry transfer method. In experiments with AICAR (Sigma, A9978), cells were exposed to the drug at 0.4 mM concentration for 72 hr before lysate preparation. Membranes were incubated in a blocking solution of TBS containing 5% nonfat milk and 0.05% Triton X-100. The membranes then incubated overnight with primary antibody solutions diluted in a buffer containing PBS with 3% BSA and 0.2% sodium azide. The following day, membranes were washed three times in TBS containing 0.05% Triton X-100 (TBST) and then incubated in secondary antibody (mouse HRP or rabbit HRP) diluted in blocking solution at room temperature for 30 min. The membranes were then rinsed in TBST three times and treated with Western Lightning Plus ECL reagent (PerkinElmer, NEL105001EA). Membranes were exposed to GE Healthcare Hyperfilm ECL (28906839) for visualization.

## Blue native gel electrophoresis

Procedures were performed according to established protocols (*Díaz et al., 2009*; *Timón-Gómez et al., 2020b*). Crude mitochondria were enriched according to an established protocol (*Wieckowski et al., 2009*). Briefly, cell homogenization took place in isolation buffer (225 mM mannitol [Sigma, M9647], 75 mM sucrose [Thermo Fisher S5-500], 0.1 mM EGTA, and 30 mM Tris–HCl pH 7.4) using 20 strokes in a Potter-Elvehjem homogenizer at 6000 rpm at 4 °C. Centrifugation at 600 $g$ for 5 min was used to collect unbroken cells and nuclei and mitochondria were recovered from this supernatant by centrifugation at 7000 g for 10 min. After 1 wash of this pellet, membrane solubilization took place in 1.5 M 6-aminocaproic acid (Sigma A2504), 50 mM Bis-Tris pH 7.0 buffer with cOmplete antiprotease (Roche) and freshly prepared 4 g/g (digitonin/protein) (Miillipore, 300410). n-dodecyl-β-d-maltoside (DDM) was used at 4 g/g (DDM/protein). 3–12% gradient native gels (Invitrogen, BN2011BX10) were used to separate proteins by blue native gel electrophoresis. Molecular weight standards used were 10 mg/ml Ferritin (404 and 880 kDa, Sigma F4503) and BSA (66 and 132 kDa) (Roche, 03116956001). The first dimension was run at 150 V for 30 min at room temperature with Coomassie blue (Serva, 17524), then at 250 V for 150 min at 4 °C in cathode buffer without Coomassie. For immunoblot, proteins were transferred to PVDF membranes and probed with the indicated antibodies, as detailed above. For separation in a second dimension, the lanes from the native gels were cut and loaded on a 10% denaturing SDS-PAGE gel with a single broad lane.

## Co-Immunoprecipitation

Stable cell lines expressing FLAG-tagged SLC25A1 were prepared by transfecting human neuroblastoma SH-SY5Y cells (ATCC, CRL-2266; RRID:CVCL_0019) with ORF expression clone containing N terminally tagged FLAG-SLC25A1 (GeneCopoeia, EX-A1932-Lv1020GS) (*Gokhale et al., 2019*). These cell lines were grown in DMEM containing 10% FBS, 100 µg/ml penicillin and streptomycin, and puromycin 2 µg/ml (Invitrogen, A1113803). For co-imunoprecipitation experiments, cells were grown in 10 cm dishes. On the day of the experiment, the plated cells were placed on ice and rinsed twice with cold PBS (Corning, 21–040) containing 0.1 mM $CaCl_2$ and 1.0 mm $MgCl_2$. Lysis buffer containing 150 mm NaCl, 10 mM HEPES, 1 mM EGTA, and 0.1 mM $MgCl_2$, pH 7.4 (Buffer A) with 0.5% Triton X-100 and Complete anti-protease (Roche, 11245200) was added to the plates and cells were scraped and put into Eppendorf tubes. Cell lysates were incubated on ice for 30 min and centrifuged at 16,100×$g$ for 10 min. Bradford Assay (Bio-Rad, 5000006) was used to determine protein concentration of the recovered clarified supernatant. A total of 500 µg of the soluble protein extract was incubated with 30 µl Dynal magnetic beads (Invitrogen, 110.31) coated with 1 µg of mouse monoclonal FLAG antibody (RRID: AB_259529). The beads combined with the cell lysates were incubated on an end-to-end rotator for 2 hr at 4 °C. In some cases, as controls, beads were incubated with the lysis buffer, without any antibodies or the immunoprecipitation was outcompeted with the 3XFLAG peptide (340 µm; Sigma, F4799). Beads were then washed six times with Buffer A with 0.1% Triton X-100 followed by elution with Laemmli buffer. Samples were then analyzed by SDS-PAGE, immunoblot using polyclonal antibodies against FLAG (RRID: AB_67407) and NDUFS3 (AB_2151109).

## NanoString mRNA quantification

HAP1 cells or SH-SY5Y cells were grown to confluency in a 10 cm sterile dish in normal growth media. SH-SY5Y SLC31A1 knockout cells were treated with vehicle (cell culture grade water) or BCS (Sigma, B1125) at 200 µm for 24 hr. Three replicate plates were used for each experimental condition in experiments with HAP1 or SH-SY5Y cells. Astrocytes were seeded at a density of 15,000 cells/cm$^2$ on six-well tissue culture dishes and grown until they reached 80% confluency. At least 500,000 cells per replicate were used. Astrocytes were treated either with a DMSO vehicle control or 80 nM Antimycin A (Sigma, A8674) for 48 hr, with media being replaced every 24 hr. For preparation of samples, all cells were washed twice in ice-cold PBS containing 0.1 mM $CaCl_2$ and 1.0 mM $MgCl_2$. 1 ml of Trizol was added to the samples and the Trizol mixture was flash frozen and stored at –80 °C for a few weeks until RNA Extraction and NanoString processing was completed by the Emory Integrated Genomics Core. The Core assessed RNA quality before proceeding with the NanoString protocol. The NanoString Neuroinflammation gene panel kit (115000230) or Metabolic Pathways Panel (115000367) was used for mRNA quantification. mRNA counts were normalized by the expression of the housekeeping gene TADA2B. Data were analyzed using Qlucore Omics Explorer Version 3.7 (24) as indicated below.

## Preparation of cell pellets for proteomics and RNAseq

Wild type or gene-edited HAP1 cells (SLC25A1 and SLC25A4) were grown in IMDM supplemented with 10% FBS and 100 µg/ml penicillin and streptomycin at 37 °C in a 10% $CO_2$ incubator. Cells were grown on 10 cm tissue culture dishes to 85% confluency. Cells were placed on ice, washed 3 times with ice cold PBS (Corning, 21–040), and then incubated with PBS with 10 mM EDTA for 30 min at 4 °C. Mechanical agitation with a 10 ml pipette was used to remove cells from plates. The collected cells were then centrifuged at 800x$g$ for 5 min at 4 °C. Following centrifugation, the supernatant was removed and the pellet was washed with ice cold PBS. The resuspended pellet was then spun at 16,100 x $g$ for 5 min. The supernatant was aspirated away and the remaining pellet was flash frozen on dry ice for at least 5 min and stored at –80 °C until further use. For proteomic and transcriptomic analysis, the experiment was conducted at least in triplicate.

## TMT mass spectrometry for proteomics

Cell pellets were lysed in 200 µL of urea lysis buffer (8 M urea, 100 mM NaH2PO4, pH 8.5), supplemented with 2 µL (100 x stock) HALT protease and phosphatase inhibitor cocktail (Pierce). Lysates were then subjected to hree rounds of probe sonication. Each round consisted of 5 s of activation at 30% amplitude and 15 of seconds of rest on ice. Protein concentration was determined by bicinchoninic acid (BCA) analysis and 100 µg of each lysate was aliquoted and volumes were equilibrated with additional lysis buffer. Aliquots were diluted with 50 mM and was treated with 1 mM DTT and 5 mM IAA in sequential steps. Both steps were performed in room temperature with end-to-end rotation for 30 min. The alkylation step with IAA was performed in the dark. Lysyl endopeptidase (Wako) was added at a 1:50 (w/w) enzyme to protein ratio and the samples were digested for overnight. Samples were then diluted with 50 mM triethylammonium bicarbonate (TEAB) to a urea concentration of 1 M. Trypsin (Promega) was added at a 1:50 (w/w) enzyme to protein ratio and digestion proceeded overnight. Resulting peptides were desalted with a Sep-Pak C18 column (Waters). An aliquot equivalent to 20 µg of total protein was taken out of each sample and combined to obtain a global internal standard (GIS) use later for TMT labeling. All samples (16 individual and 4 GIS) were then dried under vacuum.

   TMT labeling was performed according to the manufacturer's protocol. Briefly (*Ping et al., 2018*), the reagents were allowed to equilibrate to room temperature. Dried peptide samples (90 µg each) were resuspended in 100 µl of 100 mm TEAB buffer (supplied with the kit). Anhydrous acetonitrile (41 µl) was added to each labeling reagent tube and the peptide solutions were transferred into their respective channel tubes. The reaction was incubated for 1 hr and quenched for 15 min afterward with 8 µl of 5% hydroxylamine. All samples were combined and dried down. Peptides were resuspended in 100 µl of 90% acetonitrile and 0.01% acetic acid. The entire sample was loaded onto an offline electrostatic repulsion–hydrophilic interaction chromatography fractionation HPLC system and 96 fractions were collected. The fractions were combined into 24 fractions and dried down. Dried peptide fractions were resuspended in 100 µl of peptide loading buffer (0.1% formic acid, 0.03% trifluoroacetic acid, 1% acetonitrile). Peptide mixtures (2 µl) were separated on a self-packed C18 (1.9 µm Dr. Maisch, Germany) fused silica column (25 cm ×75 µm internal diameter; New Objective) by a Easy-nLC

1200 and monitored on a Fusion Lumos mass spectrometer (ThermoFisher Scientific). Elution was performed over a 140 min gradient at a rate of 350 nl/min with buffer B ranging from 3% to 90% (buffer A: 0.1% formic acid in water, buffer B: 0.1% formic in 80% acetonitrile). The mass spectrometer cycle was programmed to collect at the top speed for 3 s cycles. The MS scans (375–1500 m/z range, 400,000 AGC, 50ms maximum ion time) were collected at a resolution of 120,000 at m/z 200 in profile mode. HCD MS/MS spectra (1.2 m/z isolation width, 36% collision energy, 50,000 AGC target, 86ms maximum ion time) were collected in the Orbitrap at a resolution of 50000. Dynamic exclusion was set to exclude previous sequenced precursor ions for 15 s within a 10 ppm window. Precursor ions with +1 and+8 or higher charge states were excluded from sequencing.

MS/MS spectra were searched against human database from Uniprot (downloaded on 04/2015) with Proteome Discoverer 2.1.1.21 (ThermoFisher Scientific). Methionine oxidation (+15.9949 Da), asparagine, and glutamine deamidation (+0.9840 Da) and protein N-terminal acetylation (+42.0106 Da) were variable modifications (up to 3 allowed per peptide); static modifications included cysteine carbamidomethyl (+57.0215 Da), peptide n terminus TMT (+229.16293 Da), and lysine TMT (+229.16293 Da). Only fully tryptic peptides were considered with up to two miscleavages in the database search. A precursor mass tolerance of ±20 ppm and a fragment mass tolerance of 0.6 Da were applied. Spectra matches were filtered by Percolator to a peptide-spectrum matches false discovery rate of <1%. Only razor and unique peptides were used for abundance calculations. Ratio of sample over the GIS of normalized channel abundances were used for comparison across all samples. Data files have been uploaded to ProteomeExchange.

Animal husbandry and euthanasia was carried out as approved by the Emory University Institutional Animal Care and Use Committees. For mouse brain proteomic studies, we performed TMT mass spectrometry on whole cortical brain homogenates from 43 WT and 43 5xFAD mice (MMRC#034848-JAX) and sampled age groups from 1.8 mo to 14.4 months. All groups contained equal numbers of males and females, with n=4 males and 4 females per age group. The methods used for these studies have been previously published (*Johnson et al., 2022*). Briefly, brain tissues were homogenized using a bullet blender along with sonication in 8 M Urea lysis buffer containing HALT protease and phosphatase inhibitor (ThermoFisher). Proteins were reduced, alkylated and then digested with Lysyl endopeptidase and Trypsin, followed by peptide cleanup. TMT (16-plex kit) peptide labeling was performed as per manufacturer's instructions, with inclusion of one global internal standard (GIS) in each batch. All samples in a given batch were randomized across six TMT batches, while maintaining nearly-equal representation of age, sex, and genotype across all six batches. A complete description of the TMT mass spectrometry study, including methods for sample preparation, mass spectrometry methodology and data processing, are available online (https://www.synapse.org/#!Synapse:syn27023828) and a comprehensive analysis of these data will be published separately. Mass spectrometry raw data were processed in Proteome Discover (Ver 2.1) and then searched against Uniprot mouse database (version 2020), and then processed downstream as described for human brain TMT mass spectrometry studies above. Batch effect was adjusted using bootstrap regression which modeled genotype, age, sex and batch, but covariance with batch only was removed (*Wingo et al., 2020*). From the 8535 proteins identified in this mouse brain proteome, we analyzed data related to APOE and 914 proteins that were also found in the Mitocarta database (*Rath et al., 2021*).

## ICP mass spectrometry

Samples were prepared for ICP-MS, processed, and analyzed as described in *Lane et al., 2022*. In brief, media was changed the day before samples were collected. Cells were detached using trypsin, centrifuged at 130 *g* (800 rpm) for 5 min at 4 °C, and resuspended in ice cold PBS. The cell suspension was distributed into microcentrifuge tubes and centrifuged at 210 *g* (1500 rpm) for 5 min at 4 °C. The supernatant was removed and the cells were flash frozen in dry ice and stored at –80 °C. twemn24 hours before running samples on the ICP-MS instrument, samples were dissolved in 20 µL of 70% nitric acid and incubated at 95 C for 10 min. Immediately prior to running samples, they were diluted 1:40 in 2% nitric acid in a deep 96-well plate (to a total volume of 800 µL). Quantitation of trace elements was performed using a Thermo iCAP-TQ series ICP-MS operated in oxygen reaction mode with detection of elements of interest with the third quadrupole. In experiments with BCS, cells were treated for 24 hr using 200 µM concentration. Detailed operating procedures and acquisition parameters are described in *Lane et al., 2022*.

## RNAseq and data analysis

RNA isolation, library construction, and sequencing were performed by the Beijing Genomics Institute. Total RNA concentration was measured with Agilent 2100 Bio analyzer (Agilent RNA 6000 Nano Kit), QC metrics: RIN values, 28 S/18 S ratio and fragment length distribution. mRNA purification was achieved by poly-T oligo immobilized to magnetic beads. the mRNA was fragmented using divalent cations plus elevated temperature. RNA fragments were copied into first strand cDNA by reverse transcriptase and random primers. Second strand cDNA synthesis was performed with DNA Polymerase I and RNase H. A single 'A' base and subsequent ligation of adapter was done on cDNA fragments. Products were purified and enriched with PCR amplification. The PCR yield was quantified using a Qubit and samples were pooled together to make a single strand DNA circle (ssDNA circle) producing the final library. DNA nanoballs were generated with the ssDNA circle by rolling circle replication to enlarge fluorescent signals during the sequencing process. DNA nanoballs were loaded into the patterned nanoarrays and pair-end reads of 100 bp were read through on the DNBseq platform. The DNBseq platform combined the DNA nanoball-based nanoarrays and stepwise sequencing using Combinational Probe-Anchor Synthesis Sequencing Method. On average, we produced ~4.41 Gb bases per sample. The average mapping ratio with reference genome was 92.56%, the average mapping ratio with gene was 72.22%; 17,240 genes were identified. 16,875 novel transcripts were identified. Read quality metrics: 4.56% of the total amount of reads contained more than 5% unknown N base; 5.92% of the total amount of reads which contained adaptors; 1.38% of the total reads were considered low quality meaning more than 20% of bases in the total read have quality score lower than 15. 88.14% of total amount of reads were considered clean reads and used for further analysis.

Sequencing reads were uploaded to the Galaxy web platform. We used the public server usegalaxy. eu to analyze the data (*Afgan et al., 2018*). FastQC was used to remove samples of suboptimal quality (*Andrews, 2010*). All mapping was performed using Galaxy server (v. 19.09) running Hisat2 (Galaxy Version 2.1.0+galaxy5), HTseq-Count (Galaxy Version 1.6.2), and DeSeq2 (Galaxy Version 2.11.40.2) (*Anders et al., 2015*; *Kim et al., 2015*; *Love et al., 2014*). The Genome Reference Consortium build of the reference sequence (GRCh38/hg38) and the GTF files (NCBI) were used and can be acquired from iGenome (Illumina). Hisat2 was run with the following parameters: paired-end, unstranded, default settings were used except for a GTF file was used for transcript assembly. Alignments were visualized using IGV viewer (IGV-Web app version 1.7.0, igv.js version 2.10.5) with Ensembl v90 annotation file and Human (GRCh38/hg38) genome (*Jin et al., 2020*; *Robinson et al., 2011*).

Aligned SAM/BAM files were processed using HTseq-count (Default settings except used GRCh38 GTF file and output for DESeq2 and gene length file). HTseq-count output files and raw read files are publicly available (GEO with accession GSE201889). The HTseq-count compiled file is GSE201889_ RawHTseqCounts_ALL. Gene counts were normalized using DESeq2 (*Love et al., 2014*) followed by a regularized log transformation. Differential Expression was determined using DESeq2 with the following settings: Factors were cell type, pairwise comparisons between mutant cell lines versus control line was done, output all normalized tables, size estimation was the standard median ratio, fit type was parametric, outliers were filtered using a Cook's distance cutoff.

## Human cognitive trajectory and proteome correlations

The Banner Sun Health Research Institute participants (Banner) project is a longitudinal clinicopathological study of normal aging, Alzheimer's disease (AD), and Parkinson's disease (PD). Most subjects were enrolled as cognitively normal volunteers from the retirement communities of the greater Phoenix, Arizona, USA (*Beach et al., 2015*). Recruitment efforts were also directed at subjects with AD and PD from the community and neurologists' offices. Subjects received standardized general medical, neurological, and neuropsychological tests annually during life and more than 90% received full pathological examinations after death (*Beach et al., 2015*).

Person-specific cognitive trajectory was estimated using a linear mixed model. In this model, the annual MMSE score was the longitudinal outcome, follow-up year as the independent variable, sex as the covariate, and with random intercept and random slope per subject using the lme4 R package (version 1.1–19).

Proteomic quantification from the dorsolateral prefrontal cortex of post-mortem brain tissue using mass spectrometry was described in detail here (*Wingo et al., 2019*). Raw data were analyzed using MaxQuant v1.5.2.8 with Thermo Foundation 2.0 for RAW file reading capability. Co-fragmented

peptide search was enabled to deconvolute multiplex spectra. The false discovery rate (FDR) for peptide spectral matches, proteins, and site decoy fractions were set to 1%. Protein quantification was estimated by label free quantification (LFQ) algorithm by MaxQuant and only considered razor plus unique peptides for each protein isoform. After label-free quantification, 3710 proteins were detected. Only proteins quantified in at least 90% of the samples were included in the analysis. $\log_2$ transformation of the proteomic profiles was performed. Effect of protein sequencing batch was removed using Combat (*Johnson et al., 2007*). Effects of sex, age at death, and post-mortem interval (PMI) were removed using bootstrap regression. Association between cognitive trajectory and data-sets listed in *Figure 7E* was examined using linear regression, in which cognitive trajectory was the outcome and the first principal component of the dataset was the predictor.

## Composite protein abundance calculations

Composite protein abundances of respiratory complexes and SLC25An paralogs in the 5xFAD mouse were calculated as the first principal component of individual protein abundances across 86 samples in the data set, using the WGCNA moduleEigengenes function with the standard imputation method allowed, similar to the calculation of synthetic eigengenes as described in *Johnson et al., 2022*; *Wingo et al., 2019*. Given that protein abundances were $\log_2$-transformed, changes in composite values represent $\log_2$ fold change.

## Bioinformatic analyses and statistical analyses

Data from proteomes, RNAseq, NanoString, and lipidomics were processed with Qlucore Omics Explorer Version 3.6 (33) normalizing log2 data to a mean of 0 and a variance of 1. Qlucore Omics was used to generate volcano plots, Euclidean hierarchical clustering, PCI and 2D- tSNE. 2D-tSNE was calculated with a perplexity of 5. All other statistical analyses were performed with Prism v9.2.0 (283) using two tailed statistics and Alpha of 0.05 using test specified in each figure legend. No outlier iden-tification and exclusion were applied. Asterisks denoting significance followed Prism specifications. Estimation statistics in *Figure 5A* was performed as described (*Ho et al., 2019*).

Gene ontology analyses were carried out with Cluego. ClueGo v2.58 run on Cytoscape v3.8.2 (*Bindea et al., 2009*; *Shannon et al., 2003*). ClueGo was run querying GO BP, REACTOME, and KEGG considering all evidence, Medium Level of Network Specificity, and selecting pathways with a Bonferroni corrected *P*-value <0.001. ClueGo was run with Go Term Fusion. Analysis of Nanostring Ontologies was performed with Metascape (*Zhou et al., 2019*). All statistically enriched terms based on the default choices under Express Analysis, cumulative hypergeometric p-values and enrichment factors were calculated and used for filtering. Remaining significant terms were then hierarchically clustered into a tree based on Kappa-statistical similarities among their gene memberships. Kappa score of 0.3 was applied as the threshold to cast the tree into term clusters (*Zhou et al., 2019*).

The interactome of NDUFS3 and SLC25A1 was generated from the proximity interaction datasets by *Antonicka et al., 2020*. Data were imported into Cytoscape v3.8.2.

Enrichment (Representation Factor) and exact hypergeometric probability of gene set enrichments with hits from the Human Secretome (*Uhlén et al., 2019*) and Human Mitocarta 3.0 (*Rath et al., 2021*) in *Figures 1N and 5D* were calculated with the engine nemates.org using the 20,577 gene count of the Uniprot Human Proteome Reference.

## Acknowledgements

This work was supported by grants from the NIH 1RF1AG060285 and Developmental Project through the Goizueta Alzheimer's Disease Research Center (ADRC - P30AG066511) to VF, U01AG061357 to NTS, F31AG067623 and 5T32NS007480 to MEW, R01NS11430 and RF1AG071587 to SRangaraju, F32AG064862 to SRayaprolu, and 1F31NS127419 to ARL. ARL is supported by an ARCS Founda-tion Award and a Robert W Woodruff Fellowship. This study was supported in part by the Emory Integrated Genomics, Immunoassay, and the Integrated Metabolomics and Lipidomics Cores, which are subsidized by the Emory University School of Medicine. Additional support was provided by the Georgia Clinical & Translational Science Alliance of the National Institutes of Health under Award Number UL1TR002378. The content is solely the responsibility of the authors and does not necessarily reflect the official views of the National Institutes of Health. Elemental analyses were performed by Anne M Roberts. VF is grateful for mitochondria provided by Maria Olga Gonzalez.

## Additional information

### Funding

| Funder | Grant reference number | Author |
| --- | --- | --- |
| National Institute on Aging | 1RF1AG060285 | Victor Faundez |
| Alzheimer's Disease Research Center, Emory University | P30AG066511 | Victor Faundez |
| National Institute on Aging | U01AG061357 | Nicholas T Seyfried |
| National Institute of Neurological Disorders and Stroke | F31AG067623 | Meghan E Wynne |
| National Institute of Neurological Disorders and Stroke | 5T32NS007480 | Meghan E Wynne |
| National Institute of Neurological Disorders and Stroke | R01NS11430 | Srikant Rangaraju |
| National Institute on Aging | RF1AG071587 | Srikant Rangaraju |
| National Institute on Aging | F32AG064862 | Sruti Rayaprolu |
| National Institute of Neurological Disorders and Stroke | 1F31NS127419 | Alicia R Lane |
| ARCS Foundation | | Alicia R Lane |
| Emory University | Robert W Woodruff Fellowship | Alicia R Lane |

The funders had no role in study design, data collection and interpretation, or the decision to submit the work for publication.

### Author contributions

Meghan E Wynne, Conceptualization, Data curation, Formal analysis, Investigation, Writing - original draft, Writing – review and editing; Oluwaseun Ogunbona, Conceptualization, Formal analysis, Investigation, Methodology, Writing – review and editing; Alicia R Lane, Avanti Gokhale, Stephanie A Zlatic, Chongchong Xu, Sruti Rayaprolu, Investigation, Writing – review and editing; Zhexing Wen, Eric A Ortlund, Nicholas T Seyfried, Supervision, Investigation, Writing – review and editing; Duc M Duong, Anna Ivanova, Formal analysis, Investigation, Writing – review and editing; Eric B Dammer, Amanda Crocker, Data curation, Formal analysis, Investigation, Writing – review and editing; Blaine R Roberts, Data curation, Formal analysis, Investigation, Methodology, Writing – review and editing; Vinit Shanbhag, Nanami Senoo, Selvaraju Kandasamy, Resources, Writing – review and editing; Michael Petris, Resources, Supervision, Writing – review and editing; Steven Michael Claypool, Antoni Barrientos, Resources, Supervision, Validation, Writing – review and editing; Aliza Wingo, Thomas S Wingo, Conceptualization, Data curation, Formal analysis, Validation, Visualization, Writing – review and editing; Srikant Rangaraju, Data curation, Formal analysis, Supervision, Investigation, Writing – review and editing; Allan I Levey, Resources, Funding acquisition, Writing – review and editing; Erica Werner, Conceptualization, Supervision, Validation, Investigation, Methodology, Project administration, Writing – review and editing; Victor Faundez, Conceptualization, Data curation, Formal analysis, Supervision, Funding acquisition, Methodology, Writing - original draft, Project administration, Writing – review and editing

### Author ORCIDs

Alicia R Lane ![ORCID] http://orcid.org/0000-0002-6404-7559
Anna Ivanova ![ORCID] http://orcid.org/0000-0002-6221-6240
Steven Michael Claypool ![ORCID] http://orcid.org/0000-0001-5316-1623
Antoni Barrientos ![ORCID] http://orcid.org/0000-0001-9018-3231

Thomas S Wingo ⓘ http://orcid.org/0000-0002-7679-6282
Allan I Levey ⓘ http://orcid.org/0000-0002-3153-502X
Erica Werner ⓘ http://orcid.org/0000-0002-8183-1601
Victor Faundez ⓘ http://orcid.org/0000-0002-2114-5271

**Decision letter and Author response**
Decision letter https://doi.org/10.7554/eLife.85779.sa1
Author response https://doi.org/10.7554/eLife.85779.sa2

## Additional files

### Supplementary files
• Supplementary file 1. Raw Data for Proteomics, Transcriptomics and Gene Ontologies.
• MDAR checklist

### Data availability
The mass spectrometry proteomics data have been deposited to the ProteomeXchange Consortium via the PRIDE (*Deutsch et al., 2020*) partner repository with dataset identifiers: PXD038974 and PXD017501. RNAseq data were deposited in GEO with accession GSE201889.

The following datasets were generated:

| Author(s) | Year | Dataset title | Dataset URL | Database and Identifier |
|---|---|---|---|---|
| Crocker A, Faundez V | 2022 | APOE Expression and Secretion are Modulated by Copper-Dependent and -Independent Mitochondrial Dysfunction | https://www.ncbi.nlm.nih.gov/geo/query/acc.cgi?acc=GSE201889 | NCBI Gene Expression Omnibus, GSE201889 |
| Faundez V | 2023 | Faundez Comparison of 2 Cell lines (Hek and Hep) | https://proteomecentral.proteomexchange.org/cgi/GetDataset?ID=PXD038974 | ProteomeXchange, PXD038974 |
| Faundez V | 2023 | Faundez TMT 12 samples - 2 batches with 24 fractions each | https://proteomecentral.proteomexchange.org/cgi/GetDataset?ID=PXD017501 | ProteomeXchange, PXD017501 |

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
