## [Editor Report]

This study presents compelling evidence that ApoE is upregulated in various models of mitochondrial respiratory chain dysfunction. The work is of fundamental interest for those studying neurodegeneration and the role of ApoE in Alzheimer's disease; future work will be needed to reveal the molecular basis of this dramatic phenotype.

---

## [Decision Letter]

**Decision letter after peer review:**

[Editors’ note: the authors submitted for reconsideration following the decision after peer review. What follows is the decision letter after the first round of review.]

Thank you for submitting the paper "APOE Expression and Secretion are Modulated by Copper-Dependent and -Independent Mitochondrial Dysfunction" for consideration by *eLife*. Your article has been reviewed by 3 peer reviewers, one of whom is a member of our Board of Reviewing Editors, and the evaluation has been overseen by a Senior Editor. The reviewers have opted to remain anonymous.

We are sorry to say that, after consultation with the reviewers, we have decided that this manuscript will not be considered further for publication by *eLife* at this stage.

As you will see from the detailed comments below, the reviewers have raised several concerns related to the depth of mechanistic understanding and the largely correlative nature of the data provided. However, the reviewers would be pleased to consider a new submission that addresses their major concerns discussed below.

*Reviewer #1 (Recommendations for the authors):*

Wynne et al. asked how the proteome and transcriptome of HAP1 cells are affected by the knockout of mitochondrial citrate carrier SLC25A1 and ADP/ATP carrier SLC25A4. Next the authors searched for proteins/transcripts that were similarly regulated at both protein and transcript levels in both tested knockout cell lines and found only 27 of such proteins, among which the one highly differentially regulated was APOE – the protein whose certain polymorphic alleles are attributed as one of the major risk factors for developing Alzheimer's Disease. Next, the authors explored how other conditions affecting mitochondria impact the cytoplasm and serum levels of APOE. The authors observed that cells with a knockout of SLC25A1 or SLC25A4 share a phenotype of decreased abundance of complex I and complex III as well as defects in the assembly of the functional complexes. Therefore they reasoned that affecting complex I and III alone by knocking out crucial complex I assembly factors and treating cells with complex III inhibitor antimycin would affect APOE expression/secretion. Indeed both genetic and pharmacological manipulation on complex I and III led to upregulation of APOE expression/secretion. Similarly, alternations in APOE levels in cell lysates and medium were also observed after affecting plasma membrane and mitochondrial copper transporters and knocking down the COX17 complex IV assembly factor. Next, some of the above experiments were performed on astrocytes and neurons differentiated from induced pluripotent stem cells. In line with the previous literature data, the effects on APOE upregulation were pronounced in astrocytes but not neurons. Finally, the authors present data from a well-established mouse model of AD (5xFAD mice) showing that APOE overexpression positively correlated with aging and a decrease in expression of complex I, III, and IV in 5xFAD mice but not in controls. The study concludes with statistical analysis aiming at testing the association between the expression of mRNAs and proteins differentially regulated in SLC25A1 mutant in the collection of post-mortem brain proteomes and the cognitive performance tests of the brain donors. The authors conclude that expression of proteins upregulated in the studied mutant was positively correlated with cognitive decline, while increased expression of protein products of mRNAs downregulated in the mutant was associated with the protection from cognitive decline.

General strengths of the study

1. The observation of an increase in APOE expression/secretion being downstream of conditions affecting mitochondria is of high relevance as current studies are focused on the effects of APOE on mitochondria. Hence, the study by Wynne et al. provides the foundation for the understanding of intercellular non-cell-autonomous signaling responsible for APOE-mediated mitochondria metabolism alternations in response to mitochondrial stress.

2. Results showing the increase in APOE expression/secretion are performed in different cellular models including those with implications for AD such as astrocytes.

General weaknesses of the study

1. The manuscript in its current form lacks proper presentation of the data on APOE overexpression that would allow the reader to easily compare the magnitude of the effect in different conditions and cell lines studied. It is not discussed and presented in the figure what the differences in basal/induced APOE in different models and conditions studied are. The manuscript is lacking the bar plots displaying a comparison of the normalized to control APOE expression values for different conditions and basal non-normalized APOE levels in different cell lines studied.

2. Although the authors excluded particular mechanistic explanations for the observed increase in APOE expression/secretion they did not manage to provide a comprehensive mechanistic explanation of the observed phenomenon. In that context as the authors study citrate carrier, it would be interesting to check citrate levels and whether manipulating citrate levels alone would affect the expression of APOE, and whether baseline citrate levels in studied models correspond to the magnitude of the observed APOE overexpression.

3. The manuscript lacks a clear description of the metabolic context for the studied conditions – what is the impact of SLC25A1 and SLC25A4 on metabolic activity-viability in response to stress, ATP/ADP, glycolysis, citrate levels, copper levels, etc. As there is a certain degree of functional redundancy within the SLC25 family it would be interesting to highlight and discuss SLC25A1/A4 paralogs with redundant functions. Double knockout experiments and/or combined siRNA screen may help to answer the question of whether APOE upregulation in mutants is related to canonical channeling function or some other "moonlighting" signaling mechanism specific to the particular paralogs of transporters. It is not discussed why in triple ATP/ADP carrier knockout the upregulation of APOE was only 50 % while in single SLC25A4 knockout more than 4 fold – is it due to a switch to glycolytic metabolism, the addition of pyruvate, uridine to the medium, or something else?

Palmieri, F. & Monne, M. Discoveries, metabolic roles and diseases of mitochondrial carriers: A review. Biochim Biophys. Acta 1863, 2362-2378 (2016).

Shi, X., Reinstadler, B., Shah, H. et al. Combinatorial GxGxE CRISPR screen identifies SLC25A39 in mitochondrial glutathione transport linking iron homeostasis to OXPHOS. Nat Commun 13, 2483 (2022). https://doi.org/10.1038/s41467-022-30126-9

4. The authors attribute APOE overexpression to copper ion levels and assembly of complex IV however experiments on cells with a defect in copper transporters lack measurements of copper; the functional redundancy of copper transporters is not discussed. Are those having functional paralogs?

5. In the present form of the manuscript it is not clear how authors view the mode of APOE overexpression – apart from experiments showing an increase in APOE mRNA abundance there are no experiments explaining if it is due to an increase in transcription and/or mRNA stability. There are also no experiments and discussion on APOE regulation on the level of translation/degradation.

*Reviewer #2 (Recommendations for the authors):*

Overall, this' work is innovative for identifying dysfunctional mitochondria that can increase APOE in diverse cell types and an Alzheimer's disease mouse model. This work provides insights into how mitochondria dysfunction may influence brain and cognition function partially through modulation of APOE. While exciting, some key questions are unanswered. First, what specifically triggers increased APOE expression in the setting of various mitochondrial insults is not clear. Additionally, it is not clear how mitochondrial dysfunction is signaled to increase the expression of APOE from the nuclear genome, although some mechanisms, such as AMPK and the integrated stress response are ruled out. Finally, it is not clear what function up-regulating APOE in response to mitochondrial dysfunction serves and whether this response is adaptive or maladaptive. This study nonetheless establishes a novel link between mitochondrial dysfunction and APOE in the pathogenesis of Alzheimer's disease.

While innovative and the conclusions are generally well supported and described, the manuscript would be enhanced by clarifying some unclear statements and a few additional experiments to support the authors' major findings.

1. The authors demonstrate that several mitochondrial perturbations lead to APOE increases, but a unifying mitochondrial deficit among these is not identified as the trigger of APOE increase. This should be further explored with further experiments to manipulate electron transport and/or mitochondrial membrane potential. For instance, hypoxia increases APOE, which might point to decreased electron transport as a common trigger? Can decreased mitochondrial membrane potential cause APOE increase. The authors could test whether oligomycin + antimycin has a stronger effect than antimycin treatment alone, as the addition of the F1FO-ATP synthase inhibitor will keep it from running in reverse to maintain MMP in the setting of decreased electron transport. Similarly, the authors could test whether mitochondrial uncouplers such as FCCP or BAM15 similarly increase APOE expression. These dissipate the MMP but increase electron transport.

2. Relatedly, it is suggested that the mechanistic link between SLC25A1 and SLC25A4 and APOE increase is through destabilization of respiratory complexes, in particular CIII, as well as higher-order complexes containing complexes I, III, and/or IV. To support this model in Figures 3C and D, the authors show changes in higher order respiratory complexes by BN-PAGE. The effect of SLC25A1 and SLC25A4 on the respiratory complexes, however, remains unclear. Depending on the detergent used different supercomplexes are stabilized (e.g., CI containing supercomplexes are better stabilized by Triton-X100 or digitonin compared to DDM). I was not able to find which detergent was used in these experiments, but it would be worth checking for supercomplexes using the milder detergents Triton-X100 or digitonin if this was not done. Also, only SLC25A1 KO is explored in this experiment. The authors should also assess SLC25A4. Finally, it is notable that in Figure 1K among the few shared differentially regulated proteins between SLC25A1 and SLC25A4 KO is COX7A2L in addition to CIII subunits. As COX7A2L is known to stabilize CIII-CIV within supercomplexes, the authors should explore whether this is part of the mechanism explaining the decreased formation of higher order complexes.

3. The physiologic significance of increased APOE in response to mitochondrial dysfunction remains unclear. In figure 1, it is established that conditioned media from WT cells promotes the growth of WT and SLC25A1 KO cells while conditioned media from SLC25A1 KO cells promotes the growth of WT but not SLC25A1 KO cells. This is interpreted in the discussion to suggest that "APOE dependent removal of toxic factors [from, e.g., SLC25A1 KO cells] may be the mechanism" (in ln 619 – 621). However, it's not clear whether this is why WT conditioned media rescues the growth of SLC25A1 KO cells or whether conditioned media from SLC25A1 KO cells promotes the growth of WT cells? Don't these results suggest that WT-conditioned media is providing something missing from SLC25A1 KO media that promotes the growth of SLC25A1 cells? Additionally, it remains very speculative that this factor is related to APOE. Does knockdown of APOE from either WT or SLC25A1 cells influence the effect of the conditioned media in this assay?

APOE4 is a major genetic risk factor for Alzheimer's disease. In their manuscript, Wynne et al. show that intracellular and secreted APOE protein increases in response to mitochondrial dysfunction, including perturbations mitochondrial complexes I and III, and factors involved in complex III biogenesis such as copper and SLC25A1 and SLC25A4. The correlation between APOE and respiratory chain subunits was further explored in 5xFAD mouse model of Alzheimer's disease, demonstrating a positive correlation between mitochondrial dysfunction in this model and APOE4 increase. Finally, they found that during aging, SLC25A1 sensitive gene expression correlates with human cognitive trajectory.

*Reviewer #3 (Recommendations for the authors):*

This study uses multiple systems approaches using cell lines deleted for two mitochondrial carriers and identifies changes in the secretome at the transcriptional and proteome level. Among these changes, increased ApoE transcription, protein level and secretion appeared particularly interesting and conserved, leading the authors to expand their study to several other conditions of mitochondrial dysfunction. The authors analyze cell lines deleted for components of the respiratory chain, for genes involved in copper trafficking, and for astrocytes treated with mitochondrial poisons, and demonstrate increased secretion of ApoE in all these conditions, albeit in very different amounts. In addition, they correlate ApoE levels in the brain of a mouse model for Alzheimer's disease with decreased mitochondrial dysfunction. Since alleles of ApoE are known risk factors for Alzheimer's disease, the results are of potential interest.

The weakness of the study is that data are entirely correlative and there is no attempt to understand the causal relationship between mitochondrial dysfunction and increased ApoE transcription and secretion.

1) The statement that SLC25A1Δ cells condition media differently than wild-type cells is not supported by the data. If this were the case, the conditioned media of SLC25A1Δ cells would affect differently also WT cells. Instead, it seems from the data that the dialyzed SLC25A1Δ conditioned medium lacks some metabolite that cannot be produced by SLC25A1Δ cells. Since this effect is only evident with the dialyzed medium it is unlikely that it can be attributed to ApoE. Moreover, to really appreciate the data of Figure 1A. graphs should show the y-axis starting from 0 and show WT and KO cells in the same graph normalized to WT. How different is the growth of SLC25A1Δ cells?

2) Levels of ApoE in the lysate and media from different cell lines and experimental conditions range from 80ng/microg to 3pg/mg, depending on the cell line examined. The authors should comment on this high variability, especially since they try to make the case that this ApoE secretion is relevant across many models of mitochondrial dysfunction. Is this physiologically relevant in all cases?

3) In Figure 3C, the Coomassie stained gels and the blotted ones appear very different. Also, the molecular weight markers indicate different runs, so it is unclear why there are these big differences in the western blots, but much less is apparent in the Coomassie and in panel D. The positions of the complexes should be indicated also on the Coomassie gel. I could not find an indication in the methods about the amount and type of detergent used. Are these performed using DDM or digitonin? How many times have these experiments been repeated?

4) In several parts of the manuscript, the authors argue that the changes in the secretome were equivalent to changes in the mitochondrial proteome as evidence for the relevance of the observed changes in the secretome. This argument seems rather weak since these omics changes can reflect a direct effect of a gene loss as well as very downstream events. Ultimately, what would be really important here is to demonstrate how this increased secretion of ApoE occurs and dissect at least in part how mitochondrial dysfunction affects ApoE expression transcriptionally.

5) The last part of the manuscript correlates changes in the transcriptome, proteome, and interactome of SLC25A1Δ cells with the cognitive progression of AD patients. As the authors write, it is already known that AD patients' cognitive decline negatively correlates with mitochondrial function. Although catching, these data remain very correlative, and also not very surprising.

6) In Figure S1, why is AICAR not increasing pAMPK in WT cells?

7) The title emphasizes copper-dependent and independent mitochondrial dysfunction. This puts an emphasis on copper metabolism that is totally unnecessary. The rationale all over the paper for the choice of models to use is not always very clear.

8) The scheme in Figure4B should be redrawn to show Cox17 in the IMS.

[Editors’ note: further revisions were suggested prior to acceptance, as described below.]

Thank you for resubmitting your work entitled "APOE Expression and Secretion are Modulated by Mitochondrial Dysfunction" for further consideration by *eLife*. Your revised article has been overseen by Suzanne Pfeffer (Senior Editor) and a Reviewing Editor.

The manuscript has been improved but there are some remaining issues that need to be addressed, as outlined below:

The authors should aim to specify the mode of APOE abundance regulation (the original comment no 5 of Reviewer 1) with some basic experiments and a short discussion. For example, for the HAP1 SLC25A1 mutant, the increase in mRNA is 3-fold but the increase in APOE cell lysate concentration is almost 8-fold. These minor revisions may include experiments with protein degradation inhibitors to evaluate a potential regulation of the level of degradation.

The manuscript would benefit from streamlining and focusing. The authors could not dissect a mechanism, but could convincingly show that ApoE upregulation occurs across several models with different respiratory dysfunction. This should be at the center of the manuscript.

The sentence in the Abstract: "This upregulation of secretory proteins was of similar extent as modifications to the mitochondrial annotated proteome" is confusing and prone to misunderstanding and should be rephrased.

The claim that changes in cholesterol synthesis are not involved is too strong. The authors have not formally proved that this is not the case in δ SLC25A1 cells. It is possible that it is beneficial for cells with respiratory defects to increase APOE secretion, but different mutants achieve this with different mechanisms.

---

## [Author Response]

[Editors’ note: the authors resubmitted a revised version of the paper for consideration. What follows is the authors’ response to the first round of review.]

Reviewer #1 (Recommendations for the authors):Wynne et al. asked how the proteome and transcriptome of HAP1 cells are affected by the knockout of mitochondrial citrate carrier SLC25A1 and ADP/ATP carrier SLC25A4. Next the authors searched for proteins/transcripts that were similarly regulated at both protein and transcript levels in both tested knockout cell lines and found only 27 of such proteins, among which the one highly differentially regulated was APOE – the protein whose certain polymorphic alleles are attributed as one of the major risk factors for developing Alzheimer's Disease. Next, the authors explored how other conditions affecting mitochondria impact the cytoplasm and serum levels of APOE. The authors observed that cells with a knockout of SLC25A1 or SLC25A4 share a phenotype of decreased abundance of complex I and complex III as well as defects in the assembly of the functional complexes. Therefore they reasoned that affecting complex I and III alone by knocking out crucial complex I assembly factors and treating cells with complex III inhibitor antimycin would affect APOE expression/secretion. Indeed both genetic and pharmacological manipulation on complex I and III led to upregulation of APOE expression/secretion. Similarly, alternations in APOE levels in cell lysates and medium were also observed after affecting plasma membrane and mitochondrial copper transporters and knocking down the COX17 complex IV assembly factor. Next, some of the above experiments were performed on astrocytes and neurons differentiated from induced pluripotent stem cells. In line with the previous literature data, the effects on APOE upregulation were pronounced in astrocytes but not neurons. Finally, the authors present data from a well-established mouse model of AD (5xFAD mice) showing that APOE overexpression positively correlated with aging and a decrease in expression of complex I, III, and IV in 5xFAD mice but not in controls. The study concludes with statistical analysis aiming at testing the association between the expression of mRNAs and proteins differentially regulated in SLC25A1 mutant in the collection of post-mortem brain proteomes and the cognitive performance tests of the brain donors. The authors conclude that expression of proteins upregulated in the studied mutant was positively correlated with cognitive decline, while increased expression of protein products of mRNAs downregulated in the mutant was associated with the protection from cognitive decline.General strengths of the study1. The observation of an increase in APOE expression/secretion being downstream of conditions affecting mitochondria is of high relevance as current studies are focused on the effects of APOE on mitochondria. Hence, the study by Wynne et al. provides the foundation for the understanding of intercellular non-cell-autonomous signaling responsible for APOE-mediated mitochondria metabolism alternations in response to mitochondrial stress.2. Results showing the increase in APOE expression/secretion are performed in different cellular models including those with implications for AD such as astrocytes.General weaknesses of the study1. The manuscript in its current form lacks proper presentation of the data on APOE overexpression that would allow the reader to easily compare the magnitude of the effect in different conditions and cell lines studied. It is not discussed and presented in the figure what the differences in basal/induced APOE in different models and conditions studied are. The manuscript is lacking the bar plots displaying a comparison of the normalized to control APOE expression values for different conditions and basal non-normalized APOE levels in different cell lines studied.

We thank the reviewer for this suggestion that adds clarity to the manuscript. We modified all figures as suggested by depicting fold change of APOE from WT and/or vehicle condition in every graph, allowing for easy comparison across all experiments.

2. Although the authors excluded particular mechanistic explanations for the observed increase in APOE expression/secretion they did not manage to provide a comprehensive mechanistic explanation of the observed phenomenon. In that context as the authors study citrate carrier, it would be interesting to check citrate levels and whether manipulating citrate levels alone would affect the expression of APOE, and whether baseline citrate levels in studied models correspond to the magnitude of the observed APOE overexpression.

The reviewer suggests an interesting idea. We think is unlikely that decreased cytoplasmic citrate levels in SLC25A1 cells are causally related to increased APOE levels. We tested a role of citrate in APOE levels by two approaches presented in Figure S5. First, we confirmed the presence of the plasma membrane citrate transporter SLC13A5 in HAP1 cells. We then measured the effect of citrate addition to the media of either wild type or SLC25A1-null cells on APOE levels. Citrate incubation does not change expression of APOE in either genotype. Second, we mutated ACLY or ATP citrate lyase, which is the primary enzyme that catalyzes the synthesis of cytosolic acetylCoA from cytoplasmic citrate (Zaidi et al., 2012). ACLY mutants cause accumulation of cytoplasmic citrate, which is the opposite effect of an SLC25A1 mutation (Baardman et al., 2020; Majd et al., 2018; Rensvold et al., 2022). These ACLY null cells increased APOE levels and secretion. Thus, we believe that decreased cytoplasmic citrate is an unlikely mediator of the APOE response in SLC25A1 null cells. Third, we used the Pagliarini metabolomics data curated in MITOMICS.app. This database reports two of the cell lines used in our studies, SLC25A1 and NDUFS3 KO HAP1 cells (Rensvold et al., 2022). Both mutations increase the levels of APOE in our studies but total cellular citrate levels are only affected (decreased) in SLC25A1 cells. We conclude that citrate is not a mediator of APOE increased levels in mutants that affect the respiratory chain.

3. The manuscript lacks a clear description of the metabolic context for the studied conditions – what is the impact of SLC25A1 and SLC25A4 on metabolic activity-viability in response to stress, ATP/ADP, glycolysis, citrate levels, copper levels, etc.

This is an important question that we have addressed in part by adding new copper and ECAR studies in several mutants. We previously reported the effects of SLC25A1 and SLC25A4 mutations on the ATP/ADP ratios (Gokhale et al., 2019). ATP levels are decreased only in SLC25A4 mutant HAP1 cells making unlikely a bioenergetic model for increased APOE levels in SLC25A1 mutants we have analyzed.

As suggested by the reviewer, we measured the total cell copper levels in many of the cells utilized in our studies. Copper content is modified in cells that carry mutations in copper transporters (ATP7A, SLC31A1, SLC25A3) and in HEK293 cells that are triple KO for ADP/ATP carriers. There is no universal copper phenotype across all mutants tested.

Finally, we have added the ECAR data to revised Figure 5J and Figure S4. This data shows that there is no correlation between ECAR and APOE levels (p=0.2). To stress this point, we used as a control a PARK2 mutant SH-SY5Y cell line that possesses an elevated rate of extracellular acidification but that does not increase APOE levels. Our data suggest that glycolytic adaptations in mutant cells are unlikely to contribute to the increased APOE level phenotype.

What about other aspects of intermediate metabolism? We think that to address the reviewer’s comment in full we should perform untargeted metabolomics studies in whole cells and isolated mitochondria in several of the 14 mutants we used in our studies. Such studies are akin of those performed by Sabatini and Pagliarini (Chen et al., 2016; Rensvold et al., 2022). However, the material and time resources needed are beyond our reach at this point. We believe we have built a solid case for the integrity and function of the respiratory chain as a trigger of increased APOE expression, which is a common phenotype in mutants studied here. Since the published metabolic profiles of inhibiting different complexes of the respiratory chain differ both in total cell extract and isolated mitochondria (Chen et al., 2016), we suspect that a common metabolite may be an analyte not commonly considered in targeted metabolomes. We hope the reviewer will waive a request for a full metabolic characterization of cells.

As there is a certain degree of functional redundancy within the SLC25 family it would be interesting to highlight and discuss SLC25A1/A4 paralogs with redundant functions. Double knockout experiments and/or combined siRNA screen may help to answer the question of whether APOE upregulation in mutants is related to canonical channeling function or some other "moonlighting" signaling mechanism specific to the particular paralogs of transporters. It is not discussed why in triple ATP/ADP carrier knockout the upregulation of APOE was only 50 % while in single SLC25A4 knockout more than 4 fold – is it due to a switch to glycolytic metabolism, the addition of pyruvate, uridine to the medium, or something else?

This is a great suggestion that we address with new experimentation in revised manuscript Figure 4 and Figure S2.

HAP1 cells express mainly SLC25A4 (Figure S2E) while HEK293 cells express preponderantly SLC25A5 and SLC25A6 as ADP/ATP carriers (Lu et al., 2017). We tested redundancy of ADP/ATP carriers in these cells. SLC25A4 KO in HAP1 cell is sufficient to disrupt the proteome, respiratory chain integrity and function, and to induce APOE. However, a SLC25A4 KO does not have any effects in HEK293 cells. This mutation minimally affects mitochondrial respiration (Lu et al., 2017). Only a KO of all three ADP/ATP carriers alters the proteome, respiratory chain integrity and function, and induces APOE in HEK293 cells (Figure 4). These findings demonstrate redundancy among carriers of similar function.

Second, we performed prolonged inhibition of ADP/ATP carriers with bongkrekic acid (Figure S2). This inhibitor targets all three ADP/ATP carriers (Gutierrez-Aguilar and Baines, 2013).

Bongkrekic acid does not induce APOE even at concentrations that decrease mitochondrial respiration by 50%. This suggests that the ADP/ATP carrier functions are dispensable for the induction of APOE. These findings point to structural defects caused by ADP/ATP carrier null mutations such as the integrity of the respiratory chain, which we document in Figure S2 and 4.

Finally, to address if there is redundancy of SLC25A1 and ADP/ATP carriers, we applied bongkrekic acid to SLC25A1 null HAP1 cells. We reasoned this mutation could sensitize SLC25A1 mutants to bongkrekic acid much like we observed for antimycin. SLC25A1 null cells are more sensitive to bongkrekic acid (Figure S2), however this toxin does not increase APOE expression. We conclude that there are no additive effects between SLC25A1 mutations and the function of ADP/ATP carriers in HAP1 cells.

4. The authors attribute APOE overexpression to copper ion levels and assembly of complex IV however experiments on cells with a defect in copper transporters lack measurements of copper; the functional redundancy of copper transporters is not discussed. Are those having functional paralogs?

We have added the copper determinations in the copper mutants as requested.

We suspect that there are adaptations in ATP7A KO cells as they have normal copper levels (Figure S3) and normal mitochondrial respiration (data not shown). We are exploring adaptive mechanisms, which is of great interest to us. We would like to request from the reviewer a waiver in discussing this issue, which we are working on for a future manuscript.

We have extended our studies to diverse copper-dependent and -independent assembly factors for complex IV (see Figure 5). All of them increased expression of APOE between ~10-40 fold. These experiments stress the point that the integrity of complex IV is necessary to maintain APOE levels in cells.

5. In the present form of the manuscript it is not clear how authors view the mode of APOE overexpression – apart from experiments showing an increase in APOE mRNA abundance there are no experiments explaining if it is due to an increase in transcription and/or mRNA stability. There are also no experiments and discussion on APOE regulation on the level of translation/degradation.

This are interesting suggestions that would require extensive experimentation in several mutants. We think such experiments would be appropriate for a different manuscript. We hope the reviewer will grant us a waiver of this extensive experimentation that we feel will not change the main point we are making in this submission.

Reviewer #2 (Recommendations for the authors):Overall, this' work is innovative for identifying dysfunctional mitochondria that can increase APOE in diverse cell types and an Alzheimer's disease mouse model. This work provides insights into how mitochondria dysfunction may influence brain and cognition function partially through modulation of APOE. While exciting, some key questions are unanswered. First, what specifically triggers increased APOE expression in the setting of various mitochondrial insults is not clear. Additionally, it is not clear how mitochondrial dysfunction is signaled to increase the expression of APOE from the nuclear genome, although some mechanisms, such as AMPK and the integrated stress response are ruled out. Finally, it is not clear what function up-regulating APOE in response to mitochondrial dysfunction serves and whether this response is adaptive or maladaptive. This study nonetheless establishes a novel link between mitochondrial dysfunction and APOE in the pathogenesis of Alzheimer's disease.While innovative and the conclusions are generally well supported and described, the manuscript would be enhanced by clarifying some unclear statements and a few additional experiments to support the authors' major findings.1. The authors demonstrate that several mitochondrial perturbations lead to APOE increases, but a unifying mitochondrial deficit among these is not identified as the trigger of APOE increase. This should be further explored with further experiments to manipulate electron transport and/or mitochondrial membrane potential. For instance, hypoxia increases APOE, which might point to decreased electron transport as a common trigger? Can decreased mitochondrial membrane potential cause APOE increase. The authors could test whether oligomycin + antimycin has a stronger effect than antimycin treatment alone, as the addition of the F1FO-ATP synthase inhibitor will keep it from running in reverse to maintain MMP in the setting of decreased electron transport. Similarly, the authors could test whether mitochondrial uncouplers such as FCCP or BAM15 similarly increase APOE expression. These dissipate the MMP but increase electron transport.

These are great suggestions. We tested the effects of FCCP, piericidin A, and oligomycin in wild type HAP1 cells. The expression of APOE was increased in piericidin A- and oligomycin-treated cells (Figure S6G). These data support a model where the function and integrity of the electron transport chain modulate APOE expression. Furthermore, the lack of effect of FCCP on APOE levels adds new strength to our assertion that the increase APOE after electron transport chain disruption is independent of the mitochondrial stress response.

We did not test a combination of oligomycin + antimycin since the 48h incubation would kill cells. We added new data that include several new assembly factors for complex IV and demonstrate defects in the integrity of the electron transport chain in SLC25A4 KO HAP1 cells and triple ADP/ATP carrier mutant HEK293 cells. These data support the hypothesis that the integrity of the electron transport chain is a factor that controls the expression of APOE.

2. Relatedly, it is suggested that the mechanistic link between SLC25A1 and SLC25A4 and APOE increase is through destabilization of respiratory complexes, in particular CIII, as well as higher-order complexes containing complexes I, III, and/or IV. To support this model in Figures 3C and D, the authors show changes in higher order respiratory complexes by BN-PAGE. The effect of SLC25A1 and SLC25A4 on the respiratory complexes, however, remains unclear. Depending on the detergent used different supercomplexes are stabilized (e.g., CI containing supercomplexes are better stabilized by Triton-X100 or digitonin compared to DDM). I was not able to find which detergent was used in these experiments, but it would be worth checking for supercomplexes using the milder detergents Triton-X100 or digitonin if this was not done. Also, only SLC25A1 KO is explored in this experiment. The authors should also assess SLC25A4. Finally, it is notable that in Figure 1K among the few shared differentially regulated proteins between SLC25A1 and SLC25A4 KO is COX7A2L in addition to CIII subunits. As COX7A2L is known to stabilize CIII-CIV within supercomplexes, the authors should explore whether this is part of the mechanism explaining the decreased formation of higher order complexes.

The description of the detergent was misplaced in the “Preparing cell lysates for immunoblot” section of the prior version. We have amended the text and these details are now in the Blue Native section of the revised version.

The data in Figure 3B were obtained with digitonin. New blue native data in in SLC25A4 KO HAP1 cells and triple ADP/ATP carrier mutant HEK293 cells use either DDM or digitonin to show the effects on these mutations on supercomplexes.

The suggestion of testing COX7A2L mutants is an outstanding idea. We tested this mutant in HEK293 cells in conjunction with a collection of complex IV assembly factor mutants. All these mutants are isogenic. COX7A2L does not change the expression of APOE (Figure 5B). However, all other complex IV assembly factors increase APOE levels >10 fold. We conclude that the formation of a COX7A2L-assisted III-IV complex does not affect the levels of APOE but the integrity of complex IV is necessary to maintain APOE expression.

3. The physiologic significance of increased APOE in response to mitochondrial dysfunction remains unclear. In figure 1, it is established that conditioned media from WT cells promotes the growth of WT and SLC25A1 KO cells while conditioned media from SLC25A1 KO cells promotes the growth of WT but not SLC25A1 KO cells. This is interpreted in the discussion to suggest that "APOE dependent removal of toxic factors [from, e.g., SLC25A1 KO cells] may be the mechanism" (in ln 619 – 621). However, it's not clear whether this is why WT conditioned media rescues the growth of SLC25A1 KO cells or whether conditioned media from SLC25A1 KO cells promotes the growth of WT cells? Don't these results suggest that WT-conditioned media is providing something missing from SLC25A1 KO media that promotes the growth of SLC25A1 cells? Additionally, it remains very speculative that this factor is related to APOE. Does knockdown of APOE from either WT or SLC25A1 cells influence the effect of the conditioned media in this assay?

We agree with the reviewer that it is not possible to conclude that "APOE dependent removal of toxic factors [from, e.g., SLC25A1 KO cells] may be the mechanism". We intended this sentence as a speculation in the discussion. We have amended the text to clearly indicate this is a speculation. Concerning the interpretation of the reviewer that “WT-conditioned media is providing something missing from SLC25A1 KO media that promotes the growth of SLC25A1 cells”, we agree that this is an alternative explanation. We concur with the reviewer that the generation of SLC25A1 and APOE double KO (DKO) cells would allow us to resolve between these two hypotheses. We are in the process of generating these DKO mutants. This will take a couple of months followed by clone isolation and their characterization. We hope the DKO clones will survive. While this is doable, the experiments in Figure 1A took 6 months once the clones were at hand and characterized as there is a need for big amounts of media collected for these experiments. This will put us with a potential answer in about a year. We would like to request a waiver for this experimental suggestion due to the prolonged nature of these experiments just to address an interesting but speculative point.

Reviewer #3 (Recommendations for the authors):This study uses multiple systems approaches using cell lines deleted for two mitochondrial carriers and identifies changes in the secretome at the transcriptional and proteome level. Among these changes, increased ApoE transcription, protein level and secretion appeared particularly interesting and conserved, leading the authors to expand their study to several other conditions of mitochondrial dysfunction. The authors analyze cell lines deleted for components of the respiratory chain, for genes involved in copper trafficking, and for astrocytes treated with mitochondrial poisons, and demonstrate increased secretion of ApoE in all these conditions, albeit in very different amounts. In addition, they correlate ApoE levels in the brain of a mouse model for Alzheimer's disease with decreased mitochondrial dysfunction. Since alleles of ApoE are known risk factors for Alzheimer's disease, the results are of potential interest.The weakness of the study is that data are entirely correlative and there is no attempt to understand the causal relationship between mitochondrial dysfunction and increased ApoE transcription and secretion.1) The statement that SLC25A1Δ cells condition media differently than wild-type cells is not supported by the data. If this were the case, the conditioned media of SLC25A1Δ cells would affect differently also WT cells. Instead, it seems from the data that the dialyzed SLC25A1Δ conditioned medium lacks some metabolite that cannot be produced by SLC25A1Δ cells. Since this effect is only evident with the dialyzed medium it is unlikely that it can be attributed to ApoE. Moreover, to really appreciate the data of Figure 1A. graphs should show the y-axis starting from 0 and show WT and KO cells in the same graph normalized to WT. How different is the growth of SLC25A1Δ cells?

We understand a metabolite as small organic molecules (molecular weight < 1500 Da, Lehninger Principles of Biochemistry 8th Edition). If this is the definition used by the reviewer, then we do not agree with the reviewer interpretation that “SLC25A1Δ conditioned medium lacks some metabolite that cannot be produced by SLC25A1Δ cells”. The reviewer’s hypothesis predicts that wild type dialyzed media should not support the growth of SLC25A1 KO cells, as such putative metabolite produced only by wild type cells would be dialyzed away. In contrast, wild type dialyzed media rescues the growth phenotype.

Concerning the statement “Since this effect is only evident with the dialyzed medium it is unlikely that it can be attributed to ApoE”. We do not agree with the reviewer. The dialysis membrane we used has a cut-off of 10kDa. APOE is 34 kDa. Thus, APOE cannot be dialyzed away.

We have modified the graphs as suggested by the reviewer showing the 0 at the y-axis. In the previous version of Figure 1A data were presented in a log2 y axis scale, now data are in a linear scale.

The reviewer asked us to present data in Figure 1A all normalized to wild type cells. We would like to request for a waiver of this request. Wild type and SLC25A1 KO cells differ in growth by 30%. We seed equal number of cells at day 0 and wait for two days to measure cell amounts after conditioned media treatments. After two days, cell numbers will be low in mutants as compared to wild types in untreated cells.

We split graphs between wild type and SLC25A1 KO cells because we opted for a statistical hypothesis testing with 2-Way ANOVA followed by multiple comparisons testing. Our 2-way ANOVA focuses on the variables conditioned media type (wild type or mutant) and plus-minus dialysis. The experiment was powered to 99.9% with n=5 for these two variables. If we merge the two graphs it will change the statistical design by adding as a variable cell genotype. Cell genotype is a variable that we are not interested in as we know mutant cells grow at a slower pace. By adding cell genotype, we would need to perform a 3-Way ANOVA. If we add cell genotype as a variable and perform 3-Way ANOVA, the current data presented in Figure 1A would become powered to a ~51%. We would have to do the experiment with n=10 to achieve a power of 99.3%. As a point of reference, the experiments in Figure 1A took 6 months due to the need to grow cells in large batches. We hope the reviewer will see our predicament.

We think that the suggested change of adding cell genotype as a variable will not alter the whole body of work presented in this manuscript. The only point we want to make is very simple: there are differences in the conditioned media between wild type and mutant cells. Finally, we would like to stress that our approach and analysis is no different from the one used by other investigators (for example see (O'Brien et al., 2000) and (Guttenplan et al., 2021))

2) Levels of ApoE in the lysate and media from different cell lines and experimental conditions range from 80ng/microg to 3pg/mg, depending on the cell line examined. The authors should comment on this high variability, especially since they try to make the case that this ApoE secretion is relevant across many models of mitochondrial dysfunction. Is this physiologically relevant in all cases?

We thank the reviewer for noticing this. We found there was a mixed up between Symbol and Arial font confusing µg and mg in our transitions between Mac and PC computers. All determinations should have been expressed as pg/µg of lysate. We amended all figures. Now the range of concentrations is within one order of magnitude. Cell lines secrete APOE at about ~1-10 x 10^4^ pg/µg lysate while human iPSC-derived astrocytes secrete ~70 x 10^4^ pg/µg lysate. Thus, we think our APOE determinations in cell lines are within a range physiological for human astrocytes, the cells that express the most APOE in brain. We have added this information to the Results section.

3) In Figure 3C, the Coomassie stained gels and the blotted ones appear very different. Also, the molecular weight markers indicate different runs, so it is unclear why there are these big differences in the western blots, but much less is apparent in the Coomassie and in panel D. The positions of the complexes should be indicated also on the Coomassie gel. I could not find an indication in the methods about the amount and type of detergent used. Are these performed using DDM or digitonin? How many times have these experiments been repeated?

We apologize for misplacing the critical information about detergent. We used digitonin in Figure 3B. We amended the text.

The differences between blots and the Coomassie we believe is due to the fact that we used a P2 mitochondrial-enriched fraction rather than isolated mitochondria. The Coomassie and blots in Figure 3B are from the same run. However, we cut the gel in two pieces, one for blotting and one for staining. We did this because the commercial MW standards for Coomassie were not suitable for MW determinations in immunoblot. We solved this issue in new Figures4 and S2 where we implemented most of the reviewer suggestions. We used DDM and digitonin. DDM was utilized to reveal multicomplexes sensitive to DDM. Furthermore, we used the same gel for Coomassie and blotting as we developed our own MW standards suitable for both Coomassie and blotting. We also changed the antibody to detect complex I to NDUFB11. We noticed that the anti NDUFS3 used in Figure 3B did not react with all complex I species in Blue Native blots.

Experiments were done twice as one dimension BN electrophoresis and once as a 2D gel.

4) In several parts of the manuscript, the authors argue that the changes in the secretome were equivalent to changes in the mitochondrial proteome as evidence for the relevance of the observed changes in the secretome. This argument seems rather weak since these omics changes can reflect a direct effect of a gene loss as well as very downstream events.

The assertion that changes in “the secretome were equivalent to changes in the mitochondrial proteome” just indicates that the number of differentially expressed proteins belonging to the annotated secretome and mitochondrial proteome is similar in their fold of enrichment even when the mutated genes are mitochondrial. We find this interesting and novel as there are few examples of secreted factors whose expression level goes up in mitochondrial mutants, such as mitokines (FGF21 and GDF15) and α-fetoprotein. The assertion does not intend to make any claims as to whether these changes are direct or indirect for either the secretome or the mitochondrial proteome. We have revised the text to make clear this point and prevent unintended interpretations of this statement.

Ultimately, what would be really important here is to demonstrate how this increased secretion of ApoE occurs and dissect at least in part how mitochondrial dysfunction affects ApoE expression transcriptionally.

Like the reviewer, we are very interested in the mechanism that connects the disruption of the electron transport chain and the nuclear expression of APOE and its secretion. We suspect that the connection between mitochondria and the expression of APOE is a novel pathway. Thus, the reviewer suggestion could be answered by an unbiased CRISPR screen looking for suppressors or enhancers of the APOE secretion phenotype. We want to do this because our candidate approach excluded several options (Figure 2, S4, S5, and S6).

We have invested most of our efforts to define triggering events at the mitochondria. A chief focus is the loss of integrity and function of the electron transport chain. In our view this is the first link in a cascade of molecular events. We would like to quote Peter Novick who stated that “a pathway of a hundred genes starts with a single mutant…” (Novick, 2014). In our case, we have deployed 14 mutants that compromise the integrity of the electron transport chain as a first step. We hope that a CRISPR screen will give us the other “hundred genes”. The magnitude of such an effort exceeds what could be done in a reasonable time. We hope the reviewer will side with us.

5) The last part of the manuscript correlates changes in the transcriptome, proteome, and interactome of SLC25A1Δ cells with the cognitive progression of AD patients. As the authors write, it is already known that AD patients' cognitive decline negatively correlates with mitochondrial function. Although catching, these data remain very correlative, and also not very surprising.

The correlation described by our collaborator Dr. Wingo is driven by 11 mitochondrial hub proteins, five of them subunits of complex I. Our findings expand this to orthogonal datasets where the hub gene is SLC25A1. These datasets are larger than the 11 hub proteins. For example, the SLC25A1 interactome is 75 proteins which are sufficient to drive the correlation. This is new and exciting information using hypothesis driven interrogation of the human Banner dataset rather than discovery proteomics as done by our collaborator in (Wingo et al., 2019). The perception of novelty is shared by our co-author, Dr. Wingo, who made the initial discovery.

Concerning, the criticism that the “data remain very correlative”, we agree. However, we do not know other way to do this on human postmortem samples where experimental options are limited. To assure the reviewer that correlative expression analyses are a sound approach, we have done hypothesis-driven interrogation of our 5xFAD mouse proteome. We tested two predictions of our model that disruption of respiratory complexes is causally linked to increased APOE. Namely, that the expression of respiratory complexes negatively correlates with the expression of APOE in a 5xFAD genotype-dependent manner. The second prediction is that disruption of respiratory complexes should precede the increase of APOE in the 5xFAD mouse model during aging. Both predictions are satisfied by the new analyses. These new findings are now presented in Figure 7.

6) In Figure S1, why is AICAR not increasing pAMPK in WT cells?

We titrated the amount of AICAR to concentrations below IC25 lethal dose for SLC25A1 mutant cells. The AICAR needed to have an effect in wild type cells is likely to kill mutant cells.

7) The title emphasizes copper-dependent and independent mitochondrial dysfunction. This puts an emphasis on copper metabolism that is totally unnecessary. The rationale all over the paper for the choice of models to use is not always very clear.

We have amended the text to explain the rationale of our choices of mutants and modified the title of the paper. We thank the reviewer for his/her suggestion to make our title more concise and precise.

8) The scheme in Figure4B should be redrawn to show Cox17 in the IMS.

We have done as indicated.

[Editors’ note: what follows is the authors’ response to the second round of review.]

The manuscript has been improved but there are some remaining issues that need to be addressed, as outlined below:The authors should aim to specify the mode of APOE abundance regulation (the original comment no 5 of Reviewer 1) with some basic experiments and a short discussion. For example, for the HAP1 SLC25A1 mutant, the increase in mRNA is 3-fold but the increase in APOE cell lysate concentration is almost 8-fold. These minor revisions may include experiments with protein degradation inhibitors to evaluate a potential regulation of the level of degradation.

We thank the editors for this interesting suggestion. We screened diverse drugs, their concentrations, and time of incubation. We searched for optimal conditions that would not compromise cell viability while allowing us to detect APOE in cell lysates and conditioned media. We found that the 8 hours of media conditioning was the shortest time where we could detect APOE levels differences observed between wild type and SLC25A1 null cells.

We used as benchmarks brefeldin A and cycloheximide because the mechanism of these two drugs made clear predictions of their effects in APOE content in cell lysates and conditioned media. For example, brefeldin A abrogated the presence of APOE from media while increasing cellular APOE levels 3-fold in both genotypes (Figure 2-supplement 1). To address the editors request, we used the lysosomal inhibitor E-64 (PMID: 7039681). We selected this lysosomal inhibitor because it did not compromise the viability of our wild type and mutant cells at concentrations used in the literature to inhibit cathepsins (PMID: 27592448- 29386126). E-64 did not modify APOE content in media from both genotypes (Figure 2-supplement 1). However, it modestly increased APOE by 20% in SLC25A1 null cells. We think these experiments make unlikely a contribution of lysosomal degradation to APOE levels.

We also tried other proteolysis inhibitors yet those prove to be toxic to cells.

The manuscript would benefit from streamlining and focusing. The authors could not dissect a mechanism, but could convincingly show that ApoE upregulation occurs across several models with different respiratory dysfunction. This should be at the center of the manuscript.

We thank the editors for this comment. We tried to address it by pointing in every section of the manuscript the main conclusion ‘genetic and pharmacological disruption of the electron transport chain causes upregulation apolipoprotein E (APOE)’. We hope that by stating this conclusion along the manuscript, we have achieved an enhanced focus.

The sentence in the Abstract: "This upregulation of secretory proteins was of similar extent as modifications to the mitochondrial annotated proteome" is confusing and prone to misunderstanding and should be rephrased.

We have done as requested. We also modified similar statements in the body of the manuscript.

The claim that changes in cholesterol synthesis are not involved is too strong. The authors have not formally proved that this is not the case in δ SLC25A1 cells. It is possible that it is beneficial for cells with respiratory defects to increase APOE secretion, but different mutants achieve this with different mechanisms.

We agree with the editors, we have softened this and similar statements accordingly.